PREPARED FOR SUBMISSION TO JHEP

April 14, 2022

# 5D and 6D SCFTs from $\mathbb{C}^3$ orbifolds

**Jiahua Tian**[a] **Yi-Nan Wang**[b,c,d]

[a] *The Abdus Salam International Centre for Theoretical Physics,*
*Strada Costiera 11, 34151, Trieste, Italy*

[b] *School of Physics and State Key Laboratory of Nuclear Physics and Technology,*
*Peking University, Beijing 100871, China*

[c] *Center for High Energy Physics, Peking University,*
*Beijing 100871, China*

[d] *Mathematical Institute, University of Oxford,*
*Andrew-Wiles Building, Woodstock Road, Oxford, OX2 6GG, UK*

*E-mail:* jtian@ictp.it, ynwang@pku.edu.cn

ABSTRACT: We study the orbifold singularities $X = \mathbb{C}^3/\Gamma$ where $\Gamma$ is a finite subgroup of $SU(3)$. M-theory on this orbifold singularity gives rise to a 5d SCFT, which is investigated with two methods. The first approach is via 3d McKay correspondence which relates the group theoretic data of $\Gamma$ to the physical properties of the 5d SCFT. In particular, the 1-form symmetry of the 5d SCFT is read off from the McKay quiver of $\Gamma$ in an elegant way. The second method is to explicitly resolve the singularity $X$ and study the Coulomb branch information of the 5d SCFT, which is applied to toric, non-toric hypersurface and complete intersection cases. Many new theories are constructed, either with or without an IR quiver gauge theory description. We find that many resolved Calabi-Yau threefolds, $\widetilde{X}$, contain compact exceptional divisors that are singular by themselves. Moreover, for certain cases of $\Gamma$, the orbifold singularity $\mathbb{C}^3/\Gamma$ can be embedded in an elliptic model and gives rise to a 6d (1,0) SCFT in the F-theory construction. Such 6d theory is naturally related to the 5d SCFT defined on the same singularity. We find examples of rank-1 6d SCFTs without a gauge group, which are potentially different from the rank-1 E-string theory.

# 1  Introduction and summary

In five-dimensional Minkowski spacetime there exists infinitely many non-trivial strongly-coupled $\mathcal{N} = 1$ superconformal field theories. The original examples of 5d SCFTs include Seiberg's rank-1 and rank-$N$ $E_n$ theories, which can be thought as the UV completions of $SU(2) + (n-1)\boldsymbol{F}$ and $Sp(N) + 1\boldsymbol{AS} + (n-1)\boldsymbol{F}$ gauge theories, respectively [1–3]. After that it was found there are two different ways to construct 5d SCFTs systematically, in the framework of superstring/M-theory. For a relatively complete overview on these approaches, see [4].

The first approach is to define a 5d SCFT $T_X^{5d}$ as M-theory on a canonical threefold singularity $X$ [5]. More specifically, one can take either an isolated singularity where $X$ is only singular at the codimension-three point $\{0\}$ or a non-isolated singularity where $X$ is also singular along codimension-two loci. For the isolated singularities, the Coulomb branch and Higgs branch information of $T_X^{5d}$ can be read off from the resolution $\widetilde{X}$ and deformation $\widehat{X}$ of the singularity [6–8]. For the non-isolated singularities, one can only directly read off the Coulomb branch from the resolution $\widetilde{X}$, but the flavor symmetry (algebra) $G_F$ is nicely encoded in the codimension-two singularities of $X$ [9–14]. Alternatively, one can directly classify the resolved space $\widetilde{X}$ according to the topology of exceptional divisors $S_i$, which are subject to the "shrinkable" condition [15–25]. This technique was further extended to include cases without a straight-forward geometric description [26, 27]. In the special cases of toric geometry, the theory has a IIA construction as well [28–32].

The second approach is to construct the 5d SCFT with a $(p, q)$ 5-brane web system in IIB superstring theory, where both the Coulomb and Higgs branch information can be extracted via the shifting of branes [33–51]. More recently, this approach was also reformulated and generalized in the framework of "generalized toric polytopes" (GTP) [52–54].

In this paper, we further explore the set of 5d SCFTs $T_X^{5d}$ defined as M-theory on orbifold singularities $X = \mathbb{C}^3/\Gamma$, where $\Gamma$ is a finite subgroup of $SU(3)$. In general, $\mathbb{C}^3/\Gamma$ has non-isolated singularities and we only study the Coulomb branch data of $T_X^{5d}$ from the resolved space $\widetilde{X}$. Mathematically, the relation between $\widetilde{X}$ and $\Gamma$ was studied under the framework of 3d McKay correspondence [55, 56], which we briefly review in section 2.1. In short, after the crepant resolution, $\widetilde{X}$ has no terminal singularity, and $b_3(\widetilde{X})=0$. The different conjugacy classes $\mathfrak{g}$ of $\Gamma$ one-to-one correspond to different cycles in $\widetilde{X}$. For each conjugacy class $\mathfrak{g}$ where each element has the eigenvalues

$$\{\lambda_i\} = \{\exp(2\pi i m) \, , \, \exp(2\pi i n) \, , \, \exp(2\pi i p)\} \quad (0 \leq m, n, p < 1), \qquad (1.1)$$

its *age* is defined as the sum $(m+n+p)$. The set of conjugacy classes with age 2 is denoted as $\Gamma_2$, which corresponds to the set of compact divisors $S_i \subset \widetilde{X}$. The set of conjugacy classes with age 1 is denoted as $\Gamma_1$ whose elements correspond to the Poincaré dual of $S_i$ and flavor 2-cycles $C_\alpha$ (corresponding to non-compact divisors $D_\alpha$) that generate the flavor symmetry $G_F$ of $T_X^{5d}$. Thus, one can do a simple group theoretic computation to get the rank $r = |\Gamma_2|$ and flavor rank $f = |\Gamma_1| - |\Gamma_2|$ of the 5d SCFT. We also propose a way to

read off codimension-two singularities of $X$ based on the structure of subgroups of $\Gamma$ in section 2.3.1.

The classification of finite subgroups $\Gamma$ of $SU(3)$ was known for a long time, following [57–60]. They were also applied in the studies of 4d quiver gauge theories [61–65] and particle physics model building [58, 66–70]. In table 1, we list the different classes of $\Gamma$ following the classifications in [71, 72]. In this paper, we also use some notations in [73]. The details of the generators for these groups are presented in section 3.

(A) The abelian subgroups, including $\mathbb{Z}_n$ and $\mathbb{Z}_m \times \mathbb{Z}_n$.

(B) The non-abelian subgroups with the generators

$$
\begin{pmatrix} \sigma_{2\times 2} & 0 \\ 0 & \det(\sigma_{2\times 2})^{-1} \end{pmatrix} \tag{1.2}
$$

where $\sigma_{2\times 2}$ are generators of a $U(2)$ subgroup. In this paper, the finite subgroup of $SU(2)$ of $D$ and $E$ types are denoted as $\mathbb{D}_n$ (the binary dihedral group with order $4n$) and $\mathbb{E}_n$ (including the binary tetrahedral group $\mathbb{T}$, the binary octahedral group $\mathbb{O}$ and the binary icosahedral group $\mathbb{I}$).

(C) These subgroups have the structure of $\Gamma = (\mathbb{Z}_m \times \mathbb{Z}_n) \rtimes \mathbb{Z}_3$, where the $\mathbb{Z}_3$ generator is the off-diagonal matrix

$$
E = \begin{pmatrix} 0 & 1 & 0 \\ 0 & 0 & 1 \\ 1 & 0 & 0 \end{pmatrix} . \tag{1.3}
$$

(D) These subgroups have the structure of $\Gamma = [(\mathbb{Z}_m \times \mathbb{Z}_n) \rtimes \mathbb{Z}_3] \rtimes \mathbb{Z}_2$, where the $\mathbb{Z}_3$ generator is $E$ and the $\mathbb{Z}_2$ generator is

$$
I = \begin{pmatrix} 0 & 0 & 1 \\ 0 & 1 & 0 \\ 1 & 0 & 0 \end{pmatrix} . \tag{1.4}
$$

(E) An exceptional subgroup of $SU(3)$ of order 108, which is also denoted as $H_{36}$.

(F) An exceptional subgroup of $SU(3)$ of order 216, which is denoted as $H_{72}$.

(G) An exceptional subgroup of $SU(3)$ of order 648, which is denoted as $H_{216}$.

(H) A subgroup of order 60, $H_{60}$, which is isomorphic to the alternating group $\mathbb{A}_5$.

(I) A subgroup of order 168, $H_{168}$, which is isomorphic to the permutation group generated by $(1234567)$, $(142)(356)$ and $(12)(35)$.

(L) An exceptional subgroups of $SU(3)$ of order 1080, which is denoted as $H_{360}$.

| Class | $\Gamma \in SU(3)$ | $|\Gamma|$ |
|---|---|---|
| (A) | $\mathbb{Z}_n$ | $n$ |
| | $\mathbb{Z}_m \times \mathbb{Z}_n$ | $mn$ |
| (B) | $G_m$ | $8m$ |
| | $G_{p,q}$ | $8pq^2$ |
| | $G'_m$ | $8m$ |
| | $E^{(1)}$ | 72 |
| | $E^{(2)}$ | 24 |
| | $E^{(3)}$ | 96 |
| | $E^{(4)}$ | 48 |
| | $E^{(5)}$ | 96 |
| | $E^{(6)}$ | 48 |
| | $E^{(7)}$ | 144 |
| | $E^{(8)}$ | 192 |
| | $E^{(9)}$ | 240 |
| | $E^{(10)}$ | 360 |
| | $E^{(11)}$ | 600 |
| (C) | $\Delta(3n^2)$ | $3n^2$ |
| | $C_{n,l}^{(k)}$ | $3nl$ |
| | $D_{3l,l}^{(1)}, l = 2k$ | $9l^2$ |
| (D) | $\Delta(6n^2)$ | $6n^2$ |
| (E) | $H_{36} = \Sigma_{36\times3}$ | 108 |
| (F) | $H_{72} = \Sigma_{72\times3}$ | 216 |
| (G) | $H_{216} = \Sigma_{216\times3}$ | 648 |
| (H) | $H_{60} = \Sigma_{60}$ | 60 |
| (I) | $H_{168} = \Sigma_{168}$ | 168 |
| (L) | $H_{360} = \Sigma_{360\times3}$ | 1080 |

**Table 1**: Classification of finite subgroups $\Gamma$ of $SU(3)$ and the list of $\Gamma$ used in this paper with their order $|\Gamma|$.

For a given $\Gamma$, one can write down all the invariant polynomials $F_i$ ($i = 1, \ldots, p$) in terms of the complex coordinates $(Z_1, Z_2, Z_3)$ of $\mathbb{C}^3$. Then the singular equation of $X = \mathbb{C}^3/\Gamma$ can be written as $q$ independent relations among $F_i$. If $q = 1$, $p = 4$, $X$ is a hypersurface. If $q > 1$ and $(p - q) = 3$, $X$ is a complete intersection of $q$ equations. If $q > 1$ and $(p - q) < 3$, $X$ is a non-complete intersection of $q$ equations. For example, for the group $\Gamma = \mathbb{Z}_n \times \mathbb{Z}_n$ with generators

$$\mathrm{diag}(\omega, 1, \omega^{-1}) \ , \ \mathrm{diag}(\omega, \omega^{-1}, 1) \quad (\omega = e^{2\pi i/n}) \tag{1.5}$$

acting on $(Z_1, Z_2, Z_3)$, the invariant polynomials are

$$F_1 = Z_1^n \ , \ F_2 = Z_2^n \ , \ F_3 = Z_3^n \ , \ F_4 = Z_1 Z_2 Z_3 \ . \tag{1.6}$$

There is a single relation among $F_i$, which is the hypersurface description of the singularity $\mathbb{C}^3/(\mathbb{Z}_n \times \mathbb{Z}_n)$:

$$F_1 F_2 F_3 = F_4^n \,. \tag{1.7}$$

In this paper, we only explicitly study the the following cases, where the resolution $\widetilde{X}$ is easier to perform:

1. Abelian groups $\Gamma = \mathbb{Z}_n$ or $\mathbb{Z}_m \times \mathbb{Z}_n$. In these cases, $X = \mathbb{C}^3/\Gamma$ is always toric, which can be resolved in the standard triangulation method. We discuss the geometry and physics of these cases in section 3.1.

2. $X$ is a hypersurface in $\mathbb{C}^4$, where the resolution procedure is reviewed in section 2.2.1.

3. $X$ is a complete intersection of two equations in $\mathbb{C}^5$, where the resolution procedure is discussed in section 2.2.2.

The case 2 and 3 were classified in [71, 74], where the notations for $\Gamma$ are different. We list the 5d rank $r$ and flavor rank $f$ of $T_X^{5d}$ and the codimension-two ADE singularities of $X$ in table 2. Note that we only listed the cases of non-abelian $\Gamma$ there, while examples of abelian $\Gamma$ can be found in table 5 and 6.

For some of the cases, the singular equation $X$ can also be embedded in a non-compact elliptic Calabi-Yau threefold. Namely, if $X$ can be rewritten as

$$y^2 = x^3 + fx + g \,, \tag{1.8}$$

where $f$ and $g$ are holomorphic functions in other coordinates, then it can be embedded in the singular Weierstrass model

$$y^2 = x^3 + fxZ^4 + gZ^6 \,. \tag{1.9}$$

$[x : y : Z]$ are the projective coordinates of $\mathbb{P}^{2,3,1}$, and $Z = 0$ is the zero section of the elliptic CY3. In this description, we can also define the 6d (1,0) SCFT $T_X^{6d}$ and study its tensor branch generated by blowing up the base $B_2$ [75–78]. In section 2.5 we explicitly give the relation between $T_X^{6d}$ and $T_X^{5d}$ from the same singularity $X$. Namely, if $T_X^{6d}$ is very-Higgsable, in the terminology of [79], $T_X^{5d}$ is generated by decoupling a matter hypermultiplet from the 5d KK theory of $T_X^{6d}$. If $T_X^{6d}$ is non-very-Higgsable, $T_X^{5d}$ is constructed by decoupling vector multiplets from the 5d KK theory of $T_X^{6d}$. This lines up with the philosophy of [13].

We present examples of this 6d/5d correspondence in section 4.1, 4.2 and 4.3. In particular, in the case of $\Delta(48)$ (section 4.2.1), the tensor branch takes the form of type $I_1$ singular fiber on a $(-1)$-curve. We argue that it corresponds to a 6d SCFT that is potentially different from the rank-1 E-string, using both geometry and anomaly arguments. Similar arguments can be applied to the case of $H_{168}$ as well, whose tensor branch takes the form of type $II$ singular fiber on a $(-1)$-curve, see appendix B.5.1.

For a 5d SCFT $T_X^{5d}$ from $X = \mathbb{C}^3/\Gamma$, a question of interest is how to read off the global symmetry from group theoretic data of $\Gamma$. We separate this question into the usual (0-form) flavor symmetry and the 1-form symmetry.

| $\Gamma$ | | $r$ | $f$ | codimension 2 ADE |
|---|---|---|---|---|
| $\Delta(3n^2)$ | $3 \mid n$ | $\frac{1}{6}\left(n^2 - 3n + 6\right)$ | $n + 5$ | $A_{n-1} \times A_2^3$ |
| | $3 \nmid n$ | $\frac{1}{6}\left(n^2 - 3n + 2\right)$ | $n + 1$ | $A_{n-1} \times A_2$ |
| $\Delta(6n^2),\, n = 2k+1$ | $3 \mid n$ | $\frac{1}{12}\left(n^2 + 6n - 3\right)$ | $\frac{1}{2}n + \frac{7}{2}$ | $A_{n-1} \times A_1 \times A_2^3$ |
| | $3 \nmid n$ | $\frac{1}{12}\left(n^2 + 6n - 7\right)$ | $\frac{1}{2}n + \frac{3}{2}$ | $A_{n-1} \times A_1 \times A_2$ |
| $\Delta(6n^2),\, n = 2k$ | $3 \mid n$ | $\frac{1}{12}\left(n^2 + 6n - 12\right)$ | $\frac{1}{2}n + 5$ | $D_{n/2+2} \times A_1 \times A_2^3$ |
| | $3 \nmid n$ | $\frac{1}{12}\left(n^2 + 6n - 16\right)$ | $\frac{1}{2}n + 3$ | $D_{n/2+2} \times A_1 \times A_2$ |
| $C_{3l,l}^{(1)}$ | | $\frac{1}{2}l^2 - \frac{1}{2}l + 1$ | $l + 5$ | $A_{l-1} \times A_2^3$ |
| $C_{7l,l}^{(2)}$ | $3 \mid l$ | $\frac{1}{6}\left(7l^2 - 3l + 6\right)$ | $l + 5$ | $A_{l-1} \times A_2^3$ |
| | $3 \nmid l$ | $\frac{1}{6}\left(7l^2 - 3l + 2\right)$ | $l + 1$ | $A_{l-1} \times A_2$ |
| $D_{3l,l}^{(1)},\, l = 2k$ | | $\frac{1}{4}l^2 + 2l - 1$ | $\frac{1}{2}l + 5$ | $D_{l/2+2} \times A_1 \times A_2^2$ |
| $G_m$ | $2 \mid m$ | $\frac{1}{2}m$ | $m + 5$ | $D_{m+2} \times A_1^3$ |
| | $2 \nmid m$ | $\frac{1}{2}m - \frac{1}{2}$ | $m + 3$ | $D_{m+2} \times A_1^2$ |
| $G_{p,q}$ | $2 \mid p$ | $pq^2 - \frac{1}{2}pq + 2q - 2$ | $pq + 2q + 3$ | $D_{pq+2} \times A_1^2 \times A_{2q-1}$ |
| | $2 \nmid p$ | $pq^2 - \frac{1}{2}pq + \frac{1}{2}q - 1$ | $pq + 2q + 1$ | $D_{pq+2} \times A_1 \times A_{2q-1}$ |
| $G_m'$ | $2 \mid m$ | $\frac{1}{2}m$ | $m + 2$ | $D_{m+2} \times A_1$ |
| | $2 \nmid m$ | $\frac{1}{2}(m+1)$ | $m + 4$ | $D_{m+2} \times A_1^2$ |
| $E^{(1)}$ | | $5$ | $10$ | $E_6 \times A_2^2$ |
| $E^{(2)}$ | | $1$ | $4$ | $D_4 \times A_2$ |
| $E^{(3)}$ | | $3$ | $9$ | $E_7 \times A_1^2$ |
| $E^{(4)}$ | | $1$ | $5$ | $E_6 \times A_1$ |
| $E^{(5)}$ | | $4$ | $7$ | $E_6 \times A_3$ |
| $E^{(6)}$ | | $3$ | $7$ | $E_6 \times A_1$ |
| $E^{(7)}$ | | $7$ | $9$ | $E_7 \times A_2$ |
| $E^{(8)}$ | | $10$ | $11$ | $E_7 \times A_3 \times A_1$ |
| $E^{(9)}$ | | $4$ | $9$ | $E_8 \times A_1$ |
| $E^{(10)}$ | | $8$ | $10$ | $E_8 \times A_2$ |
| $E^{(11)}$ | | $16$ | $12$ | $E_8 \times A_4$ |
| $H_{36}$ | | $4$ | $5$ | $A_3 \times A_2 \times A_2$ |
| $H_{60}$ | | $0$ | $4$ | $A_4 \times A_2 \times A_1$ |
| $H_{72}$ | | $5$ | $5$ | $D_4 \times A_2$ |
| $H_{168}$ | | $1$ | $3$ | $A_3 \times A_2$ |
| $H_{216}$ | | $9$ | $5$ | $D_4 \times A_2^2$ |
| $H_{360}$ | | $5$ | $6$ | $A_4 \times A_3 \times A_2^2$ |

**Table 2**: Summary of the data associated to the non-abelian groups $\Gamma$ covered in this paper.

| $\Gamma$ | $r$ | $f$ | $G_F$ | IR description |
|---|---|---|---|---|
| $G_{2k}$ | $k$ | $2k+5$ | $SO(4k+8) \times SU(2)$ | $Sp(k) + (2k+4)\boldsymbol{F}$ |
| $G'_{2k+1}$ | $k+1$ | $2k+5$ | $SO(4k+10)$ | $4\boldsymbol{F} - SU(2) - SU(2)_0 - \cdots - SU(2)_0$ |
| $E^{(6)}$ | $3$ | $8$ | $E_6 \times SU(2) \times U(1)$ | $SU(3)_0 - SU(2) - 5\boldsymbol{F}$ |
| $\Delta(27)$ | $1$ | $8$ | $E_8$ | $SU(2) + 7\boldsymbol{F}$ |
| $\Delta(54)$ | $2$ | $5$ | $Sp(4) \times Sp(1)$ | $G_2 + 4\boldsymbol{F}$ |
| $H_{36}$ | $4$ | $5$ | $SU(4) \times Sp(2)$ | $\begin{array}{c} 2\boldsymbol{F} \\ \mid \\ SU(2)_0 - G_2 - SU(2)_0 \end{array}$ |

**Table 3**: A number of $T_X^{5d}$ with known 5d gauge theory descriptions.

1. Lie algebra of the 0-form flavor symmetry $G_F$

   We find out that $G_F$ may be different from the factors read off directly from codimension-two singularities. These subtleties are discussed in section 2.3.

   As an example, in section 4.2.2 we discuss the case of $\Delta(48)$. The singularity $X = \mathbb{C}^3/\Delta(48)$ has a type $A_3$ and a type $A_2$ codimension-two singularity, which naively gives rise to a rank-1 theory with $G_F = SU(4) \times SU(3)$. Nonetheless, due to the splitting of flavor curve on the compact divisor $S_1$, the flavor symmetry factor $SU(4)$ is broken to $SU(2)^2 \times U(1)$.

2. 1-form symmetry $\Gamma^{(1)}$

   The higher form symmetry of 5d SUSY quantum field theories has been studied in [6–8, 80–83], using geometric and field theoretic approaches. In this paper, we propose a method to compute the electric 1-form symmetry $\Gamma^{(1)}$ of $T_X^{5d}$ using the McKay quiver of $\Gamma$, see section 2.4. Most of the cases with a non-trivial $\Gamma^{(1)}$ are the abelian ones, $\Gamma = \mathbb{Z}_n$, and the results are cross-checked with toric geometry methods in section 3.1. We present a partial list of examples in table 5.

We list a number of theories with their IR gauge theory descriptions and $G_F$ in table 3, where $\Gamma$ is non-abelian. For the cases of abelian $\Gamma$, see table 5 and table 6 for a (non-complete) list.

Besides these cases, we also studied the resolutions for a number of higher rank SCFTs. For $\Delta(3n^2)$ theory with $n = 3k$, the flavor symmetry is enhanced to $G_F = SU(3k) \times E_6$. For example, the $\Delta(108)$ theory is a rank-4 SCFT with $G_F = SU(6) \times E_6$ and the triple intersection numbers in (4.18). The $\Delta(243)$ theory is a rank-10 SCFT with $G_F = SU(9) \times E_6$ and triple intersection numbers in (4.23).

Moreover, the $C_{3,1}^{(3)}$ theory is a rank-4 SCFT with $f = 8$, the triple intersection numbers are shown in (B.103). The $D_{6,2}$ theory is a rank-4 SCFT with $f = 6$, the triple intersection numbers are shown in (B.107). We also presented the resolution sequences for $\Delta(6n^2)$, $n = 2k$, $H_{72}$ and $H_{216}$ without computing the triple intersection numbers in appendix B.

The structure of the paper is as follows: in section 2 we discuss the general mathematical setup of McKay correspondence, resolution of singularities, flavor symmetry, the

way of computing 1-form symmetry and the 6d/5d correspondence. In section 3 we give a list of finite subgroups of $SU(3)$ that we study in this paper. In particular the section 3.1 covers the cases of abelian $\Gamma$, which corresponds to toric Calabi-Yau threefolds, and the 5d physics is also discussed. In section 4 we present a number of notable examples for the 6d/5d correspondence, where the 6d tensor branch geometry and the 5d resolved geometry are discussed in details. In appendix A we list a number of $\Gamma$ and their maximal abelian subgroups, from which one can read off the codimension-two singularities of $\mathbb{C}^3/\Gamma$. In appendix B we present the details of other non-abelian $\Gamma$ examples, their resolution geometry and 6d/5d physics.

## 2 3d McKay correspondence, resolution and physics

### 2.1 Ito-Reid theorem on $\mathbb{C}^3/\Gamma$ orbifolds

In this section, we review Ito-Reid's result on the McKay correspondence of $\mathbb{C}^3/\Gamma$ orbifolds [84], where $\Gamma$ is a finite subgroup of $SU(3)$. We denote the conjugacy classes of $\Gamma$ by $\mathfrak{g}_1, \mathfrak{g}_2, \ldots, \mathfrak{g}_n$. All the group elements $g \in SL(3, \mathbb{C})$ in the same conjugacy class $\mathfrak{g}_i$ have the same eigenvalues $\{\lambda_i\}$. If these group elements are all of order $r$:

$$g^r = 1 \,, \tag{2.1}$$

then the eigenvalues can be written as

$$\{\lambda_i\} = \left\{ \exp\left(\frac{2\pi i a}{r}\right), \exp\left(\frac{2\pi i b}{r}\right), \exp\left(\frac{2\pi i c}{r}\right) \right\} := \frac{1}{r}(a, b, c) \ (0 \le a, b, c < r) \,. \tag{2.2}$$

We define the *age* of a group element $g$ and its corresponding conjugacy class $\mathfrak{g}_i$ to be

$$\mathrm{age}(g) = \mathrm{age}(\mathfrak{g}_i) := \frac{1}{r}(a + b + c) \,. \tag{2.3}$$

Hence the identity element $\mathbf{1}$ has

$$\mathrm{age}(\mathbf{1}) = 0 \tag{2.4}$$

and is the only element whose age is 0. The other group elements either have $\mathrm{age}(g) = 1$, which are called *junior* elements, or $\mathrm{age}(g) = 2$. We denote the set of conjugacy classes $\mathfrak{g}_i$ with $\mathrm{age}(\mathfrak{g}_i) = m$ by $\Gamma_m$.

Ito-Reid proved that for every $\mathbb{C}^3/\Gamma$ orbifold, there exists a crepant resolution

$$\phi : \widetilde{X} \to \mathbb{C}^3/\Gamma \,, \tag{2.5}$$

such that

$$\begin{aligned} b_{2m}(\widetilde{X}) &= |\Gamma_m| \quad (m = 0, 1, 2) \\ b_3(\widetilde{X}) &= 0 \,, \end{aligned} \tag{2.6}$$

and there is no residual terminal singularity.

Hence the total number of conjugacy classes is equal to the Euler characteristic of $\widetilde{X}$:

$$\begin{aligned} \chi(\widetilde{X}) &= b_0(\widetilde{X}) + b_2(\widetilde{X}) + b_4(\widetilde{X}) \\ &= 1 + |\Gamma_1| + |\Gamma_2| \,. \end{aligned} \tag{2.7}$$

Namely, there is a one-to-one correspondence between the 2-cycles in $\widetilde{X}$ and the junior conjugacy classes in $\Gamma_1$. There is also a one-to-one correspondence between 4-cycles in $\widetilde{X}$ and the conjugacy classes in $\Gamma_2$. In the language of 5d SCFT, it means that for a 5d SCFT constructed from M-theory on the orbifold singularity $\mathbb{C}^3/\Gamma$, its rank $r$ and flavor rank $f$ can be computed group theoretically [1]:

$$
\begin{aligned}
r &= b_4(\widetilde{X}) \\
&= |\Gamma_2|
\end{aligned}
\tag{2.8}
$$

$$
\begin{aligned}
f &= b_2(\widetilde{X}) - b_4(\widetilde{X}) \\
&= |\Gamma_1| - |\Gamma_2| \,.
\end{aligned}
\tag{2.9}
$$

In practice, the conjugacy classes corresponding to $f$ can be computed in the following way. For any junior conjugacy class $\mathfrak{g}_i \in \Gamma_1$ with elements $g_{i,1}, \ldots, g_{i,m}$. The inverse elements $g_{i,1}^{-1}, \ldots, g_{i,m}^{-1}$ all belong to the same conjugacy class $\mathfrak{g}_j$. There are the following two cases:

1. $\mathfrak{g}_j \in \Gamma_2$. $\mathfrak{g}_i$ and $\mathfrak{g}_j$ correspond to a pair of 2-cycle and 4-cycle that are Poincaré dual to each other.

2. $\mathfrak{g}_j \in \Gamma_1$, $\mathfrak{g}_i$ corresponds to a flavor curve, which is Poincaré dual to a non-compact 4-cycle $D_i$. Hence it contributes to the flavor rank $f$. We denote the set of these conjugacy classes by $\Gamma_{1,f}$.

## 2.2 Resolution

In this section, we review the basics of resolution techniques used in this paper, which can be applied to more general canonical threefold singularities as well. Note that for a given singularity, we always choose a crepant resolution $\phi : \widetilde{X} \to \mathbb{C}^3/\Gamma$ such that all compact 2-cycles are contained inside the compact 4-cycles (for the cases of $b_4(\widetilde{X}) > 0$). Physically, the resulting configuration of compact surfaces describes the Coulomb branch of $T_X^{5d}$ with the maximal flavor rank $f$, which matches the group theory result (2.9). For the cases of abelian $\Gamma$, the threefold singularity is always toric, and one can apply the standard toric resolution as a maximal triangulation of polygons [85–87], see section 3.1.

### 2.2.1 Hypersurface

Starting from a singular hypersurface equation $X \subset \mathbb{C}^4$, we consider the crepant resolution of singularity $X$

$$
\phi : \widetilde{X} \to X \,, \quad K_{\widetilde{X}} = \phi^* K_X \,.
\tag{2.10}
$$

The resolution involves a sequence of blow-ups of the ambient space. We use the notation in [6, 11, 88], and the possible blow-ups are [89, 90]:

---

[1]Because $b_3(\widetilde{X}) = 0$, there is no extra contribution to $f$ as in [6].

1. Weighted blow-up of the locus $x_1 = x_2 = x_3 = x_4 = 0$ on the ambient space, with weights $(a_1, a_2, a_3, a_4) = (3, 2, 1, 1)$, $(2, 1, 1, 1)$ or $(1, 1, 1, 1)$. This blow-up is denoted by $(x_1^{(a_1)}, x_2^{(a_2)}, x_3^{(a_3)}, x_4^{(a_4)}; \delta)$, or simply $(x_1, x_2, x_3, x_4; \delta)$ iff the weights are exactly $(1, 1, 1, 1)$. The exceptional divisor on the ambient space is $\delta = 0$, which has the topology $\mathbb{P}^{a_1, a_2, a_3, a_4}$. After the blow-up, the singular equation is properly transformed by $x_i \to x_i \delta^{a_i}$, and the whole equation is divided by $\delta^{\sum_{i=1}^4 a_i - 1}$. A new SR ideal generator $x_1 x_2 x_3 x_4$ appears as well.

2. Blow-up of the locus $x_1 = x_2 = x_3 = 0$ on the ambient space, with weights $(w_1, w_2, w_3) = (1, 1, 1)$. The blow-up is denoted by $(x_1, x_2, x_3; \delta)$. After the blow-up, the singular equation is properly transformed by $x_i \to x_i \delta$, and the whole equation is divided by $\delta^2$. A new SR ideal generator $x_1 x_2 x_3$ appears.

3. Blow-up of the locus $x_1 = x_2 = 0$ on the ambient space, with weights $(w_1, w_2) = (1, 1)$. The blow-up is denoted by $(x_1, x_2; \delta)$. After the blow-up, the singular equation is properly transformed by $x_i \to x_i \delta$, and the whole equation is divided by $\delta$. A new SR ideal generator $x_1 x_2$ appears.

After the resolution sequence, if the SR ideal does not involve polynomial generators, then the ambient space $\Sigma$ can still be considered as toric. The intersection numbers among divisors $S_{\Sigma, i}$ on $\Sigma$ are computed in the standard way, and the intersection numbers of divisors $S_i$ on the anticanonical hypersurface $\widetilde{X}$ can be computed as

$$S_i \cdot S_j \cdot S_k = -K_\Sigma \cdot S_{\Sigma, i} \cdot S_{\Sigma, j} \cdot S_{\Sigma, k} \,. \tag{2.11}$$

Subtlety arises when the blow-up locus is singular by itself. For example, if we try to resolve the equation $F(x, y, z, w) = 0$ at the codimension-two locus

$$x = y = f(z, w) = 0 \,, \tag{2.12}$$

where $f(z, w)$ is singular at $z = w = 0$. In this case, we can define a new variable

$$U \equiv f(z, w) \,, \tag{2.13}$$

and the hypersurface equation is equivalent to a complete intersection

$$\begin{cases} F(x, y, z, w, U) = 0 \\ U - f(z, w) = 0 \,. \end{cases} \tag{2.14}$$

After the blow-up $(x, y, f(z, w); \delta)$, the proper transform of second equation is

$$U\delta - f(z, w) = 0 \,, \tag{2.15}$$

which is still singular at

$$U = \delta = z = w = 0 \,, \tag{2.16}$$

and additional blow-ups are required.

In general, if the resolution locus is singular, one always need to introduce a new variable and transform the hypersurface equation into a complete intersection.

### 2.2.2 Complete intersection in ambient 5-fold

In this section, we consider the canonical threefold singularity $X$ given by the complete intersection of two equations $f_1$ and $f_2$ in $\mathbb{C}^5$. One can similarly resolve $X$ by blowing up the fivefold ambient space $\Sigma$. We denote a blow-up of variables $x_1, x_2, \ldots, x_k$ with weight $(a_1, a_2, \ldots, a_k)$ by

$$\left(x_1^{(a_1)}, x_2^{(a_2)}, \ldots, x_k^{(a_k)}; \delta\right), \tag{2.17}$$

where $\delta = 0$ is the exceptional divisor $E$. If $a_1 = a_2 = \cdots = a_k = 1$, the subscripts of $x_i$ can be omitted. Then after the replacement

$$x_i = x_i' \delta^{a_i} \quad (i = 1, \ldots, k), \tag{2.18}$$

$f_1$ and $f_2$ takes the form of

$$f_j = f_j' \delta^{p_j} \quad (j = 1, 2). \tag{2.19}$$

$f_j'$ is an irreducible holomorphic polynomial in the new variable $x_i$. The new complete intersection $X'$ is given by $f_1' = f_2' = 0$.

Now we derive the condition for crepant resolution. Denote the divisors associated to $x_i$ by $D_i$, hence the new divisor associated to $x_i'$ is $D_i - a_i E$ after the blow-up. We denote the divisor class of $f_i = 0$ in $X$ by $F_i$. Before the blow-up, from the adjunction formula the canonical class of $X$ is given by

$$K_X = (K_\Sigma + F_1 + F_2) \cdot F_1 \cdot F_2. \tag{2.20}$$

The Calabi-Yau condition of $X$ means $K_\Sigma = -F_1 - F_2$. After the blow up, $K_\Sigma$ is transformed as

$$K_{\Sigma'} = \phi^*(K_\Sigma) + \sum_{i=1}^{k} a_i E - E. \tag{2.21}$$

Hence the canonical class of the resolved $X'$

$$K_{X'} = (K_{\Sigma'} + F_1' + F_2') \cdot F_1' \cdot F_2' \tag{2.22}$$

is trivial iff $K_{\Sigma'} + F_1' + F_2' = 0$. Since the proper transformation of $F_i$,

$$F_i' = F_i - p_i E, \tag{2.23}$$

the condition for a crepant resolution is

$$\sum_{i=1}^{k} a_i - 1 = p_1 + p_2. \tag{2.24}$$

After the blow-up, we replace the notations of $F'$, $x'$ and $f'$ by $F$, $x$ and $f$ to simplify the notations. Also, a new SR ideal generator $x_1 x_2 \ldots x_k$ in the ambient space will be added.

The triple intersection numbers of divisors $S_i$ in $\widetilde{X}$ can be computed from the ambient space divisors $S_{\Sigma_i}$ as

$$S_i \cdot S_j \cdot S_k = F_1 \cdot F_2 \cdot S_{\Sigma,i} \cdot S_{\Sigma,j} \cdot S_{\Sigma,k} \tag{2.25}$$

Let us present a simple example of resolving a complete intersection, and computing the triple intersection number of compact divisors in the resolved CY3. Consider the complete intersection of two generic quadratic equations in $\mathbb{C}^5$:

$$\begin{cases} f_1(x, y, z, u, v) = 0 \\ f_2(x, y, z, u, v) = 0 \,. \end{cases} \tag{2.26}$$

The crepant resolution is

$$(x, y, z, u, v; \delta_1) \,. \tag{2.27}$$

As one can check, $p_1 = p_2 = 2$ in this case, and the crepant condition (2.24) holds.

In the original $\mathbb{C}^5$, the non-compact divisor associated to $x = 0$, $y = 0$, $z = 0$, $u = 0$ and $v = 0$ are all linearly equivalent, which is denoted as $D$. Its self-intersection number is

$$D^5 = 1 \,. \tag{2.28}$$

After blowing up the 5d ambient space, these divisors are transformed to $D - E_1$, where $E_1 : \delta_1 = 0$ is the exceptional divisor in the ambient space. Because we have an SR ideal $\{xyzwv\}$, the following intersection number vanishes:

$$(D - E_1)^5 = 0 \,. \tag{2.29}$$

In this case, all the crossing terms between $D$ and $E_1$ vanish, and the only non-vanishing intersection number involving $E_1$ is

$$E_1^5 = 1 \,. \tag{2.30}$$

Now we can evaluate the triple intersection number $S_1^3$, where $S_1 = E_1 \cdot F_1 \cdot F_2$ is the compact complex surface in the resolved CY3 $\widetilde{X}$. As $F_1 = F_2 = -2E_1$ (2.23), this number is computed as

$$\begin{aligned} S_1^3 &= E_1^3 \cdot F_1 \cdot F_2 \\ &= 4E_1^5 \\ &= 4 \,. \end{aligned} \tag{2.31}$$

From the self-triple intersection number, we can see that topology of $S_1$ is $dP_5$. This is consistent with the definition of $dP_5$ as the complete intersection of two quadratics in $\mathbb{P}^4$.

### 2.2.3 Applicability of intersection number computations

Here we comment on the applicability of triple intersection number computations described in section 2.2.1 and 2.2.2, based on toric ambient spaces. If one tries to compute the triple intersection numbers among all the compact divisors $S_i$ and non-compact divisors $D_\alpha$ in $\widetilde{X}$, one needs to work out the full resolution. The full projective relations (and SR ideal) would generally involve polynomial terms, and in such cases the resolved CY3 $\widetilde{X}$ cannot be taken as a hypersurface or complete intersection inside a toric ambient space.

Nonetheless, if one only wants the triple intersection numbers among the compact divisors $S_i$, in many cases there exists a partial resolution $\phi_1 : X_1 \to X$ where the exceptional loci contains all the desired compact divisors $S_i$:

$$\phi_1^{-1}(0) = \bigcup_{i=1}^r S_i \,. \tag{2.32}$$

The projective relations on $X_1$ only involves monomial terms, and $X_1$ can still be thought as a hypersurface or complete intersection inside a toric ambient space. The partially resolved $X_1$ still contains codimension-two singularities at the subset $s_1 \in X_1$, and after the resolution $\phi_2 : \widetilde{X} \to X$, we get the fully resolved $\widetilde{X}$ with exceptional locus

$$\phi_2^{-1}(s_1) = \bigcup_{i=1}^{f} D_\alpha \,. \tag{2.33}$$

If the resolution $\phi_2$ does not involve the blowing up of ambient space divisors $S_{\Sigma,i}$ corresponding to $S_i$, the triple intersection numbers $S_i \cdot S_j \cdot S_k$ are unchanged through the resolution $\phi_2$, and one can compute them via the partial resolution $\phi_1$.

As a positive example, consider the singularity corresponding to $\Delta(27)$ in section 4.1.2:

$$w^6 - 3w^4 x + 3w^2 x^2 - x^3 - 4w^3 y + 4wxy + 8y^2 - 16w^3 z^3 + 24wxz^3 + 24yz^3 + 72z^6 = 0 \,. \tag{2.34}$$

The first partial resolution $\phi_1$ is given by a weighted blow-up $(x^{(2)}, y^{(3)}, z^{(1)}, w^{(1)}; \delta_1)$ in the toric ambient space. After this step, the partially resolved $X_1$ only has codimension-two singularity at non-toric loci, which can be further resolved to get the eight non-compact exceptional divisors. The point is, the second step of resolution $\phi_2$ does not involve the blowing up of $\delta_1$, thus it does not change the triple intersection number $S_1^3$ of $S_1 : \delta_1 = 0$ computed in the first step, and we still have

$$S_1^3 = 1 \,. \tag{2.35}$$

$S_1$ indeed has the topology of a generalized $\mathrm{dP}_8$ as a degree-6 hypersurface in the weighted projective space $\mathbb{P}^{2,3,1,1}$.

As a negative example, consider the singularity corresponding to $\Delta(48)$ in section 4.2.2:

$$w^6 - 3w^4 x + 3w^2 x^2 - x^3 - 4w^3 y + 4wxy + 8y^2 - 16w^3 z^4 + 24wxz^4 + 24yz^4 + 72z^8 = 0 \,. \tag{2.36}$$

After the partial resolution $\phi_1$ given by the same blow-up $(x^{(2)}, y^{(3)}, z^{(1)}, w^{(1)}; \delta_1)$, the resulting equation

$$w^6 - 3w^4 x + 3w^2 x^2 - x^3 - 4w^3 y + 4wxy + 8y^2 + (-16w^3 z^4 + 24wxz^4 + 24yz^4)\delta_1 + 72z^8 \delta_1^2 = 0 \,. \tag{2.37}$$

has a codimension-two singularity at

$$x = y = w^3 - 27z^4 \delta_1 = 0 \,. \tag{2.38}$$

There is no way to resolve this singularity without blowing up the $\delta_1$ coordinate. Hence one cannot simply apply the triple intersection number computation of $S_1^3$ from the toric ambient space.

## 2.3 Codimension-two singularities and flavor symmetry

### 2.3.1 Codimension-two singularity from group structure of $\Gamma$

In this section, we propose a procedure of reading off codimension-two singularities of $\mathbb{C}^3/\Gamma$, which are labeled by ADE Lie algebra $\mathfrak{g}_{ADE}$, from the group theoretic data of $\Gamma$.

*Conjecture 1*: consider the maximal subgroups $\widetilde{G}_i \in \Gamma$ of type $\mathbb{Z}_N$, $\mathbb{D}_N$, $\mathbb{E}_6$, $\mathbb{E}_7$ and $\mathbb{E}_8$, where every non-identity element belongs to the set $\Gamma_{1,f}$. We define an equivalence relation $\sim$ among $\widetilde{G}_i$: $\widetilde{G}_i \sim \widetilde{G}_j$ if and only if there exists group isomorphism $f : \widetilde{G}_1 \to \widetilde{G}_2$, such that $\forall g \in \widetilde{G}_1$, $f(g)$ and $g$ belong to the same conjugacy class of $\Gamma$. Then each equivalence class of $\widetilde{G}_i$ one-to-one corresponds to a codimension-two singularity of $\mathbb{C}^3/\Gamma$. If $\widetilde{G}_i = \mathbb{Z}_N$, $\mathbb{D}_N$ or $\mathbb{E}_N$, the codimension-two singularity has ADE type $A_{N-1}$, $D_N$ or $E_N$, respectively.

In practice, the non-abelian subgroups of $\Gamma$ can be difficult to enumerate. Inspired by [84], a simpler method is to write down all the maximal $\mathbb{Z}_N$ abelian subgroups $G_i$ of $\Gamma$ whose non-identity elements belong to the set $\Gamma_{1,f}$, modded out by the equivalence relation $\sim$.

For each maximal $\mathbb{Z}_N$ subgroup $G_i$, denote the group elements by $(\mathbf{1}, g_i, g_i^2, \ldots, g_i^{N-1})$. We only consider the subgroups $G_i$ where each non-identity element $g_i$ belongs to a conjugacy class $\mathfrak{g} \in \Gamma_{1,f}$. Each non-identity element $g_i^k$ ($k = 1, \ldots, N-1$) gives rise to a node in the Dynkin diagram $\mathfrak{g}_i$ of type $A_{N-1}$, and the nodes $g_i^k$ and $g_i^{k+1}$ ($k = 1, \ldots, N-2$) are connected by a single edge. For each $G_i$, there are two cases:

1. Each group element $g^k \in \mathbb{Z}_N \subset G_i$ belongs to a different conjugacy class $\mathfrak{g}_{a(k)}$ of $\Gamma$. In this case, the contribution to the rank of $G_F$ is $N-1$.

2. Each group element $g^k \in \mathbb{Z}_N$, ($k = 0, \ldots, \lfloor \frac{N}{2} \rfloor$) belongs to a different conjugacy class $\mathfrak{g}_{a(k)}$ of $\Gamma$. For the other group elements, $g^k$ and $g^{N-k}$ belongs to the same conjugacy class $\mathfrak{g}_{a(k)}$. In this case, the contribution to the rank of $G_F$ is $\lfloor \frac{N}{2} \rfloor$. This precisely corresponds to the action of $\mathbb{Z}_2$ outer automorphism on ADE Lie algebra, which gives rise to non-simply laced gauge groups in F-theory [91] (which is called Tate-monodromy in that context). In table 4 we list the possible (maximal) flavor symmetry factors from a certain type ADE codimension-two singularity. For an actual 5d SCFT, the flavor symmetry factor may be smaller than the expected factors in table 4, due to the fact that the compact divisor lies on the ramification point, see the discussions in section 2.3.2 and the example in section B.6.

Finally, the different $A_{N-1}$ Dynkin diagrams are glued together by identifying nodes in the same conjugacy class. In the case of 2d McKay correspondence, this is exactly the procedure leading to $ADE$ Dynkin diagrams. Nonetheless, in the cases of $SU(3)$ orbifolds, there are ambiguities in certain cases, due to the monodromy reduction effect. For example, consider the case of type $D_4$ codimension-two singularity, which is reduced to $G_2$. Using the notations in Appendix A, the abelian subgroup $\mathbb{Z}_4$ and the resulting $D_4$ Dynkin diagram can be written as

$$\mathbb{Z}_4 : \{0, 1, 2, 1\} \qquad \underset{1 \quad 2 \quad 1}{\circ\!\!-\!\!\circ\!\!-\!\!\circ} \quad \to \quad \overset{\overset{\displaystyle \circ\, 1}{|}}{\underset{1 \quad 2 \quad 1}{\circ\!\!-\!\!\circ\!\!-\!\!\circ}} \tag{2.39}$$

We labeled the conjugacy class of each element. Nonetheless, the abelian subgroup structure is exactly the same as an $A_3$ singularity, reduced to $C_2 \subset A_3$. Hence these two cases

cannot be distinguished by the structure of abelian subgroups $G_i$ alone, and one needs to study the non-abelian subgroups $\widetilde{G}_i$.

Similarly, there is an ambiguity between type $E_6$ and $D_5$ (with monodromy reduction). The abelian subgroups

$$
\begin{array}{ccccccccc}
\circ\!\!-\!\!\circ\!\!-\!\!\circ\!\!-\!\!\circ\!\!-\!\!\circ & \circ\!\!-\!\!\circ\!\!-\!\!\circ \\
1 \quad 2 \quad 3 \quad 2 \quad 1 & 4 \quad 3 \quad 4
\end{array}
\tag{2.40}
$$

which can be glued into either the Dynkin diagram of $E_6$

$$
\tag{2.41}
$$

or $D_5$

$$
\tag{2.42}
$$

We present examples of this procedure in Appendix A, for the cases of exceptional subgroups $H_i$ and $E^{(i)}$ of $SU(3)$. One can see that the resulting $\mathfrak{g}_{ADE}$ exactly matches the result from geometry.

For the flavor symmetry $G_F$ of the 5d SCFT, there are two additional subtleties:

1. The flavor symmetry factor for each ADE singularity may be different from the expected one in table 4, see the discussions in section 2.3.2 and the case of $\Delta(48)$ in the section 4.2.2 for example.

2. There can be further rank-preserving flavor symmetry enhancements, such as the enhancement of $SU(3)^3 \subset E_6$ in the case of 5d $T_3$ theory [4].

Finally, we present the simplest example, $T_N$ theory [4, 92, 93], where $\Gamma = \mathbb{Z}_N \times \mathbb{Z}_N$. Since $\Gamma$ is abelian in this case, each group element is a conjugacy class by itself. The total number of conjugacy classes is $n^2$, and the number of conjugacy classes in each set is

$$
|\Gamma_1| = \frac{N(N+3)}{2} - 2 \ , \ |\Gamma_2| = \frac{(N-1)(N-2)}{2} \ , \ |\Gamma_{1,f}| = 3N - 3 \ .
\tag{2.43}
$$

We show the case of $N = 5$ and the 24 non-identity conjugacy classes. The two $\mathbb{Z}_N$ generators are denoted as $\omega_1$ and $\omega_2$, and the conjugacy class (element) $\omega_1^p \omega_2^q$ sits on the point $(p, q)$:

$$
\tag{2.44}
$$

Here the blue dots correspond to the conjugacy classes in $\Gamma_2$, which correspond to the compact divisors in $T_5$. The black dots denotes the inverse conjugacy classes of the blue ones, which correspond to the Poincaré dual of the compact divisors. The yellow dots correspond to the elements in the set $\Gamma_{1,f}$.

As one can see, the elements in $\Gamma_{1,f}$ along with the identity element $\mathbf{1}$ exactly form three commuting $\mathbb{Z}_N$'s in $\Gamma$. They give rise to the codimension-two singularity of type $\mathfrak{g}_{ADE} = A_{N-1}^3$, and UV flavor symmetry $G_F = SU(N)^3$. When $N = 3$, the flavor symmetry is further enhanced to $G_F = E_6$, which cannot be detected by this method.

### 2.3.2  5d flavor symmetry from resolution

After the equation for an orbifold singularity $\mathbb{C}^3/\Gamma$ is written down, one can compute the loci and ADE types of codimension-two singularities via the local Jacobian rings for example. As we show in section 3.3 and 3.4, for all of the $\mathbb{C}^3/\Gamma$ covered in this paper, the 5d flavor rank $f$ always matches the number of non-compact exceptional divisors from the resolution of codimension-two singularities.

In general, after resolving a codimension-two singularity of ADE type $\mathfrak{g}_{ADE}$, there will be a number of non-compact exceptional divisors $D_j$ $(j = 1, \ldots, f)$ in $\widetilde{X}$. Along with the compact exceptional divisors $S_i$, there will be $\mathbb{P}^1$ curves

$$C_j = D_j \cdot \left(\sum_i \xi_{i,j} S_i\right) \tag{2.45}$$

with normal bundle $N_{C|\widetilde{X}} = \mathcal{O} \oplus \mathcal{O}(-2)$, which are called flavor curves in [10–12] (the integer coefficient $\xi_{i,j}$ is defined in [13], which depends on the detailed resolution sequence). M2-brane wrapping modes on $C_j$ precisely give rise to the W-boson of the non-Abelian flavor symmetry factor. The symmetrized Cartan matrix [94] is computed from the triple intersection numbers as

$$\mathcal{C}_{jk} = -\frac{2\langle \alpha, \alpha \rangle_{\max} \langle \alpha_j, \alpha_k \rangle}{\langle \alpha_j, \alpha_j \rangle \langle \alpha_k, \alpha_k \rangle} = D_j \cdot D_k \cdot \left(\sum_i \xi_{i,j} S_i\right). \tag{2.46}$$

Naively, the flavor symmetry factor from an ADE singularity type is given by table 4. The non-simply-laced factor appears when

$$f < \mathrm{rk}(\mathfrak{g}_{ADE}). \tag{2.47}$$

Nonetheless, the actual 5d flavor symmetry may be different from the expectation. As an example, consider the geometry in figure 1, where there exists a non-compact exceptional divisor $D$ from the resolution of a codimension-two $A_1$ singularity and a family of divisors $S_t$, $t \in \mathbb{C}$. For a generic $S_t(t \neq 0)$, the intersection curve $D \cdot S_t$ is a smooth flavor curve with

$$S_t \cdot D^2 = -2 \ , \ D \cdot S_t^2 = 0 \,. \tag{2.48}$$

But at $t = 0$, the curve splits into two components $C_1$ and $C_2$ with normal bundle $N_{C_1} = N_{C_2} = \mathcal{O}(-1) \oplus \mathcal{O}(-1)$. In fact, this geometry appears in the splitting $I_{n+1} \to I_n + I_1$ in

| ADE type | Flavor symmetry factor |
|:---:|:---:|
| $A_n$ | $SU(n+1)$ or $Sp(\lfloor \frac{n+1}{2} \rfloor)$ |
| $D_4$ | $SO(8)$ or $SO(7)$ or $G_2$ |
| $D_{n>4}$ | $SO(2n)$ or $SO(2n-1)$ |
| $E_6$ | $E_6$ or $F_4$ |
| $E_7$ | $E_7$ |
| $E_8$ | $E_8$ |

**Table 4**: The list of possible (maximal) flavor symmetry factors from a codimension-two singularity with type ADE, which can be different from the actual 5d flavor symmetry $G_F$. For a given type, the rank of flavor symmetry factor is determined by the number of different non-compact divisors (conjugacy classes in $\Gamma_{1,f}$) associated to a single codimension-two ADE singularity.

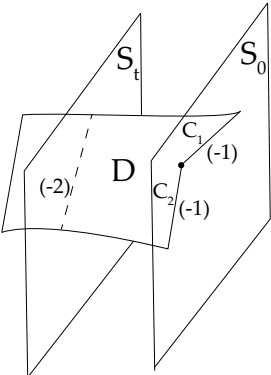

**Figure 1**: The splitting of flavor curve into two $\mathcal{O}(-1) \oplus \mathcal{O}(-1)$ curves.

F-theory, see e. g. [95]. In the 5d setup, suppose that $S_0$ is the compact divisor, and we have

$$S_0 \cdot C_1 = S_0 \cdot C_2 = -1, \tag{2.49}$$

the divisor $D$ would generate $U(1)$ flavor symmetry instead of $SU(2)$.

We will apply this discussion to the case of $\Delta(48)$ in section 4.2.2.

As another example, consider the geometry in figure 2, which is the resolution of a codimension-two $D_4$ singularity

$$x^3 + y^3 t + z^2 = 0 \tag{2.50}$$

in $\mathbb{C}^4[x, y, z, t]$. The blow up sequence is

$$
\begin{aligned}
&(x, y, z; \delta_1) \\
&(z, \delta_1; \delta_2) \\
&(\delta_1, \delta_2; \delta_3) \,.
\end{aligned}
\tag{2.51}
$$

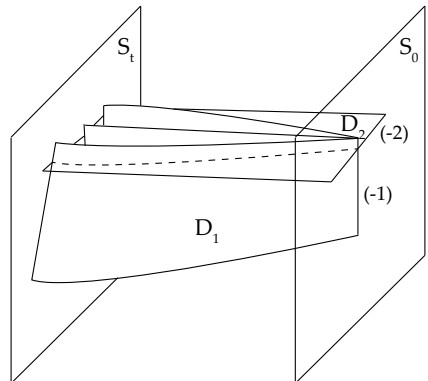

**Figure 2**: The ramification point of codimension-two $D_4$ singularity gives rise to an $\mathcal{O}(-1) \oplus \mathcal{O}(-1)$ curve on $S_0$.

The resolved equation is

$$(x^3 + y^3 t)\delta_1 + z^2 \delta_2 = 0 \tag{2.52}$$

Note that there are only two irreducible non-compact divisors $D_1 : \quad \delta_2 = 0$ and $D_2 : \quad \delta_3 = 0$ in the resolution. The fiber over $t \neq 0$ is four $\mathbb{P}^1$ curves in the shape of a $D_4$ Dynkin diagram. Over the ramification point $t = 0$, the fiber $\delta_2 = 0$ degenerates to a single $\mathbb{P}^1$

$$\delta_2 = t = x = 0, \tag{2.53}$$

which is rigid and has normal bundle $\mathcal{O}(-1) \oplus \mathcal{O}(-1)$. If $t = 0$ is the location of compact divisor $S_0$, then the codimension-two $D_4$ singularity only gives rise to flavor symmetry factor $SU(2) \times U(1)$ instead of $G_2$.

This is exactly the case of $\Gamma = E^{(2)}$ in section B.6.

## 2.4 McKay quiver and 1-form symmetry of 5d SCFT

In this section we present the method to compute the (electric) 1-form symmetry $\Gamma^{(1)}$ of the 5d SCFT, using group theoretic data of $\Gamma$. It is based on a version of 3d McKay correspondence [96], which is proved in [97] for any $\Gamma \subset SL(3, \mathbb{C})$:

*the $\Gamma$-Hilbert Scheme (Hilbert quotient) $\Gamma$-Hilb $\mathbb{C}^3$ is an irreducible crepant resolution of $\mathbb{C}^3/\Gamma$.*

We first review the definition of McKay quiver of $\Gamma$ [55, 62]. For each irrep $\rho_i$ ($i = 0, \ldots, \chi(\widetilde{X}) - 1$) of $\Gamma$, we draw a node labeled by $i$. Then we consider the tensor product with the three-dimensional natural representation $\pi$ (which can be reducible)

$$\rho_i \otimes \pi = \sum_j a_{ji} \rho_j. \tag{2.54}$$

For each non-zero $a_{ji}$ term, we draw $a_{ji}$ arrows from the $i$-th node to the $j$-th node. The resulting quiver diagram is the McKay quiver of $\Gamma$.

We define the antisymmetric adjacency matrix $A(\Gamma) = \{A_{ij}\}$ by

$$A_{ij} = a_{ji} - a_{ij}. \tag{2.55}$$

Now we apply corollary 5.3 of [96], and we have the following statement:

$A(\Gamma)$ *has the same Smith normal form as* $M(\Gamma)$, *where the latter is the intersection product matrix between* $\chi(\widetilde{X}) = 1 + b_2(\widetilde{X}) + b_4(\widetilde{X})$ *topological cycles of the resolved CY3* $\widetilde{X}$.

$M(\Gamma)$ has the block form of

$$M(\Gamma) = \begin{pmatrix} 0 & 0 & 0 \\ \hline 0 & 0 & \mathcal{M}_{b_2 \times b_4} \\ \hline 0 & (\mathcal{M}^T)_{b_4 \times b_2} & 0 \end{pmatrix}, \tag{2.56}$$

where $\mathcal{M}_{b_2 \times b_4}$ is the intersection matrix between compact 2-cycles and 4-cycles in $\widetilde{X}$.

Smith normal form of $M(\Gamma)$ (which is the same as that of $A(\Gamma)$) has the form of

$$\begin{pmatrix} \alpha_1 & 0 & 0 & 0 & \dots & 0 \\ 0 & \alpha_1 & 0 & 0 & \dots & 0 \\ 0 & 0 & \alpha_2 & 0 & \dots & 0 \\ 0 & 0 & 0 & \alpha_2 & \dots & 0 \\ \vdots & \vdots & \vdots & \vdots & \ddots & \vdots \\ 0 & 0 & 0 & 0 & \dots & 0 \end{pmatrix}$$

The Smith normal form of $\mathcal{M}_{b_2 \times b_4}$ then takes the form of

$$D = \begin{pmatrix} \alpha_1 & 0 & \dots & 0 \\ 0 & \alpha_2 & \dots & 0 \\ \vdots & \vdots & \ddots & \vdots \\ 0 & 0 & \dots & \alpha_{b_4} \\ 0 & 0 & \dots & 0 \\ \vdots & \vdots & \ddots & \vdots \\ 0 & 0 & \dots & 0 \end{pmatrix} \tag{2.57}$$

The 1-form symmetry $\Gamma^{(1)}$ of $T_X^{5d}$ is computed as [80]

$$\Gamma^{(1)} = \oplus_{i=1}^{b_4} \mathbb{Z}/(\alpha_i \mathbb{Z}). \tag{2.58}$$

As an example, consider the case of $\Gamma = \mathbb{Z}_5$, whose group generator is $a = \frac{1}{5}(2, 2, 1)$. The 1d representations of $\mathbb{Z}_5$ are denoted by $\rho_i$ $(i = 0, \dots, 4)$, where $\rho_i(a^j) = a^{ij}$.

The natural representation of $\mathbb{Z}_5$ in this case is

$$\pi = \rho_2 \oplus \rho_2 \oplus \rho_1. \tag{2.59}$$

The McKay quiver is

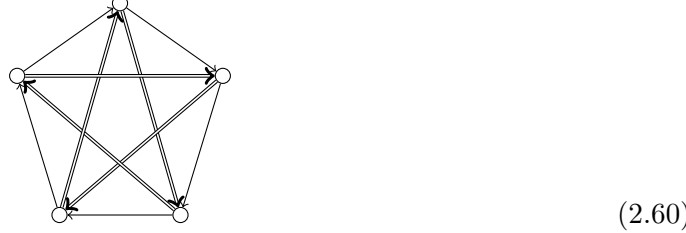

$$\hspace{10cm}(2.60)$$

The antisymmetric adjacency matrix of the McKay quiver can be computed as

$$A(\mathbb{Z}_5) = \begin{pmatrix} 0 & 1 & 2 & -2 & -1 \\ -1 & 0 & 1 & 2 & -2 \\ -2 & -1 & 0 & 1 & 2 \\ 2 & -2 & -1 & 0 & 1 \\ 1 & 2 & -2 & -1 & 0 \end{pmatrix} \hspace{3cm}(2.61)$$

The Smith normal form of $A(\mathbb{Z}_5)$ is

$$\begin{pmatrix} 5 & 0 & 0 & 0 & 0 \\ 0 & 5 & 0 & 0 & 0 \\ 0 & 0 & 1 & 0 & 0 \\ 0 & 0 & 0 & 1 & 0 \\ 0 & 0 & 0 & 0 & 0 \end{pmatrix}, \hspace{3cm}(2.62)$$

which is exactly the same $M(\Gamma)$. The intersection matrix $\mathcal{M}_{2\times 2}$ then has the Smith normal form

$$\begin{pmatrix} 5 & 0 \\ 0 & 1 \end{pmatrix}, \hspace{3cm}(2.63)$$

which gives rise to the correct 1-form symmetry $\Gamma^{(1)} = \mathbb{Z}_5$ of $T_X^{5d}$. We will check this result using toric geometry in section 3.1.1.

More generally, when $\Gamma = \mathbb{Z}_{2k+1}$, the Smith normal form of the antisymmetric adjacency matrix $A(\mathbb{Z}_{2k+1})$ is of the form $\text{diag}(2k+1, 2k+1, 1, \cdots, 1, 0)$ whose dimension is $2k+1$ and the number of diagonal 1's is $2k-2$, while when $\Gamma = \mathbb{Z}_{2k}$, the Smith normal form of the antisymmetric adjacency matrix $A(\mathbb{Z}_{2k})$ is of the form $\text{diag}(k, k, 1, \cdots, 1, 0, 0)$ whose dimension is $2k$ and the number of diagonal 1's is $2k-4$. The above results match the fact that the 1-form symmetry of $\mathbb{C}^3/\mathbb{Z}_{2k+1}$ theory is $\mathbb{Z}_{2k+1}$ and the 1-form symmetry of $\mathbb{C}^3/\mathbb{Z}_{2k}$ theory is $\mathbb{Z}_k$. For the $T_n$ series, the Smith normal form of the antisymmetric adjacency matrix is $\text{diag}(1, \cdots, 1, 0, \cdots, 0)$ whose dimension is $n^2$ and the number of diagonal 1's is $(n-1)(n-2)$ therefore the 1-form symmetry of $T_n$ theory is trivial. For the $\Delta(3n^2)$ and $\Delta(6n^2)$ series a similar result holds such that the number of diagonal 1's is $2r$ and the other diagonals are zeroes therefore the 1-form symmetry of these theories are also trivial.

It is also rather trivial to compute the 1-form symmetry of the exceptional cases $H_{36}$, $H_{60}$, $H_{72}$, $H_{168}$, $H_{216}$, $H_{360}$ whose McKay quivers can be found in [62, 98, 99]. The Smith normal form of the antisymmetric adjacency matrices of these exceptional cases are all of

the form $\mathrm{diag}(1, \cdots, 1, 0, \cdots, 0)$ whose dimension is $\chi(\tilde{X})$ and the number of diagonal 1's is $2r$. Therefore the 1-form symmetry of this set of 5D theories is trivial. Similar results can be found for other non-abelian $\Gamma$s in table 2.

We also present a non-abelian $\Gamma$ with a non-trivial 1-form symmetry, which is the following class (B) example with the generators ($\omega = e^{\pi i/3}$):

$$M_1 = \begin{pmatrix} i & 0 & 0 \\ 0 & -i & 0 \\ 0 & 0 & 1 \end{pmatrix} \ , \ M_2 = \begin{pmatrix} 0 & i & 0 \\ i & 0 & 0 \\ 0 & 0 & 1 \end{pmatrix} \ , \ M_3 = \begin{pmatrix} \omega & 0 & 0 \\ 0 & \omega & 0 \\ 0 & 0 & \omega^{-2} \end{pmatrix} . \tag{2.64}$$

The 5d SCFT has $r = 5$, $f = 4$ and $\Gamma^{(1)} = \mathbb{Z}_3$, and the singular equation $\mathbb{C}^3/\Gamma$ is neither toric, a hypersurface nor a complete intersection of two equations. We leave a complete classification of these cases in a future work.

## 2.5 Relation between 6d and 5d SCFTs

Here we consider the cases where (after the redefinition of variables) the canonical threefold singularity $X$ can be rewritten in the form of

$$y^2 = x^3 + fx + g \,, \tag{2.65}$$

where $f$ and $g$ are holomorphic functions. In this case, we can embed the singularity into a Weierstrass model

$$y^2 = x^3 + fxZ^4 + gZ^6 \,, \tag{2.66}$$

where $[x : y : Z]$ are projective coordinates of $\mathbb{P}^{2,3,1}$. We can thus define a 6d (1,0) SCFT $T_X^{6d}$ as F-theory on this elliptic singularity. Now we discuss the relation between $T_X^{6d}$ and $T_X^{5d}$, starting with the KK reduction of $T_X^{6d}$ on $S^1$, and goes on the Coulomb branch. Geometrically, the 5d KK theory is M-theory on the resolved elliptic CY3 $\widetilde{X}_{\mathrm{ell}}$, with compact 4-cycles $\bigcup S_i$. The zero section $Z = 0$ is a non-compact divisor of $\widetilde{X}_{\mathrm{ell}}$. To get the 5d theory $T_X^{5d}$, the elliptic fibration structure needs to be broken, and the zero section $Z = 0$ should not intersect the compact 4-cycles any more. In order to achieve this, there are two different geometric transitions from $\widetilde{X}_{\mathrm{ell}}$ to $\widetilde{X}$, depending on whether $T_X^{6d}$ is very-Higgsable:

1. (2.66) is a hypersurface equation and $(f, g)$ are holomorphic functions on $\mathbb{C}^2$. In this case, $T_X^{6d}$ is a very-Higgsable theory as the base $B$ in the tensor branch description can be blown down to $\mathbb{C}^2$.

   Now consider the intersection curves between the non-compact zero-section $Z = 0$ and compact 4-cycles $S_i$. Because $T_X^{6d}$ is very-Higgsable, these curves can be linearly combined into a curve $C_Z = \sum_i \xi_{i,Z} S_i \cdot Z$ with normal bundle $\mathcal{O}(-1) \oplus \mathcal{O}(-1)$. In fact the curve $C_Z$ corresponds to a $(n, g) = (-1, 0)$ node in the language of combined fiber diagram (CFD) [11, 13], and the coefficient $\xi_{i,Z}$ are the multiplicity factors. Then in the transition from the 6d to the 5d description, one decouples a hypermultiplet by flopping $C_Z$ out of $\bigcup S_i$. The resulting CY3 is no longer elliptic as the zero section $Z$

no longer intersects the new set of compact 4-cycles $\bigcup S_i'$. The 5d SCFT $T_X^{5d}$ defined as M-theory on $X$ is obtained at the origin of the new Coulomb branch, where $S_i'$s are all shrunk to a point.

For example, in the case of

$$X: \ y^2 = x^3 + z^6 + w^6 \,, \tag{2.67}$$

the 6d interpretation is the rank-1 E-string. Its KK reduction has an IR description of $SU(2) + 8\boldsymbol{F}$. After decoupling a fundamental hyper, we get $SU(2) + 7\boldsymbol{F}$ whose UV completion is the rank-1 Seiberg $E_8$ theory. In the resolved CY3 picture, the decoupling precisely corresponds to the blow down of a dP$_9$ into a dP$_8$. This is also consistent with the 5d interpretation of (2.67) [6].

2. (2.66) is a complete intersection equation, which can be interpreted as a Weierstrass model over a singular base $B$. In this case, $T_X^{6d}$ is a non-very-Higgsable theory. From the full tensor branch of $T_X^{6d}$, we perform a number of blow downs until the compact curves $C_i$ $(i = 1, \ldots, q)$ on the resulting base all have self-intersection number $n \leq -2$.

In this case, starting from the resolved elliptic CY3 $\widetilde{X}_{\mathrm{ell}}$, we cannot flop out the curve $C_Z$ because its normal bundle is not $\mathcal{O}(-1) \oplus \mathcal{O}(-1)$. Alternatively, we decompactify all the vertical divisors over $C_i$ $(i = 1, \ldots, q)$ (with the topology of Hirzebruch surfaces). Equivalently we decouple $q$ $U(1)$ vector multiplets on the Coulomb branch. As a consequence, the zero-section $Z = 0$ no longer intersects $\widetilde{X}$, and we can get the 5d SCFT $T_X^{5d}$ as the origin of the new Coulomb branch.

Examples can be found after comparing the 6d description of $\Gamma = \Delta(54)$ in section 4.3.1 and the 5d description in section 4.3.2, or comparing the 6d description of $\Gamma = H_{36}$ in section B.3 and the 5d description in section B.3.2.

These procedures precisely match the systematics of decoupling in [13].

# 3 The finite subgroups of $SU(3)$

In this section we present the group theoretic data and geometry of $\mathbb{C}^3/\Gamma$ orbifolds. In section 3.1 we study the cases where $\Gamma$ is abelian. In these cases we are able obtain maximal understanding of the corresponding 5d theories via both the group theoretic data of $\Gamma$ and the toric geometry of the $\mathbb{C}^3/\Gamma$ orbifold. In Section 3.2 we discuss the cases of non-abelian $\Gamma$ which can be thought as finite subgroups of $U(2) \subset SU(3)$. In Section 3.3 and 3.4 we study the non-abelian finite subgroups of $SU(3)$ that are not subgroups of $U(2)$. In particular, in Section 3.3 we will focus on the infinite series (cf. $A$ and $D$ series of finite subgroups of $SU(2)$) while in Section 3.4 we will focus on the exceptional cases (cf. $E$ type finite subgroups of $SU(2)$).

## 3.1 Abelian subgroups

In this section, we discuss the simple cases of $\Gamma$ being abelian. In these cases, every conjugacy class of $\Gamma$ only contains a single element, and the total number of conjugacy

classes equals to the order of $\Gamma$, we have

$$|\Gamma| = 1 + |\Gamma_1| + |\Gamma_2|\,. \tag{3.1}$$

The generators of $\Gamma$ are chosen in a way such that $\mathbb{C}^3/\Gamma$ contains a codimension-three singularity.

For example, taking $\Gamma = \mathbb{Z}_3$, the generator can be chosen as either $\frac{1}{3}(1,1,1)$ or $\frac{1}{3}(0,1,2)$. In the first case, the junior conjugacy classes and age-two conjugacy classes are

$$\Gamma_1 = \{\frac{1}{3}(1,1,1)\}\,, \ \ \Gamma_2 = \{\frac{1}{3}(2,2,2)\}\,. \tag{3.2}$$

Hence we have $r = 1$, $f = 0$, and the 5d SCFT exactly corresponds to the rank-1 Seiberg $E_0$ theory. The single compact exceptional divisor in the resolution $\widetilde{X}$ is exactly $\mathbb{P}^2$.

In the second case, the conjugacy classes are

$$\Gamma_1 = \{\frac{1}{3}(0,1,2), \frac{1}{3}(0,2,1)\}\,, \ \ \Gamma_2 = \{\}\,. \tag{3.3}$$

It corresponds to the singularity $\mathbb{C} \times \mathbb{C}^2/\mathbb{Z}_3$, which gives rise to a 7d $SU(3)$ SYM theory rather than a 5d theory. In the following discussions, such 7d cases are not included.

### 3.1.1 $\mathbb{Z}_n$

The generator of $\mathbb{Z}_n$ is chosen as $\frac{1}{n}(p,q,r)$, $p+q+r=n$. The threefold singularity $\mathbb{C}^3/\mathbb{Z}_n$ is toric. Denote the three vertices of weighted projective space $\mathbb{P}^{p,q,r}$ as $v_1, v_2, v_3 \in \mathbb{Z}^2$, then the rays in the fan of the toric threefold singularity are $(v_i, 1)$ $(i = 1,2,3)$. The resolution of $\mathbb{C}^3/\mathbb{Z}_n$ is thus given by a fine, regular star triangulation (FRST) of the 3d polytope.

Since $\mathbb{C}^3/\mathbb{Z}_n$ is toric, the (magnetic) 1-form symmetry $\Gamma^{(1)}$ of the 5d SCFT can also be computed with the procedure in [80]. Define a $3 \times (3+f)$ matrix $A_E$ to be the list of rays on the boundary of the 3d polytope, including $(v_i, 1)$ $(i = 1,2,3)$, then we compute the Smith decomposition

$$A_E = SDT\,, \tag{3.4}$$

where $D$ is the Smith normal form of $A_E$

$$D = \begin{pmatrix} \alpha_1 & 0 & 0 \\ 0 & \alpha_2 & 0 \\ 0 & 0 & \alpha_3 \\ 0 & 0 & 0 \\ & \vdots & \\ 0 & 0 & 0 \end{pmatrix} \tag{3.5}$$

The 1-form symmetry group is computed as

$$\Gamma^{(1)} = \oplus_{i=1}^{3} \mathbb{Z}/(\alpha_i\mathbb{Z})\,. \tag{3.6}$$

In particular, for $\mathbb{Z}_{2k+1}$ with the generator $\frac{1}{2k+1}(1,1,2k-1)$, the vertices can be chosen as

$$v_1 = (-1,-1)\,, \ \ v_2 = (0,1)\,, \ \ v_3 = (k,0)\,. \tag{3.7}$$

The resolution is shown as

$$(3.8)$$

The black dots correspond to compact divisors. From the counting of conjugacy classes, we also get

$$|\Gamma_1| = |\Gamma_2| = k \,, \tag{3.9}$$

which implies

$$r = k \,, \ f = 0 \,. \tag{3.10}$$

The 1-form symmetry is generally $\Gamma^{(1)} = \mathbb{Z}_{2k+1}$.

For example, for $k = 2$, the only orbifold singularity $\mathbb{C}^3/\mathbb{Z}_5$ exactly corresponds to a rank-2 SCFT with $f = 0$, which is labeled as $\mathbb{P}^2 \cup \mathbb{F}_3$ in [16]. It has 1-form symmetry $\Gamma^{(1)} = \mathbb{Z}_5$ [80].

For arbitrary $k \geq 2$, the compact complex surfaces are $\mathbb{P}^2 \cup \mathbb{F}_3 \cup \mathbb{F}_5 \cup \cdots \cup \mathbb{F}_{2k-1}$.

Similarly, for $\mathbb{Z}_{2k}$ with the generator $\frac{1}{2k}(1, 1, 2k-2)$, the vertices can be chosen as

$$v_1 = (-1, -1) \,, \ v_2 = (-1, 1) \,, \ v_3 = (k-1, 0) \,. \tag{3.11}$$

The resolution is

$$(3.12)$$

Here the non-compact divisor that gives rise to flavor symmetry is colored yellow. From the counting of conjugacy classes, we also get

$$|\Gamma_1| = k \,, \ |\Gamma_2| = k - 1 \,, \tag{3.13}$$

which is consistent with

$$r = k - 1 \,, \ f = 1 \,. \tag{3.14}$$

The 1-form symmetry is $\Gamma^{(1)} = \mathbb{Z}_k$.

The flavor symmetry of the corresponding 5d SCFT is always $G_F = SU(2)$ for this sequence of theories. In fact, they have exactly the IR gauge theory description $SU(k)_k$ ($SU(2)_0$ when $k = 2$).

Note that for certain $\mathbb{Z}_n$ generators, such as $\frac{1}{2k}(1, k-1, k)$, the singularity is isomorphic to $\mathbb{C}^3/\mathbb{Z}_2 \times \mathbb{Z}_k$.

Apart from these two infinite series, we also list a number of $\mathbb{C}^3/\mathbb{Z}_n$ orbifolds with their resolutions for small $n$.

1. $n = 7$, generator $\frac{1}{7}(1, 2, 4)$, $r = 3$, $f = 0$, $\Gamma^{(1)} = \mathbb{Z}_7$ (this theory is known as $B_4$ in [80])

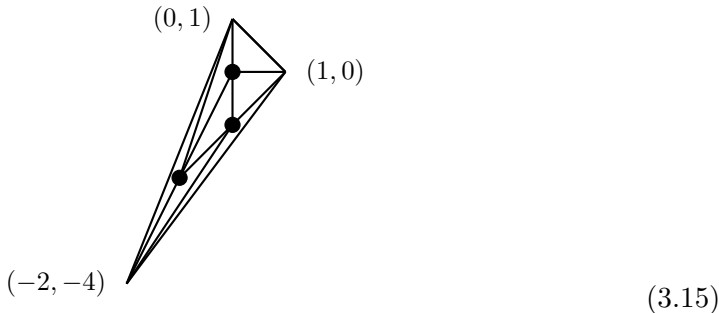

$$(3.15)$$

2. $n = 8$, generator $\frac{1}{8}(1, 2, 5)$, $r = 3$, $f = 1$, $G_F = SU(2)$, $\Gamma^{(1)} = \mathbb{Z}_4$

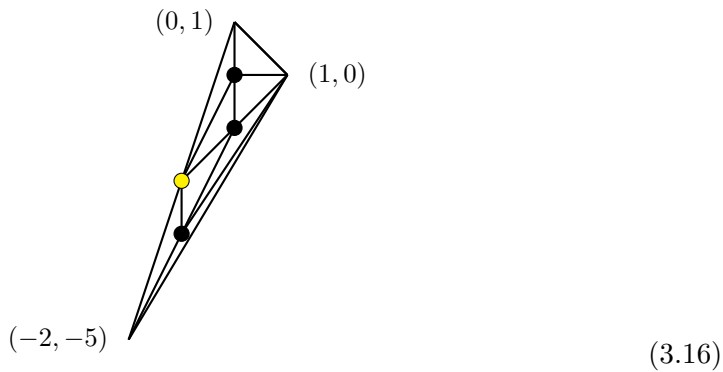

$$(3.16)$$

3. $n = 9$, generator $\frac{1}{9}(1, 2, 6)$, $r = 3$, $f = 2$, $G_F = SU(3)$, $\Gamma^{(1)} = \mathbb{Z}_3$

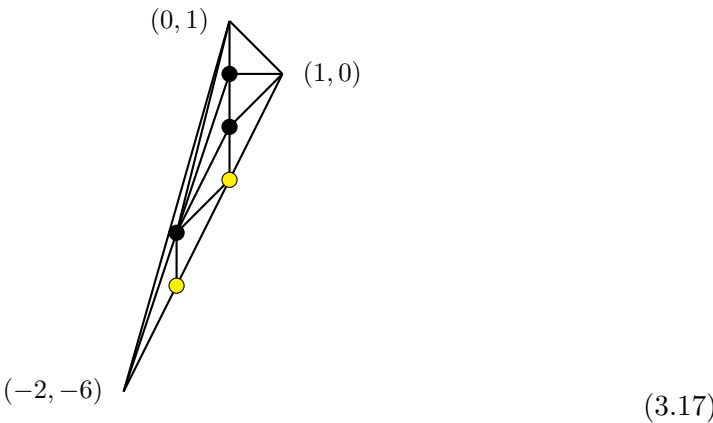

$$(3.17)$$

We list the physical data of $\mathbb{C}^3/\mathbb{Z}_n$ theories with small $n$ in table 5.

### 3.1.2 $\mathbb{Z}_m \times \mathbb{Z}_n$

In this case, the generators of $\mathbb{Z}_m$ and $\mathbb{Z}_n$ are chosen as $\frac{1}{m}(1, m-1, 0)$ and $\frac{1}{n}(0, 1, n-1)$. The orbifold $\mathbb{C}^3/\mathbb{Z}_m \times \mathbb{Z}_n$ is a toric threefold singularity, with rays $(0, 0, 1)$, $(m, 0, 1)$, $(0, n, 1)$.

| $n$ | Generator | $r$ | $f$ | $G_F$ | $\Gamma^{(1)}$ | IR gauge theory |
|---|---|---|---|---|---|---|
| 3 | $\frac{1}{3}(1,1,1)$ | 1 | 0 | $-$ | $\mathbb{Z}_3$ | $-$ |
| 4 | $\frac{1}{4}(1,1,2)$ | 1 | 1 | $SU(2)$ | $\mathbb{Z}_2$ | $SU(2)_0$ |
| 5 | $\frac{1}{5}(1,1,3)$ | 2 | 0 | $-$ | $\mathbb{Z}_5$ | $-$ |
| 6 | $\frac{1}{6}(1,1,4)$ | 2 | 1 | $SU(2)$ | $\mathbb{Z}_3$ | $SU(3)_3$ |
| 6 | $\frac{1}{6}(1,2,3)$ | 1 | 3 | $SU(3) \times SU(2)$ | $-$ | $SU(2) + 2\boldsymbol{F}$ |
| 7 | $\frac{1}{7}(1,1,5)$ | 3 | 0 | $-$ | $\mathbb{Z}_7$ | $-$ |
| 7 | $\frac{1}{7}(1,2,4)$ | 3 | 0 | $-$ | $\mathbb{Z}_7$ | $-$ |
| 8 | $\frac{1}{8}(1,1,6)$ | 3 | 1 | $SU(2)$ | $\mathbb{Z}_4$ | $SU(4)_4$ |
| 8 | $\frac{1}{8}(1,2,5)$ | 3 | 1 | $SU(2)$ | $\mathbb{Z}_4$ | $-$ |
| 8 | $\frac{1}{8}(1,3,4)$ | 2 | 3 | $SU(4)$ | $\mathbb{Z}_2$ | $SU(2)_0 - SU(2)_0$ |
| 9 | $\frac{1}{9}(1,1,7)$ | 4 | 0 | $-$ | $\mathbb{Z}_9$ | $-$ |
| 9 | $\frac{1}{9}(1,2,6)$ | 3 | 2 | $SU(3)$ | $\mathbb{Z}_3$ | $-$ |
| 10 | $\frac{1}{10}(1,1,8)$ | 4 | 1 | $SU(2)$ | $\mathbb{Z}_5$ | $SU(5)_5$ |
| 10 | $\frac{1}{10}(1,2,7)$ | 4 | 1 | $SU(2)$ | $\mathbb{Z}_5$ | $-$ |
| 10 | $\frac{1}{10}(1,3,6)$ | 4 | 1 | $SU(2)$ | $\mathbb{Z}_5$ | $-$ |
| 10 | $\frac{1}{10}(1,4,5)$ | 2 | 5 | $SU(5) \times SU(2)$ | - | $SU(2)_0 - SU(2) - 2\boldsymbol{F}$ |
| 11 | $\frac{1}{11}(1,1,9)$ | 5 | 0 | $-$ | $\mathbb{Z}_{11}$ | $-$ |
| 11 | $\frac{1}{11}(1,2,8)$ | 5 | 0 | $-$ | $\mathbb{Z}_{11}$ | $-$ |
| 12 | $\frac{1}{12}(1,1,10)$ | 5 | 1 | $SU(2)$ | $\mathbb{Z}_6$ | $SU(6)_6$ |
| 12 | $\frac{1}{12}(1,2,9)$ | 4 | 3 | $SU(3) \times SU(2)$ | $\mathbb{Z}_2$ | $SU(4)_4 - SU(2)_0$ |
| 12 | $\frac{1}{12}(1,3,8)$ | 3 | 5 | $SU(4) \times SU(3)$ | $-$ | $SU(3)_2 - SU(2)_0 - 2\boldsymbol{F}$ |
| 12 | $\frac{1}{12}(1,4,7)$ | 4 | 3 | $SU(4)$ | $\mathbb{Z}_3$ | $-$ |
| 12 | $\frac{1}{12}(1,5,6)$ | 3 | 5 | $SU(6)$ | $\mathbb{Z}_2$ | $SU(2)_0 - SU(2)_0 - SU(2)_0$ |
| 13 | $\frac{1}{13}(1,1,11)$ | 6 | 0 | $-$ | $\mathbb{Z}_{13}$ | $-$ |
| 13 | $\frac{1}{13}(1,2,10)$ | 6 | 0 | $-$ | $\mathbb{Z}_{13}$ | $-$ |
| 13 | $\frac{1}{13}(1,3,9)$ | 6 | 0 | $-$ | $\mathbb{Z}_{13}$ | $-$ |

**Table 5**: List of $\mathbb{C}^3/\mathbb{Z}_n$ with their generators, $r$, $f$, UV flavor symmetry $G_F$, 1-form symmetry $\Gamma^{(1)}$ and IR gauge theory descriptions of the corresponding 5d SCFT.

For example, if $m = n$, the singularity $\mathbb{C}^3/\mathbb{Z}_n \times \mathbb{Z}_n$ exactly gives the 5d $T_n$ theory. If $m = 3$, $n = 2$, the singularity $\mathbb{C}^3/\mathbb{Z}_3 \times \mathbb{Z}_2$ corresponds to rank-1 $E_3$ theory, which has the following toric diagram:

$$(3.18)$$

From the picture, we can easily read off $G_F = SU(3) \times SU(2)$.

We list the basic data of $\mathbb{C}^3/\mathbb{Z}_m \times \mathbb{Z}_n$ theories with small $m, n$ in table 6. The 1-form symmetry is always trivial for these cases.

| $m$ | $n$ | $r$ | $f$ | $G_F$ | IR gauge theory |
|---|---|---|---|---|---|
| 3 | 2 | 1 | 3 | $SU(3) \times SU(2)$ | $SU(2) + 2\boldsymbol{F}$ |
| 4 | 2 | 1 | 5 | $SO(10) \supset SU(4) \times SU(2)^2$ | $SU(2) + 4\boldsymbol{F}$ |
| 5 | 2 | 2 | 5 | $SU(5) \times SU(2)$ | $SU(2)_0 - SU(2) - 2\boldsymbol{F}$ |
| 6 | 2 | 2 | 7 | $SU(6) \times SU(2)^2$ | $SU(3)_0 + 6\boldsymbol{F}/2\boldsymbol{F} - SU(2) - SU(2) - 2\boldsymbol{F}$ |
| 7 | 2 | 3 | 7 | $SU(7) \times SU(2)$ | $SU(2)_0 - SU(2)_0 - SU(2) - 2\boldsymbol{F}$ |
| 8 | 2 | 3 | 9 | $SU(8) \times SU(2)^2$ | $SU(4)_0 + 8\boldsymbol{F}/2\boldsymbol{F} - SU(2) - SU(2)_0 - SU(2) - 2\boldsymbol{F}$ |
| 3 | 3 | 1 | 6 | $E_6 \supset SU(3)^3$ | $SU(2) + 5\boldsymbol{F}$ |
| 4 | 3 | 3 | 5 | $SU(4) \times SU(3)$ | $SU(3)_2 - SU(2) - 2\boldsymbol{F}$ |
| 5 | 3 | 4 | 6 | $SU(6) \times SU(4)$ | $-$ |
| 6 | 3 | 4 | 9 | $SU(6) \times SU(3)^2$ | $SU(2)_0 - SU(4)_0 - 6\boldsymbol{F}$ |
| 7 | 3 | 6 | 8 | $SU(7) \times SU(3)$ | $-$ |
| 4 | 4 | 3 | 9 | $SU(4)^3$ | $4\boldsymbol{F} - SU(3)_0 - SU(2) - 2\boldsymbol{F}$ |
| 5 | 4 | 6 | 7 | $SU(5) \times SU(4)$ | $SU(4)_{\frac{5}{2}} - SU(3)_0 - SU(2) - 2\boldsymbol{F}$ |
| 6 | 4 | 7 | 9 | $SU(6) \times SU(4) \times SU(2)$ | $SU(2)_0 - SU(4)_{\frac{3}{2}} - SU(3)_0 - SU(2) - 2\boldsymbol{F}$ |
| 7 | 4 | 9 | 9 | $SU(7) \times SU(4)$ | $-$ |
| 5 | 5 | 6 | 12 | $SU(5)^3$ | $5\boldsymbol{F} - SU(4)_0 - SU(3)_0 - SU(2) - 2\boldsymbol{F}$ |
| 6 | 5 | 10 | 9 | $SU(6) \times SU(5)$ | $SU(5)_3 - SU(4)_0 - SU(3)_0 - SU(2) - 2\boldsymbol{F}$ |
| 7 | 5 | 12 | 10 | $SU(7) \times SU(5)$ | $-$ |
| 6 | 6 | 10 | 15 | $SU(6)^3$ | $6\boldsymbol{F} - SU(5)_0 - SU(4)_0 - SU(3)_0 - SU(2) - 2\boldsymbol{F}$ |
| 7 | 6 | 15 | 11 | $SU(7) \times SU(6)$ | $SU(6)_{\frac{7}{2}} - SU(5)_0 - SU(4)_0 - SU(3)_0 - SU(2) - 2\boldsymbol{F}$ |
| 7 | 7 | 15 | 18 | $SU(7)^3$ | $7\boldsymbol{F} - SU(6)_0 - SU(5)_0 - SU(4)_0 - SU(3)_0 - SU(2) - 2\boldsymbol{F}$ |

**Table 6**: List of $\mathbb{C}^3/\mathbb{Z}_m \times \mathbb{Z}_n$ with their $r$, $f$, UV flavor symmetry $G_F$ and IR gauge theory descriptions of the corresponding 5d SCFT.

### 3.2 Non-abelian subgroups of $U(2)$

In this section, we discuss the cases of $\Gamma$ as a non-abelian subgroup of $U(2)$, namely the class (B) in table 1. In particular, we will focus on the cases where $X = \mathbb{C}^3/\Gamma$ can be written as a hypersurface in $\mathbb{C}^4$ or a complete intersection of two equations in $\mathbb{C}^5$, which were classified in [74].

#### 3.2.1 Infinite sequences $G_m$, $G_{p,q}$ and $G_m'$

1. $G_m$

   This is the case (2) of [74]. The generators are:

$$M_1 = \begin{pmatrix} \omega & 0 & 0 \\ 0 & \omega^{-1} & 0 \\ 0 & 0 & 1 \end{pmatrix}, \ M_2 = \begin{pmatrix} -1 & 0 & 0 \\ 0 & 1 & 0 \\ 0 & 0 & -1 \end{pmatrix}, \ M_3 = \begin{pmatrix} 0 & 1 & 0 \\ 1 & 0 & 0 \\ 0 & 0 & -1 \end{pmatrix} \tag{3.19}$$

   where $\omega = e^{\pi i/m}$. From group theoretic data, when $2 \mid m$, $\chi = 2m+6$, $|\Gamma_1| = \frac{3}{2}m+5$, $|\Gamma_2| = \frac{1}{2}m$ while when $2 \nmid m$, $\chi = 2m+3$, $|\Gamma_1| = \frac{3}{2}m+\frac{5}{2}$, $|\Gamma_2| = \frac{1}{2}m-\frac{1}{2}$. In this case $\mathbb{C}^3/\Gamma$ is a hypersurface given by the equation

$$4wx^{m+1} - wxy^2 + z^2 = 0. \tag{3.20}$$

2. $G_{p,q}$

   This is the case (18) of [74]. The generators are

   $$M_1 = \begin{pmatrix} \omega & 0 & 0 \\ 0 & \omega^{-1} & 0 \\ 0 & 0 & 1 \end{pmatrix}, \quad M_2 = \begin{pmatrix} \omega^p & 0 & 0 \\ 0 & 1 & 0 \\ 0 & 0 & \omega^{-p} \end{pmatrix}, \quad M_3 = \begin{pmatrix} 0 & 1 & 0 \\ 1 & 0 & 0 \\ 0 & 0 & -1 \end{pmatrix} \quad (3.21)$$

   where $\omega = e^{\pi i/pq}$, $p \geq 1$ and $q \geq 2$. In this case $\mathbb{C}^3/\Gamma$ is a complete intersection given by the equations

   $$\begin{cases} 4ux^p - uy^2 + z^2 = 0, \\ u^q - vx = 0. \end{cases} \quad (3.22)$$

3. $G'_m$

   This corresponds to the case (19) and (20) of [74], the order of group is $|G'_m| = 8m$. When $m$ is even, the group generators are

   $$M_1 = \begin{pmatrix} 0 & -i & 0 \\ -i & 0 & 0 \\ 0 & 0 & 1 \end{pmatrix}, \quad M_2 = \begin{pmatrix} iw & 0 & 0 \\ 0 & iw^{-1} & 0 \\ 0 & 0 & -1 \end{pmatrix} \quad (3.23)$$

   where $w = e^{2\pi i/4m}$. For this group we have $\chi = 2m+3$, $|\Gamma_1| = \frac{3}{2}m+2$ and $|\Gamma_2| = \frac{1}{2}m$. In this case $\mathbb{C}^3/\Gamma$ is a complete intersection given by the equations

   $$\begin{cases} u^2 - y(x - 4y^m) = 0 \\ v^2 - xz = 0 \end{cases} \quad (3.24)$$

   While $m$ is odd, the group generators are

   $$M_1 = \begin{pmatrix} 0 & -1 & 0 \\ -1 & 0 & 0 \\ 0 & 0 & -1 \end{pmatrix}, \quad M_2 = \begin{pmatrix} iw & 0 & 0 \\ 0 & iw^{-1} & 0 \\ 0 & 0 & -1 \end{pmatrix}, \quad (3.25)$$

   where $w = e^{2\pi i/2m}$. For this group we have $r = \frac{1}{2}(m+1)$, $f = m+4$. $\mathbb{C}^3/\Gamma$ is a complete intersection

   $$\begin{cases} u^2 - y(x + 4y^m) = 0 \\ v^2 - xz = 0 \end{cases} \quad (3.26)$$

### 3.2.2 Sporadic cases $E^{(k)}$

In this section, we present other sporadic cases of $\Gamma \subset U(2)$ with a hypersurface or complete intersection description, which are labeled by $E^{(k)}$. For the details of invariant polynomials, see [74].

- $E^{(1)} \supset \mathbb{E}_6$

  This is the case (3) of [74], the order of group is 72, and the generators are

  $$M_1 = \begin{pmatrix} i & 0 & 0 \\ 0 & -i & 0 \\ 0 & 0 & 1 \end{pmatrix}, \quad M_2 = \begin{pmatrix} 0 & i & 0 \\ i & 0 & 0 \\ 0 & 0 & 1 \end{pmatrix}, \quad M_3 = \begin{pmatrix} \frac{1}{2} + \frac{i}{2} & -\frac{1}{2} + \frac{i}{2} & 0 \\ \frac{1}{2} + \frac{i}{2} & \frac{1}{2} - \frac{i}{2} & 0 \\ 0 & 0 & 1 \end{pmatrix}, \quad M_4 = \begin{pmatrix} w & 0 & 0 \\ 0 & w & 0 \\ 0 & 0 & w^{-2} \end{pmatrix}$$

  where $w = e^{2\pi i/6}$. For this group we have $\chi = 21$, $|\Gamma_1| = 15$, $r = |\Gamma_2| = 5$, $f = 10$.
  $\mathbb{C}^3/\Gamma$ is a hypersurface

  $$z^3 - w(y^2 + 108x^4) = 0. \tag{3.27}$$

- $E^{(2)} \supset \mathbb{D}_4$

  This is the case (4) of [74], the order of group is 24, and the generators are

  $$M_1 = \begin{pmatrix} 0 & -i & 0 \\ -i & 0 & 0 \\ 0 & 0 & 1 \end{pmatrix}, \quad M_2 = \begin{pmatrix} -i & 0 & 0 \\ 0 & i & 0 \\ 0 & 0 & 1 \end{pmatrix}, \quad M_3 = \frac{1}{\sqrt{2}} \begin{pmatrix} \epsilon^7 & \epsilon^{13} & 0 \\ \epsilon^7 & \epsilon & 0 \\ 0 & 0 & \sqrt{2}\epsilon^{16} \end{pmatrix}$$

  where $w = e^{2\pi i/24}$. For this group we have $\chi = 7$, $|\Gamma_1| = 5$, $r = |\Gamma_2| = 1$, $f = 4$.
  $\mathbb{C}^3/\Gamma$ is a hypersurface

  $$z^3 - w(y^3 + 12\sqrt{3}ix^2) = 0. \tag{3.28}$$

- $E^{(3)} \supset \mathbb{E}_7$

  This is the case (5) of [74], the order of group is 96, and the generators are

  $$M_1 = \begin{pmatrix} i & 0 & 0 \\ 0 & -i & 0 \\ 0 & 0 & 1 \end{pmatrix}, \quad M_2 = \begin{pmatrix} 0 & i & 0 \\ i & 0 & 0 \\ 0 & 0 & 1 \end{pmatrix}, \quad M_3 = \begin{pmatrix} \frac{1}{2} + \frac{i}{2} & -\frac{1}{2} + \frac{i}{2} \\ \frac{1}{2} + \frac{i}{2} & \frac{1}{2} - \frac{i}{2} \\ 0 & 0 & 1 \end{pmatrix},$$

  $$M_4 = \begin{pmatrix} \epsilon^3 & 0 & 0 \\ 0 & \epsilon^5 & 0 \\ 0 & 0 & 1 \end{pmatrix}, M_5 = \begin{pmatrix} w & 0 & 0 \\ 0 & w & 0 \\ 0 & 0 & w^{-2} \end{pmatrix}$$

  where $\epsilon = e^{2\pi i/8}$ and $w = e^{2\pi i/4}$. For this group we have $\chi = 16$, $|\Gamma_1| = 12$,
  $r = |\Gamma_2| = 3$ and $f = 9$.
  $\mathbb{C}^3/\Gamma$ is a hypersurface

  $$z^2 + w(108x^3 - xy^3) = 0. \tag{3.29}$$

- $E^{(4)} \supset \mathbb{E}_6$

  This is the case (6) of [74], the order of group is 48, and the generators are

  $$M_1 = \begin{pmatrix} 0 & -i & 0 \\ -i & 0 & 0 \\ 0 & 0 & 1 \end{pmatrix}, \quad M_2 = \begin{pmatrix} -i & 0 & 0 \\ 0 & i & 0 \\ 0 & 0 & 1 \end{pmatrix}, \quad M_3 = \frac{1}{2} \begin{pmatrix} -1-i & 1-i & 0 \\ -1-i & -1+i & 0 \\ 0 & 0 & 2 \end{pmatrix},$$

  $$M_4 = \frac{1}{\sqrt{2}} \begin{pmatrix} \epsilon^5 & 0 & 0 \\ 0 & \epsilon^7 & 0 \\ 0 & 0 & -1 \end{pmatrix}$$

where $w = e^{2\pi i/8}$. For this group we have $\chi = 8$, $|\Gamma_1| = 6$, $r = |\Gamma_2| = 1$ and $f = 5$.

$\mathbb{C}^3/\Gamma$ is a hypersurface

$$z^2 + w(108x^4 - y^3) = 0. \tag{3.30}$$

- $E^{(5)} \supset \mathbb{E}_6$

  This is the case (7) of [74], the order of group is 96, and the generators are

  $$M_1 = \begin{pmatrix} 0 & -i & 0 \\ -i & 0 & 0 \\ 0 & 0 & 1 \end{pmatrix}, \quad M_2 = \begin{pmatrix} -1 & 0 & 0 \\ 0 & 1 & 0 \\ 0 & 0 & -1 \end{pmatrix}, \quad M_3 = \frac{1}{2}\begin{pmatrix} -1+i & -1-i & 0 \\ -1+i & 1+i & 0 \\ 0 & 0 & 2 \end{pmatrix},$$

  $$M_4 = \frac{1}{\sqrt{2}}\begin{pmatrix} -1 & 0 & 0 \\ 0 & -i & 0 \\ 0 & 0 & -i \end{pmatrix}$$

  For this group we have $\chi = 16$, $|\Gamma_1| = 11$, $r = |\Gamma_2| = 4$ and $f = 7$.

  $\mathbb{C}^3/\Gamma$ is a hypersurface

  $$108z^4 + w(y^2 - x^3) = 0. \tag{3.31}$$

- $E^{(6)} \supset \mathbb{E}_6$

  This is the case (21) of [74], the order of group is 48, and the generators are

  $$M_1 = \begin{pmatrix} i & 0 & 0 \\ 0 & -i & 0 \\ 0 & 0 & 1 \end{pmatrix}, \quad M_2 = \begin{pmatrix} 0 & i & 0 \\ i & 0 & 0 \\ 0 & 0 & 1 \end{pmatrix}, \quad M_3 = \begin{pmatrix} \frac{1}{2}+\frac{i}{2} & -\frac{1}{2}+\frac{i}{2} & 0 \\ \frac{1}{2}+\frac{i}{2} & \frac{1}{2}-\frac{i}{2} & 0 \\ 0 & 0 & 1 \end{pmatrix}, \quad M_4 = \begin{pmatrix} i & 0 & 0 \\ 0 & i & 0 \\ 0 & 0 & -1 \end{pmatrix}.$$

  For this group we have $\chi = 14$, $|\Gamma_1| = 10$, $r = |\Gamma_2| = 3$ and $f = 7$.

  $\mathbb{C}^3/\Gamma$ is a complete intersection

  $$\begin{cases} y^3 - z^2 = 108x^2 \\ u^2 - vx = 0. \end{cases} \tag{3.32}$$

- $E^{(7)} \supset \mathbb{E}_7$

  This is the case (22) of [74], the order of group is 144, and the generators are

  $$M_1 = \begin{pmatrix} i & 0 & 0 \\ 0 & -i & 0 \\ 0 & 0 & 1 \end{pmatrix}, \quad M_2 = \begin{pmatrix} 0 & i & 0 \\ i & 0 & 0 \\ 0 & 0 & 1 \end{pmatrix}, \quad M_3 = \begin{pmatrix} \frac{1}{2}+\frac{i}{2} & -\frac{1}{2}+\frac{i}{2} & 0 \\ \frac{1}{2}+\frac{i}{2} & \frac{1}{2}-\frac{i}{2} & 0 \\ 0 & 0 & 1 \end{pmatrix},$$

  $$M_4 = \begin{pmatrix} \epsilon^3 & 0 & 0 \\ 0 & \epsilon^5 & 0 \\ 0 & 0 & 1 \end{pmatrix}, M_5 = \begin{pmatrix} w & 0 & 0 \\ 0 & w & 0 \\ 0 & 0 & w^{-2} \end{pmatrix}$$

  where $\epsilon = e^{2\pi i/8}$ and $w = e^{2\pi i/6}$. For this group we have $\chi = 24$, $|\Gamma_1| = 16$, $r = |\Gamma_2| = 7$ and $f = 9$.

$\mathbb{C}^3/\Gamma$ is a complete intersection

$$\begin{cases} yv - u^3 = 0 \\ z^2 + 108x^3 = xy\,. \end{cases} \qquad (3.33)$$

- $E^{(8)} \supset \mathbb{E}_7$

  This is the case (23) of [74], the order of group is 192, and the generators are

  $$M_1 = \begin{pmatrix} i & 0 & 0 \\ 0 & -i & 0 \\ 0 & 0 & 1 \end{pmatrix}, \quad M_2 = \begin{pmatrix} 0 & i & 0 \\ i & 0 & 0 \\ 0 & 0 & 1 \end{pmatrix}, \quad M_3 = \begin{pmatrix} \frac{1}{2}+\frac{i}{2} & -\frac{1}{2}+\frac{i}{2} & 0 \\ \frac{1}{2}+\frac{i}{2} & \frac{1}{2}-\frac{i}{2} & 0 \\ 0 & 0 & 1 \end{pmatrix},$$

  $$M_4 = \begin{pmatrix} \epsilon^3 & 0 & 0 \\ 0 & \epsilon^5 & 0 \\ 0 & 0 & 1 \end{pmatrix}, M_5 = \begin{pmatrix} w & 0 & 0 \\ 0 & w & 0 \\ 0 & 0 & w^{-2} \end{pmatrix}$$

  where $\epsilon = e^{2\pi i/8}$ and $w = e^{2\pi i/8}$. For this group we have $\chi = 32$, $|\Gamma_1| = 21$, $r = |\Gamma_2| = 10$ and $f = 11$.

  $\mathbb{C}^3/\Gamma$ is a complete intersection

  $$\begin{cases} z^2 + 108ux - uy^3 = 0 \\ u^2 - vx = 0 \end{cases} \qquad (3.34)$$

- $E^{(9)} \supset \mathbb{E}_8$

  This case and the next two cases correspond to the case (8) of [74]. The order of group is 240, and the generators are

  $$M_1 = \frac{1}{\sqrt{5}} \begin{pmatrix} \eta^4 - \eta & \eta^2 - \eta^3 & 0 \\ \eta^2 - \eta^3 & \eta - \eta^4 & 0 \\ 0 & 0 & \sqrt{5} \end{pmatrix}, \quad M_2 = \frac{1}{\sqrt{5}} \begin{pmatrix} \eta^2 - \eta^4 & \eta^4 - \eta & 0 \\ 1 - \eta & \eta^3 - \eta & 0 \\ 0 & 0 & \sqrt{5} \end{pmatrix}, \quad M_3 = \begin{pmatrix} w & 0 & 0 \\ 0 & w & 0 \\ 0 & 0 & w^{-2} \end{pmatrix}$$

  where $\eta = e^{2\pi i/5}$ and $w = e^{2\pi i/4}$. For this group we have $\chi = 18$, $|\Gamma_1| = 13$, $r = |\Gamma_2| = 4$ and $f = 9$.

  $\mathbb{C}^3/\Gamma$ is a hypersurface

  $$z^2 + w(-1728x^5 + y^3) = 0\,. \qquad (3.35)$$

- $E^{(10)} \supset \mathbb{E}_8$

  The order of group is 360, and the generators are

  $$M_1 = \frac{1}{\sqrt{5}} \begin{pmatrix} \eta^4 - \eta & \eta^2 - \eta^3 & 0 \\ \eta^2 - \eta^3 & \eta - \eta^4 & 0 \\ 0 & 0 & \sqrt{5} \end{pmatrix}, \quad M_2 = \frac{1}{\sqrt{5}} \begin{pmatrix} \eta^2 - \eta^4 & \eta^4 - \eta & 0 \\ 1 - \eta & \eta^3 - \eta & 0 \\ 0 & 0 & \sqrt{5} \end{pmatrix}, \quad M_3 = \begin{pmatrix} w & 0 & 0 \\ 0 & w & 0 \\ 0 & 0 & w^{-2} \end{pmatrix}$$

where $\eta = e^{2\pi i/5}$ and $w = e^{2\pi i/6}$. For this group we have $\chi = 27$, $|\Gamma_1| = 18$, $r = |\Gamma_2| = 8$ and $f = 10$.

$\mathbb{C}^3/\Gamma$ is a hypersurface

$$z^3 + w(-1728x^5 + y^2) = 0\,. \tag{3.36}$$

- $E^{(11)} \supset \mathbb{E}_8$

  The order of group is 600, and the generators are

  $$M_1 = \frac{1}{\sqrt{5}}\begin{pmatrix} \eta^4 - \eta & \eta^2 - \eta^3 & 0 \\ \eta^2 - \eta^3 & \eta - \eta^4 & 0 \\ 0 & 0 & \sqrt{5} \end{pmatrix}\,, \ M_2 = \frac{1}{\sqrt{5}}\begin{pmatrix} \eta^2 - \eta^4 & \eta^4 - \eta & 0 \\ 1 - \eta & \eta^3 - \eta & 0 \\ 0 & 0 & \sqrt{5} \end{pmatrix}\,, \ M_3 = \begin{pmatrix} w & 0 & 0 \\ 0 & w & 0 \\ 0 & 0 & w^{-2} \end{pmatrix}$$

  where $\eta = e^{2\pi i/5}$ and $w = e^{2\pi i/10}$. For this group we have $\chi = 45$, $|\Gamma_1| = 28$, $r = |\Gamma_2| = 16$ and $f = 12$.

  $\mathbb{C}^3/\Gamma$ is a hypersurface

  $$1728z^5 - w(x^3 + y^2) = 0\,. \tag{3.37}$$

## 3.3 Infinite series of non-abelian finite subgroups of $SU(3)$

### 3.3.1 $\Delta(3n^2)$ series

The generators of $\Delta(3n^2)$ groups are:

$$E = \begin{pmatrix} 0 & 1 & 0 \\ 0 & 0 & 1 \\ 1 & 0 & 0 \end{pmatrix}\,, \ L_n = \begin{pmatrix} \omega & 0 & 0 \\ 0 & \omega^{-1} & 0 \\ 0 & 0 & 1 \end{pmatrix} \tag{3.38}$$

where $\omega = e^{2\pi i/n}$.

The number of conjugacy classes can be computed as

$$\begin{aligned}
|\Gamma_0| &= 1\,, \ |\Gamma_1| = \frac{1}{6}(n^2 + 3n + 8)\,, \ |\Gamma_2| = \frac{1}{6}(n-1)(n-2)\,, \ 3 \nmid n\,, \\
|\Gamma_0| &= 1\,, \ |\Gamma_1| = \frac{1}{6}(n^2 + 3n + 36)\,, \ |\Gamma_2| = \frac{1}{6}(n^2 - 3n + 6)\,, \ 3 \mid n\,.
\end{aligned} \tag{3.39}$$

The gauge and flavor rank are

$$\begin{aligned}
r &= \frac{1}{6}(n-1)(n-2)\,, \ f = n+1\,, \ 3 \nmid n\,, \\
r &= \frac{1}{6}(n^2 - 3n + 6)\,, \ f = n+5\,, \ 3 \mid n\,.
\end{aligned} \tag{3.40}$$

For $\mathbb{C}^3/\Delta(3n^2)$ theories, they can be written as a hypersurface singularity in $\mathbb{C}^4$:

$$-16w^3z^n + 24wxz^n + 24yz^n + 72z^{2n} + 3w^2x^2 - 3w^4x - 4w^3y + w^6 + 4wxy - x^3 + 8y^2 = 0\,. \tag{3.41}$$

### 3.3.2 $\Delta(6n^2)$ series

The generators of $\Delta(6n^2)$ groups are:

$$E = \begin{pmatrix} 0 & 1 & 0 \\ 0 & 0 & 1 \\ 1 & 0 & 0 \end{pmatrix}, \quad I = \begin{pmatrix} 0 & 0 & -1 \\ 0 & -1 & 0 \\ -1 & 0 & 0 \end{pmatrix}, \quad L_n = \begin{pmatrix} \omega & 0 & 0 \\ 0 & \omega^{-1} & 0 \\ 0 & 0 & 1 \end{pmatrix} \tag{3.42}$$

where $\omega = e^{2\pi i/n}$.

The gauge and flavor rank are

$$r = \begin{cases} \frac{1}{12}(n^2 + 6n) - 1, & n = 6k, \\ \frac{1}{12}(n+7)(n-1), & n = 6k+1 \text{ or } 6k+5, \\ \frac{1}{12}(n+8)(n-2), & n = 6k+2 \text{ or } 6k+4, \\ \frac{1}{12}(n^2 + 6n - 3), & n = 6k+3 \end{cases}$$

$$\tag{3.43}$$

$$f = \begin{cases} \frac{1}{2}n + 5, & n = 6k, \\ \frac{1}{2}n + \frac{3}{2}, & n = 6k+1 \text{ or } 6k+5, \\ \frac{1}{2}n + 3, & n = 6k+2 \text{ or } 6k+4, \\ \frac{1}{2}n + \frac{7}{2}, & n = 6k+3 \end{cases}$$

For $n = 2k + 1$ these theories can be described as a complete intersection in $\mathbb{C}^5$:

$$\begin{cases} 27u^n + 4u^{\frac{n-1}{2}}vx - 10u^{\frac{n-1}{2}}vy - xy^2 + 2y^3 + z^2 = 0, \\ v^2 - ux - 2uy = 0. \end{cases} \tag{3.44}$$

When $n$ is even, for $n = 2k$ ($k \neq 1$) these theories can be described as a hypersurface in $\mathbb{C}^4$:

$$-20w^{\frac{n}{2}+1}x^3 + 36w^{\frac{n}{2}+1}xy + 108w^{n+1} + 5wx^2y^2 - 4wx^4y + wx^6 - 2wy^3 + 4z^2 = 0. \tag{3.45}$$

### 3.3.3 $C_{n,l}^{(k)}$ series

1. $(r, k, l) = (3, 1, l)$, $3|l$

   A series of simple examples is given by $(r, k, l) = (3, 1, l)$ where $n = rl = 3l$ and $3|l$, whose generators are:

$$E = \begin{pmatrix} 0 & 1 & 0 \\ 0 & 0 & 1 \\ 1 & 0 & 0 \end{pmatrix}, \quad B_{9,1} = \begin{pmatrix} w & 0 & 0 \\ 0 & w & 0 \\ 0 & 0 & w^{-2} \end{pmatrix}, \quad G_{7,7} = \begin{pmatrix} 1 & 0 & 0 \\ 0 & w^{-3} & 0 \\ 0 & 0 & w^3 \end{pmatrix}. \tag{3.46}$$

   where $w = e^{2\pi i/n}$. For this series we have

$$|\Gamma_1| = \frac{1}{2}l^2 + \frac{1}{2}l + 6,$$
$$r = |\Gamma_2| = \frac{1}{2}l^2 - \frac{1}{2}l + 1. \tag{3.47}$$

   $\mathbb{C}^3/\Gamma$ is a hypersurface given by the equation:

$$w^l x^2 + 3w^{2l}x - 6w^l yz + 9w^{3l} - xyz + y^3 + z^3 = 0. \tag{3.48}$$

2. $(r, k, l) = (7, 2, l)$

Another example is given by $(r, k, l) = (7, 2, l)$ where $n = rl = 7l$, whose generators are:

$$E = \begin{pmatrix} 0 & 1 & 0 \\ 0 & 0 & 1 \\ 1 & 0 & 0 \end{pmatrix}, \quad B_{7,2} = \begin{pmatrix} w & 0 & 0 \\ 0 & w^2 & 0 \\ 0 & 0 & w^{-3} \end{pmatrix}, \quad G_{7l,7} = \begin{pmatrix} 1 & 0 & 0 \\ 0 & w^7 & 0 \\ 0 & 0 & w^{-7} \end{pmatrix} \qquad (3.49)$$

where $w = e^{2\pi i/n}$. When $3 \mid l$, $f = l + 5$ while when $3 \nmid l$, $f = l + 1$. For this series we have

$$
\begin{aligned}
|\Gamma_1| &= \begin{cases} \frac{21}{2}k^2 + \frac{3}{2}k + 6, \; l = 3k \\ \frac{21}{2}k^2 + \frac{17}{2}k + 3, \; l = 3k + 1 \\ \frac{21}{2}k^2 + \frac{31}{2}k + 7, \; l = 3k + 2 \end{cases} \\
r = |\Gamma_2| &= \begin{cases} \frac{21}{2}k^2 - \frac{3}{2}k + 1, \; l = 3k \\ \frac{21}{2}k^2 + \frac{11}{2}k + 1, \; l = 3k + 1 \\ \frac{21}{2}k^2 + \frac{25}{2}k + 4, \; l = 3k + 2 \end{cases}
\end{aligned} \qquad (3.50)
$$

In this case $\mathbb{C}^3/\Gamma$ is a complete intersection given by the equations:

$$
\begin{cases} u^{2l}v - u^l z + vy - x^2 = 0, \\ 9u^{4l} + 3u^{2l}y - 5u^l vx + v^3 - xz + y^2 = 0. \end{cases} \qquad (3.51)
$$

### 3.3.4 $D_{3l,l}^{(1)}$ series, $2 \mid l$

The generators are $E$, $I$, $B_{3l,1}$ where

$$B_{3l,1} = \begin{pmatrix} \omega & 0 & 0 \\ 0 & \omega & 0 \\ 0 & 0 & \omega^{-2} \end{pmatrix} \qquad (3.52)$$

and $\omega = e^{2\pi i/3l}$. We consider the case of $2 \mid l$, where $r = \frac{1}{4}l^2 + 2l - 1$, $f = \frac{1}{2}l + 5$.

In this case $\mathbb{C}^3/\Gamma$ is a complete intersection given by the equations:

$$
\begin{cases} 4v^2u^{l/2} + 12vu^l - 6x^2u^{l/2} + 6zu^{l/2} + 36u^{3l/2} - vx^2 + vz + x^3 + 3xz = 0, \\ y^2 - uz = 0. \end{cases} \qquad (3.53)
$$

## 3.4 Exceptional finite subgroups of $SU(3)$

Here we discuss the classes (E)(F)(G)(H)(I)(L), the "exceptional" finite subgroups of $SU(3)$ that are analogs of $\mathbb{E}$-type finite subgroups of $SU(2)$.

1. $H_{36}$

This group is generated by

$$M_1 = \begin{pmatrix} 1 & 0 & 0 \\ 0 & \omega & 0 \\ 0 & 0 & \omega^2 \end{pmatrix}, \quad M_2 = \begin{pmatrix} 0 & 1 & 0 \\ 0 & 0 & 1 \\ 1 & 0 & 0 \end{pmatrix}, \quad M_3 = h\begin{pmatrix} 1 & 1 & 1 \\ 1 & \omega & \omega^2 \\ 1 & \omega^2 & \omega \end{pmatrix} \qquad (3.54)$$

where $h = (\omega - \omega^2)^{-1}$ and $\omega = e^{2\pi i/3}$. The order of $H_{36}$ is 108.

$$\chi = 14, \ |\Gamma_1| = 9, \ r = |\Gamma_2| = 4, \ f = 5. \tag{3.55}$$

The singular variety $\mathbb{C}^3/H_{36}$ can be written as a complete intersection given by the equations:

$$\begin{cases} 3u^2v + 2u^3 - 36uy - 36vx - v^3 + 432z^2 = 0, \\ u^2x - u^2y - 12x^2 + 9y^2 = 0. \end{cases} \tag{3.56}$$

2. $H_{60}$

This group is generated by

$$M_1 = \begin{pmatrix} 1 & 0 & 0 \\ 0 & -1 & 0 \\ 0 & 0 & -1 \end{pmatrix}, \ M_2 = \begin{pmatrix} 0 & 1 & 0 \\ 0 & 0 & 1 \\ 1 & 0 & 0 \end{pmatrix}, \ M_3 = \frac{1}{2} \begin{pmatrix} -1 & u_2 & u_1 \\ u_2 & u_1 & -1 \\ u_1 & -1 & u_2 \end{pmatrix} \tag{3.57}$$

where $u_1 = \frac{1}{2}(-1 + \sqrt{5})$ and $u_2 = \frac{1}{2}(-1 - \sqrt{5})$. The order of $H_{60}$ is 60, and

$$\chi = 5, \ |\Gamma_1| = 4, \ r = |\Gamma_2| = 0, \ f = 4. \tag{3.58}$$

Therefore there are 4 non-compact exceptional divisors in $\widetilde{\mathbb{C}^3/H_{60}}$ and no compact exceptional divisors.

The singular variety $\mathbb{C}^3/H_{60}$ can be expressed as a hypersurface in $\mathbb{C}^4$:

$$\begin{aligned} 0 = \ & 10125w^2 - 13071240x^4y^2z + 446240256x^9y^2 - 408197440x^6y^3 + 174372625x^3y^4 \\ & + 25927020x^2yz^2 + 40449024x^7yz - 245088256x^{12}y - 17622576x^5z^2 - 13658112x^{10}z \\ & + 54329344x^{15} - 21994875xy^3z - 18907875y^5 - 2777895z^3. \end{aligned} \tag{3.59}$$

3. $H_{72}$

This group is generated by

$$M_1 = \begin{pmatrix} 1 & 0 & 0 \\ 0 & \omega & 0 \\ 0 & 0 & \omega^2 \end{pmatrix}, \ M_2 = \begin{pmatrix} 0 & 1 & 0 \\ 0 & 0 & 1 \\ 1 & 0 & 0 \end{pmatrix}, \ M_3 = h \begin{pmatrix} 1 & 1 & 1 \\ 1 & \omega & \omega^2 \\ 1 & \omega^2 & \omega \end{pmatrix}, \ M_4 = h \begin{pmatrix} 1 & 1 & \omega^2 \\ 1 & \omega & \omega \\ \omega & 1 & \omega \end{pmatrix} \tag{3.60}$$

where $h = (\omega - \omega^2)^{-1}$, $\omega = e^{2\pi i/3}$. The order of $H_{72}$ is 216, and

$$\chi = 16, \ |\Gamma_1| = 10, \ r = |\Gamma_2| = 5, \ f = 5. \tag{3.61}$$

The singular variety $\mathbb{C}^3/H_{72}$ can be described as a hypersurface in $\mathbb{C}^4$:

$$\left(-w^3 + 3wy + 432x^2\right)^2 - 4\left(3y^2z - 3yz^2 + z^3\right) = 0. \tag{3.62}$$

4. $H_{168}$

This group is generated by

$$
M_1 = \begin{pmatrix} \omega & 0 & 0 \\ 0 & \omega^2 & 0 \\ 0 & 0 & \omega^4 \end{pmatrix}, \quad M_2 = \begin{pmatrix} 0 & 1 & 0 \\ 0 & 0 & 1 \\ 1 & 0 & 0 \end{pmatrix}, \quad M_3 = h \begin{pmatrix} \omega^4 - \omega^3 & \omega^2 - \omega^5 & \omega - \omega^6 \\ \omega^2 - \omega^5 & \omega - \omega^6 & \omega^4 - \omega^3 \\ \omega - \omega^6 & \omega^4 - \omega^3 & \omega^2 - \omega^5 \end{pmatrix} \quad (3.63)
$$

where $\omega = e^{2\pi i/7}$ and $h = \frac{1}{7}(\omega + \omega^2 + \omega^4 - \omega^3 - \omega^5 - \omega^6)$. The order of $H_{168}$ is 168, and

$$
\chi = 6, \ |\Gamma_1| = 4, \ r = |\Gamma_2| = 1, \ f = 3. \tag{3.64}
$$

The singular variety $\mathbb{C}^3/H_{168}$ can be described as a hypersurface in $\mathbb{C}^4$:

$$
z^3 + 1728y^7 + 1008zy^4x - 88z^2yx^2 - 60032y^5x^3 + 1088zy^2x^4 + 22016y^3x^6 \\
- 256zx^7 - 2048yx^9 - w^2 = 0. \tag{3.65}
$$

5. $H_{216}$

This group is generated by

$$
M_1 = \begin{pmatrix} 1 & 0 & 0 \\ 0 & \omega & 0 \\ 0 & 0 & \omega^2 \end{pmatrix}, \quad M_2 = \begin{pmatrix} 0 & 1 & 0 \\ 0 & 0 & 1 \\ 1 & 0 & 0 \end{pmatrix}, \quad M_3 = h \begin{pmatrix} 1 & 1 & 1 \\ 1 & \omega & \omega^2 \\ 1 & \omega^2 & \omega \end{pmatrix}, \quad M_4 = \epsilon \begin{pmatrix} 1 & 0 & 0 \\ 0 & 1 & 0 \\ 0 & 0 & \omega \end{pmatrix} \quad (3.66)
$$

where $h = (\omega - \omega^2)^{-1}$, $\omega = e^{2\pi i/3}$ and $\epsilon = e^{4\pi i/9}$. The order of $H_{216}$ is 648, and

$$
\chi = 24, \ |\Gamma_1| = 14, \ r = |\Gamma_2| = 9, \ f = 5. \tag{3.67}
$$

The singular variety $\mathbb{C}^3/H_{216}$ can be described as a hypersurface in $\mathbb{C}^4$:

$$
y^3 - \frac{1}{4}z\left((432w^2 - z - 3y)^2 - 6912x^3\right) = 0. \tag{3.68}
$$

6. $H_{360}$

The group $H_{360}$ is generated by

$$
M_1 = \begin{pmatrix} -1 & 0 & 0 \\ 0 & 0 & -1 \\ 0 & -1 & 0 \end{pmatrix}, \quad M_2 = \begin{pmatrix} 1 & 0 & 0 \\ 0 & \omega^4 & 0 \\ 0 & 0 & \omega \end{pmatrix}, \quad M_3 = \frac{1}{\sqrt{5}} \begin{pmatrix} 1 & \sqrt{2} & \sqrt{2} \\ \sqrt{2} & s & t \\ \sqrt{2} & t & s \end{pmatrix},
$$
$$
M_4 = \frac{1}{\sqrt{5}} \begin{pmatrix} 1 & \sqrt{2}\lambda_1 & \sqrt{2}\lambda_1 \\ \sqrt{2}\lambda_2 & s & t \\ \sqrt{2}\lambda_2 & t & s \end{pmatrix} \tag{3.69}
$$

where $\omega = e^{2\pi i/5}$, $s = \frac{1}{2}(-1 - \sqrt{5})$, $t = \frac{1}{2}(-1 + \sqrt{5})$, $\lambda_1 = \frac{1}{4}(-1 + \sqrt{15}i)$ and $\lambda_2 = \frac{1}{4}(-1 - \sqrt{15}i)$. The order of $H_{360}$ is 1080, and

$$
\chi = 17, \ |\Gamma_1| = 11, \ |\Gamma_2| = 5, \ f = 6. \tag{3.70}
$$

## 4 Examples

In this section, we provide a number of notable examples of $\mathbb{C}^3/\Gamma$, their resolutions and the 5d SCFT interpretation. If the singularity can be embedded in a singular elliptic Calabi-Yau threefold, one can also study the 6d (1,0) SCFT from F-theory. The 6d and 5d theories are naturally related, see section 2.5. We put additional examples in appendix B.

### 4.1 $\Delta(3n^2)$, $n = 3k$

In this section we study the cases of $X = \mathbb{C}^3/\Delta(3n^2)$, $n = 3k$. One can dimensionally reduce $T_X^{6d}$ on $S^1$, and then decouple a 5d hypermultiplet to get $T_X^{5d}$.

#### 4.1.1 6d interpretation

Starting with the singular equation

$$-16w^3z^n+24wxz^n+24yz^n+72z^{2n}+3w^2x^2-3w^4x-4w^3y+w^6+4wxy-x^3+8y^2 = 0\,, \quad (4.1)$$

we can rescale and shift coordinates and rewrite it as

$$\begin{cases} y^2 = x^3 + \frac{1}{12}w(8U - 9w^3)x + \frac{1}{108}(8U^2 - 36Uw^3 + 27w^6) \\ U = w^3 - 27z^n \end{cases} \quad (4.2)$$

The discriminant is

$$\begin{aligned} \Delta &= 4z^n(27z^n - w^3)^3 \\ &= \frac{4}{27}U^3(U - w^3)\,. \end{aligned} \quad (4.3)$$

If $n = 3k$, the non-compact curves supporting Kodaira singular fiber are

$$\begin{aligned} z = 0 &: I_{n,s},\ G_{F,6d} = SU(n) \\ w - 3e^{2\pi im/3}z = 0 &: I_{3,s},\ G_{F,6d} = SU(3) \quad (m = 0, 1, 2)\,. \end{aligned} \quad (4.4)$$

The total flavor symmetry is

$$G_{F,6d} \supset SU(n) \times SU(3)^3\,. \quad (4.5)$$

Now we discuss the tensor branch geometry of a few $\Delta(3n^2)$ theories with small $n = 3k$. For $n = 3$, the base blow up is $(w, z; \delta)$, and the resulting $(f, g, \Delta)$ are

$$\begin{aligned} f &= -\frac{1}{12}w(w^3 + 216z^3) \\ g &= -\frac{w^6}{108} + 5w^3z^3 + 54z^6 \\ \Delta &= -4z^3(w^3 - 27z^3)^3\,, \end{aligned} \quad (4.6)$$

The tensor branch is

$$\begin{array}{ccc} & [SU(3)] & \\ & | & \\ [SU(3)]- & 1 & -[SU(3)]\,, \\ & | & \\ & [SU(3)] & \end{array} \quad (4.7)$$

with only a single $(-1)$-curve. Hence the corresponding 6d $(1,0)$ SCFT is the same as rank-1 E-string theory.

For higher $n = 3k$ $(k > 1)$, the base blow ups are

$$(w, z; \delta_1)$$
$$(w, \delta_i; \delta_{i+1}) \quad (i = 1, \ldots, k-1) \,.$$
(4.8)

The resulting $(f, g, \Delta)$ are

$$f = -\frac{1}{12} w \left( w^3 + 216 z^{3k} \prod_{i=1}^{k-1} \delta_i^{3k-3i} \right)$$

$$g = -\frac{w^6}{108} + 5 w^3 z^{3k} \prod_{i=1}^{k-1} \delta_i^{3k-3i} + 54 z^{6k} \prod_{i=1}^{k-1} \delta_i^{6k-6i}$$
(4.9)

$$\Delta = -4 z^{3k} \left( w^3 - 27 z^{3k} \prod_{i=1}^{k-1} \delta_i^{3k-3i} \right)^3 \prod_{i=1}^{k-1} \delta_i^{3k-3i} \,.$$

From the Weierstrass model, we read of the following tensor branch

$$
\begin{array}{ccccccc}
& & & & & [SU(3)] & \\
& & & & & | & \\
[SU(3k)] - \overset{\mathfrak{su}(3k-3)}{2} - \overset{\mathfrak{su}(3k-6)}{2} - \cdots - \overset{\mathfrak{su}(3)}{2} & -1- & [SU(3)] \\
& & & & & | & \\
& & & & & [SU(3)] &
\end{array}
$$
(4.10)

The compact curves from left to right are $\delta_1 = 0$, $\delta_2 = 0$, $\ldots$, $\delta_k = 0$. The flavor symmetry $SU(3)^3$ attached to the $(-1)$-curve on the right is enhanced to $E_6$, after the three $I_{3,s}$ loci collide into an $IV^*_{ns}$, and the tensor branch becomes

$$[SU(3k)] - \overset{\mathfrak{su}(3k-3)}{2} - \overset{\mathfrak{su}(3k-6)}{2} - \cdots - \overset{\mathfrak{su}(3)}{2} - 1 - [E_6] \,.$$
(4.11)

These 6d SCFTs are very-Higgsable tensor-gauge quiver theories that naturally appear in the literature, such as [79].

### 4.1.2 5d interpretation

The 5d interpretation of these theories has also been studied in [100] using a different approach. In this section we study the resolved geometry explicitly.

1. $n = 3$

   The non-isolated singular equation is

   $$w^6 - 3w^4 x + 3w^2 x^2 - x^3 - 4w^3 y + 4wxy + 8y^2 - 16w^3 z^3 + 24wxz^3 + 24yz^3 + 72z^6 = 0 \,.$$
   (4.12)

The equation is homogeneous with weights

$$d(x) = \frac{1}{3} \ , \ d(y) = \frac{1}{2} \ , \ d(z) = \frac{1}{6} \ , \ d(w) = \frac{1}{6} \ , \tag{4.13}$$

hence the singular point $x = y = z = w = 0$ can be removed after the weighted blow up of ambient space

$$(x^{(2)}, y^{(3)}, z^{(1)}, w^{(1)}; \delta_1) \,. \tag{4.14}$$

After this blow up, the equation of compact exceptional divisor $\delta_1 = 0$ has the same form as (4.12). There are still four cod-2 $A_2$ singularities at the following curves

$$x - w^2 = y = z = 0 \ , \ x - \frac{w^2}{3} = y - \frac{w^3}{9} = z^3 - \frac{w^3}{27} = 0 \,, \tag{4.15}$$

which all intersects $\delta_1 = 0$. Note that the last one has three components. Now after the four $A_2$ singularities are resolved, the compact surface $S_1 : \delta_1 = 0$ has the form of a generalized dP$_8$ of type $4\mathbf{A}_2$ (the $(-2)$-curves on $S_1$ consists of four disconnected $A_2$ Dynkin diagram).

In the limit of zero surface volume, the UV SCFT is the rank-1 Seiberg $E_8$ theory. We do not write down the fully resolved equation for simplicity.

2. $n = 6$

For $n = 6$, the whole resolution sequence giving rise to compact divisors is

$$(x^{(2)}, y^{(3)}, z^{(1)}, w^{(1)}; \delta_1) \ , \ (x^{(2)}, y^{(3)}, w^{(1)}, \delta_1^{(1)}; \delta_2) \ , \ (x_2, y, \delta_1; \delta_3)$$
$$(y, \delta_3; \delta_4) \tag{4.16}$$

The resolved equation at this step is

$$- x_2^3 \delta_3 + 4wx_2 y + 8y^2 \delta_4 + 8\delta_1^3 \delta_3 z^6 (w^3 + 3\delta_3 \delta_4 wx_2 + 3\delta_3 \delta_4^2 y)z^6 + 72\delta_1^6 \delta_3^4 \delta_4^3 z^{12} = 0 \,. \tag{4.17}$$

The compact divisors are $S_1 : \delta_1 = 0$, $S_2 : \delta_2 = 0$, $S_3 : \delta_3 = 0$, $S_4 : \delta_4 = 0$ and they are all irreducible. This matches the expected $r = 4$. The intersection numbers between them can be computed using the SR ideal from the resolution sequence

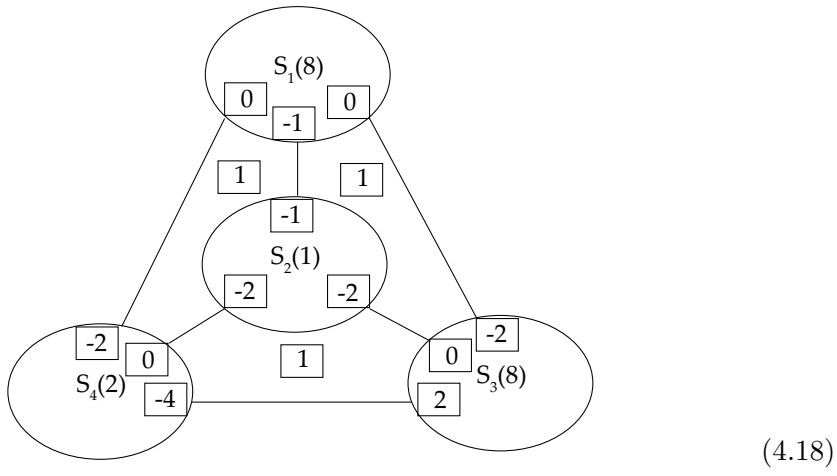

$$\tag{4.18}$$

Since $S_1 \cdot S_3$ is a section curve on $S_3$ but a ruling curve on $S_1$, the geometry does not have a ruling structure and an IR gauge theory description.

There are still three codimension-two $A_2$ singularities at the loci

$$x_2 + 6e^{\frac{2\pi im}{3}}\delta_1^2\delta_3\delta_4 z^4 = y - 3\delta_1^3\delta_3^2\delta_4 z^6 = w - 3e^{\frac{2\pi im}{3}}\delta_1\delta_3\delta_4 z^2 \quad (m = 0, 1, 2) \quad (4.19)$$

and an $A_5$ singularity at

$$x_2 = y = z = 0. \quad (4.20)$$

After shifting coordinates, these codimension-two singularities can be resolved in the usual way, without further blowing up the compact divisors $S_i$. We will not write out the fully resolved equation.

The flavor symmetry from the codimension-two singularity is $SU(6) \times SU(3)^3$, which is further enhanced to[2]

$$G_F = SU(6) \times E_6. \quad (4.21)$$

3. $n = 9$

For $n = 9$, the resolution sequence giving rise to compact divisors is

$$(x^{(2)}, y^{(3)}, z^{(1)}, w^{(1)}; \delta_1) \,,\; (x^{(2)}, y^{(3)}, w^{(1)}, \delta_1^{(1)}; \delta_2) \,,\; (x^{(2)}, y^{(3)}, w^{(1)}, \delta_2^{(1)}; \delta_3)$$
$$(x_2, y, \delta_1; \delta_4) \,,\; (x_2, y, \delta_4; \delta_5) \,,\; (x_2, y, \delta_5; \delta_6) \,,\; (y, \delta_4; \delta_7) \quad (4.22)$$
$$(y, \delta_5; \delta_8) \,,\; (x, y, \delta_2; \delta_9) \,,\; (y, \delta_9; \delta_{10})$$

The intersection numbers are shown below

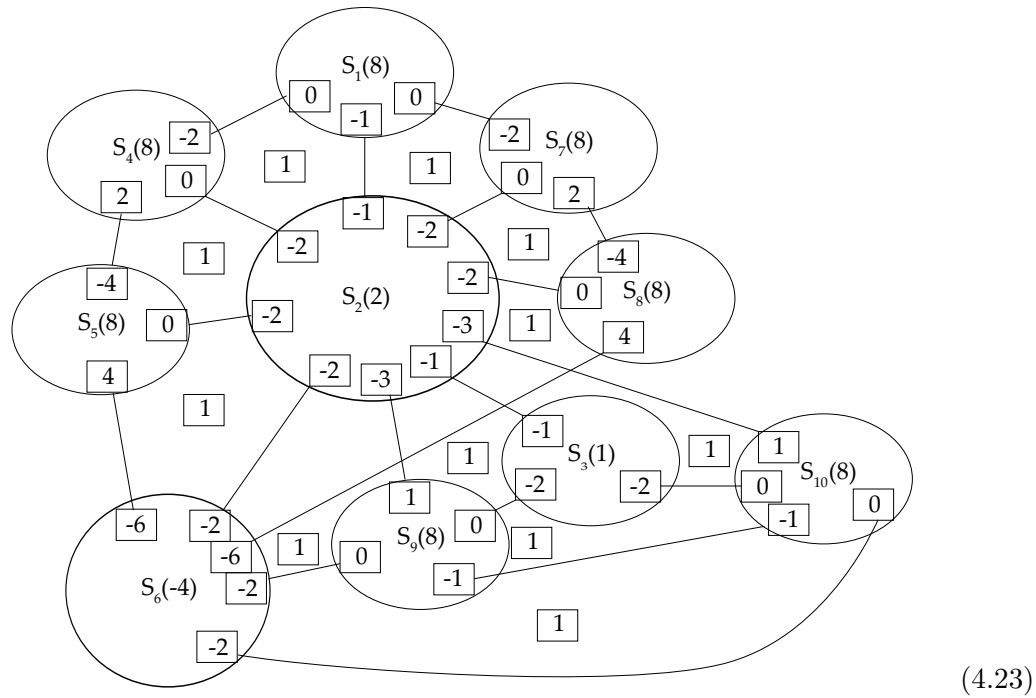

$$(4.23)$$

[2]One can also view the 5d theory as a descendant of the KK reduction of the 6d theory (4.11), and the flavor symmetry $SU(3)^3$ is already enhanced to $E_6$ in that case.

The flavor symmetry is enhanced to

$$G_F = SU(9) \times E_6 \,. \tag{4.24}$$

For higher $n = 3k$, the flavor symmetry is similarly

$$G_F = SU(3k) \times E_6 \,. \tag{4.25}$$

## 4.2 A new 6d (1,0) SCFT from $\mathbb{C}^3/\Delta(48)$

In this section we consider the case of $\Delta(3n^2)$, $n = 4$. An interesting observation is that the tensor branch geometry of $T_X^{6d}$ consists of a $(-1)$-curve with an Kodaira $I_1$ singularity. From resolution geometry and gravitational anomaly arguments, we postulate that $T_X^{6d}$ is different from the rank-1 E-string. We also work out $T_X^{5d}$ from the resolution of $\mathbb{C}^3/\Delta(48)$, which turns out to be the rank-1 Seiberg $E_5$ theory.

### 4.2.1 6d interpretation

From the equation

$$\begin{cases} y^2 = x^3 + \frac{1}{12}w(8U - 9w^3)x + \frac{1}{108}(8U^2 - 36Uw^3 + 27w^6) \\ U = w^3 - 27z^4 \,, \end{cases} \tag{4.26}$$

we can see that the non-compact curves supporting Kodaira singular fiber are

$$\begin{aligned} z = 0 &: I_{4,s}, G_{F,6d} = SU(4) \\ U = w^3 - 27z^4 = 0 &: I_{3,s}, G_{F,6d} = SU(3) \,. \end{aligned} \tag{4.27}$$

The total 6d flavor symmetry is

$$G_{F,6d} = SU(4) \times SU(3) \,. \tag{4.28}$$

To get the tensor branch, one blows up the base $(w, z; \delta)$, and the resulting $(f, g, \Delta)$ are

$$\begin{aligned} f &= -\frac{1}{12}w(w^3 + 216z^4\delta) \\ g &= -\frac{w^6}{108} + 5w^3z^4\delta + 54z^8\delta^2 \\ \Delta &= -4z^4(w^3 - 27z^4\delta)^3\delta \\ &= -4z^4U^3\delta \,, \end{aligned} \tag{4.29}$$

The tensor branch is

$$\tag{4.30}$$

with a single $(-1)$-curve carrying $I_1$ singular fiber. At the collision point $z = \delta = 0$ between $I_{4,s}$ and $I_1$, the $I_{4,s}$ fiber splits into $I_{5,s}$. At the collision point $U = \delta = 0$ between $I_{3,s}$ and $I_1$, the Kodaira fiber type becomes type IV. Hence at $z = \delta = 0$, there are 4 localized neutral hypermultiplets in the **4** representation of the flavor symmetry group $SU(4)$. At $U = \delta = 0$, the enhancement of $I_{3,s} \to IV$ does not give rise to matter hypermultiplets as there are no additional $\mathbb{P}^1$s from the enhancement.

From the above discussions of splitting of Kodaira fiber on $\delta = 0$, we can draw the configuration of curves on the vertical divisor $S$ over $\delta = 0$:

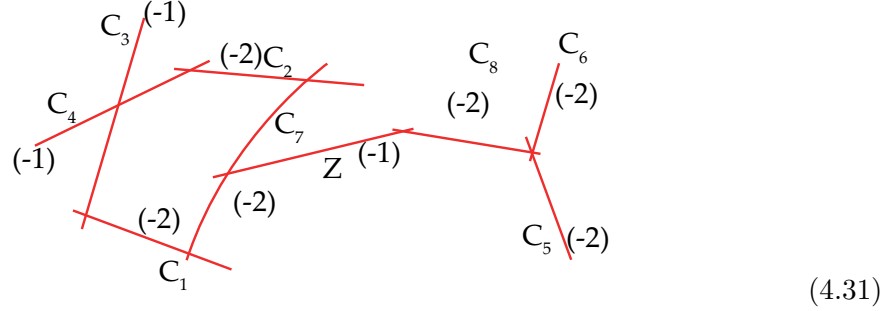

$$(4.31)$$

$C_1$, $C_2$, $C_3$, $C_4$ and $C_7$ comes from the split $I_{5,s}$, $C_5$, $C_6$ and $C_8$ comes from the type IV fiber, $Z$ corresponds to the intersection curve with the zero-section. Note that $C_3$ and $C_4$ have normal bundle $\mathcal{O}(-1) \oplus \mathcal{O}(-1)$, because M2-brane wrapping modes over them would give rise to matter hypermultiplets under $SU(4)$.

For the topology of $S$, it is different from a rational elliptic surface as the Picard rank is smaller than 10. The $h^{1,1}(S) = 7$ generators include the base curve $Z$, the generic fiber and the five $(-2)$-curves generating $G_{F,6d} = SU(3) \times SU(4)$. The generic $(I_1)$ fiber of $S$, $F$, is a cubic curve with a double point singularity. Because of the double point singularity, the curve $F$ has geometric genus $g(F) = 0$. In terms of the Picard group generator $h, e_1, \ldots, e_6$ of a dP$_6$, the fiber $F$ can be written as

$$F = 3h - 2e_1 - e_2 - e_3 - e_4 - e_5 - e_6 \,. \tag{4.32}$$

Although $S$ is a singular dP$_6$ surface rather than an elliptic surface, the uplift to 6d still works geometrically, since the genus-0 fiber $F$ is interpreted as a singular cubic curve.

An important question is whether this 6d theory is equivalent to rank-1 E-string theory. Positive evidences were provided in [101], where it is showed that for the cases of type $I_1$ or $II$ fiber over a $(-1)$-curve, the flavor symmetry group $G$ on non-compact curves intersecting the $(-1)$-curve is always a subgroup of $E_8$. This is a crucial feature of rank-1 E-string theory. Nonetheless, if the tensor branch shown in (4.30) is embedded in a compact base, after going to the origin of the tensor branch by blowing down the $(-1)$-curve, the number of tensor multiplets $T$ and charged hypermultiplets $H_{\text{charged}}$ are shifted as

$$\Delta T = -1 \,, \ \Delta H_{\text{charged}} = -4 \,. \tag{4.33}$$

The 6d gravitational anomaly cancellation can be written as

$$\Delta H_{\text{moduli}} + \Delta H_{\text{localized}} + \Delta H_{\text{charged}} - \Delta V + 29\Delta T = 0 \,, \tag{4.34}$$

where $H_{\text{moduli}}$ corresponds to the number of complex structure moduli of the CY3, which is unchanged after going into the tensor branch. $H_{\text{localized}}$ denotes additional SCFT sector which contributes to the gravitational anomaly, $V$ denotes the number of vector multiplets.

In this case, we have

$$\begin{aligned}
\Delta H_{\text{localized}} &= -\Delta H_{\text{charged}} - 29\Delta T \\
&= 33\,,
\end{aligned} \tag{4.35}$$

which is bigger than the contribution of a simple rank-1 E-string theory. This is due to the fact that additional hypermultiplets on the tensor branch become a part of the strongly interacting theory at the origin of the tensor branch. In other words, the Higgs branch dimension of this rank-1 theory is bigger than $d_H = 29$ of the rank-1 E-string theory. Conservatively, we conjecture that the 6d theory associated to $\Delta(48)$ is different from the rank-1 E-string theory, but it is possibly rank-1 E-string theory coupled to $-\Delta H_{\text{charged}} = 4$ neutral hypermultiplets.

### 4.2.2    5d interpretation

Now we discuss the 5d interpretation of the non-isolated singular equation

$$w^6 - 3w^4x + 3w^2x^2 - x^3 - 4w^3y + 4wxy + 8y^2 - 16w^3z^4 + 24wxz^4 + 24yz^4 + 72z^8 = 0\,. \tag{4.36}$$

We first do the resolution

$$(y^{(3)}, x^{(2)}, z^{(1)}, w^{(1)}; \delta_1)\,, \tag{4.37}$$

resulting in the compact exceptional divisor $S_1 : \delta_1 = 0$.

After substituting variables

$$x \to x + \frac{w^2}{3}\ ,\ y \to y + \frac{w^3}{9}\,, \tag{4.38}$$

the equation can be rewritten as

$$\begin{cases} -x^3 + 2w^2x^2 + 4wxy - \frac{8}{9}wxU + \frac{8}{81}(81y^2 - 9yU + U^2) = 0 \\ U = w^3 - 27z^4\delta_1\,. \end{cases} \tag{4.39}$$

Now we can blow up the $A_2$ singularity

$$(x, y, U; \delta_2) \tag{4.40}$$

and get the new equation

$$\begin{cases} -x^3\delta_2 + 2w^2x^2 + 4wxy - \frac{8}{9}wxU + \frac{8}{81}(81y^2 - 9yU + U^2) = 0 \\ U\delta_2 = w^3 - 27z^4\delta_1\,. \end{cases} \tag{4.41}$$

To see the remaining singularity, we work in the patch of $U \neq 0$, and substitute

$$\delta_2 = \frac{w^3 - 27z^4\delta_1}{U} \tag{4.42}$$

into the first equation. We again get a hypersurface:

$$\frac{8U^3}{81} - \frac{8}{9}U^2(wx+y) + 2U(w^2x^2 + 2wxy + 4y^2) - x^3(w^3 - 27\delta_1 z^4) = 0. \tag{4.43}$$

Further introducing variables

$$y_2 = y + \frac{wx}{6} \ , \ U_2 = U - \frac{3wx}{2} \ , \tag{4.44}$$

(4.43) can be rewritten as

$$\frac{8}{81}U_2(U_2^2 - 9U_2y_2 + 81y_2^2) - \frac{4}{27}wx(2U_2 - 9y_2)(U_2 + 9y_2) + 27x^3z^4\delta_1 = 0 \tag{4.45}$$

The type $A_3$ singularity at $U_2 = y_2 = z = 0$ can be resolved as

$$(U_2, y_2, z; \delta_3) \ , \ (U_2, y_2, \delta_3; \delta_4) \tag{4.46}$$

The resolved equation at the $U \neq 0$ patch is

$$\frac{8}{81}U_2(U_2^2 - 9U_2y_2 + 81y_2^2)\delta_3\delta_4^2 - \frac{4}{27}wx(2U_2 - 9y_2)(U_2 + 9y_2) + 27x^3z^4\delta_1\delta_3^2 = 0. \tag{4.47}$$

The projective relations are

$$\begin{aligned} &(x\delta_2, y_2\delta_2\delta_3\delta_4^2, z\delta_3\delta_4, w) \\ &(x, y_2\delta_3\delta_4^2, U_2\delta_3\delta_4^2) \\ &(U_2\delta_4, y_2\delta_4, z) \\ &(U_2, y_2, \delta_3) \end{aligned} \tag{4.48}$$

The irreducible non-compact divisors are

$$\begin{aligned} D_1 &: \ \delta_3 = 2U_2 - 9y_2 = 0 \,, \\ D_2 &: \ \delta_3 = U_2 + 9y_2 = 0 \,, \\ D_3 &: \ \delta_4 = 0 \,. \end{aligned} \tag{4.49}$$

In the $U \neq 0$ patch, the equation for the compact exceptional divisor $S_1 : \delta_1 = 0$ is

$$\delta_1 = \frac{8}{81}U_2(U_2^2 - 9U_2y_2 + 81y_2^2)\delta_3\delta_4^2 - \frac{4}{27}wx(2U_2 - 9y_2)(U_2 + 9y_2) = 0. \tag{4.50}$$

The surface $S_1$ still has a double point singularity along the curve $U_2 = y_2 = 0$. The intersection curves $\delta_1 \cdot \delta_3$ and $\delta_1 \cdot \delta_4$ are

$$\begin{aligned} \delta_1 = \delta_3 = 0 &: \quad (2U_2 - 9y_2)(U_2 + 9y_2) = 0 \\ \delta_1 = \delta_4 = 0 &: \quad (2U_2 - 9y_2)(U_2 + 9y_2) = 0 \end{aligned} \tag{4.51}$$

Each of these curves have two components, denote the curves by

$$\begin{aligned} C_1 &: \quad \delta_1 = \delta_3 = 2U_2 - 9y_2 = 0 \\ C_2 &: \quad \delta_1 = \delta_3 = U_2 + 9y_2 = 0 \\ C_3 &: \quad \delta_1 = \delta_4 = 2U_2 - 9y_2 = 0 \\ C_4 &: \quad \delta_1 = \delta_4 = U_2 + 9y_2 = 0 \end{aligned} \tag{4.52}$$

The intersection relations are

$$C_1 = D_1 \cdot S_1 \ , \ \ C_2 = D_2 \cdot S_1 \ , \ \ C_3 + C_4 = D_3 \cdot S_1 \,. \tag{4.53}$$

The other non-compact divisors are

$$\begin{aligned} D_4 : \ \ &\delta_2 = U_2 - \frac{9}{2}(1 + i\sqrt{3})y_2 = 0 \,, \\ D_5 : \ \ &\delta_2 = U_2 - \frac{9}{2}(1 - i\sqrt{3})y_2 = 0 \,, \end{aligned} \tag{4.54}$$

which give rise to the $(-2)$-curves $C_5 = D_4 \cdot S_1$ and $C_6 = D_5 \cdot S_1$. There are also two $(-1)$-curves

$$\begin{aligned} C_7 : \ \ &U = 0 \\ C_8 : \ \ &z = 0 \end{aligned} \tag{4.55}$$

which mutually intersects. The intersection relations between all these curves are shown in the following figure

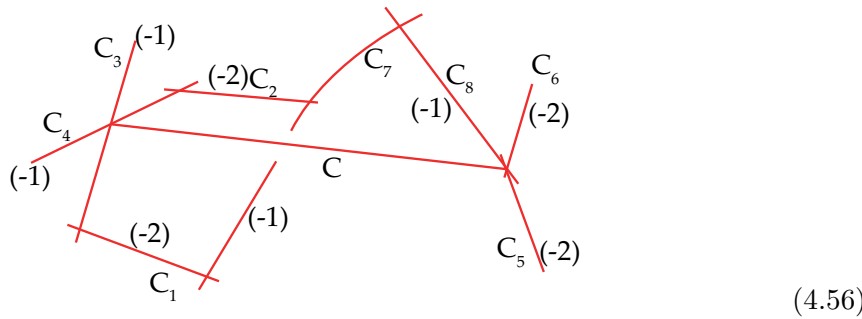

$$\tag{4.56}$$

Since $S_1$ has a double point singularity along the central curve $C : \delta_1 = U_2 = y_2 = 0$, it is not a smooth rational surface. For $C_3$ and $C_4$, they come from the same non-compact divisor $D_3$ and only give rise to a single flavor curve, which is a reminiscence of matter curve splitting in the 6d description in section 4.1.1. Note that $C_3$ and $C_4$ are $(-1)$-curves on $S_1$, as M2-brane wrapping modes over them should give rise to matter hypermultiplets. One can also generate (4.56) by blowing down the $(-1)$-curve labeled by $Z$ in the 6d geometry (4.31).

The codimension-two $A_3$ singularity generates $SU(2)^2 \times U(1)$ flavor symmetry in the 5d theory, and the total flavor symmetry is

$$G_F = SU(3) \times SU(2)^2 \times U(1) \,, \tag{4.57}$$

which can be embedded in the flavor symmetry group of rank-1 Seiberg $E_5$ theory. There are two possible physical interpretations:

1. The rank-1 SCFT from $\mathbb{C}^3/\Delta(48)$ is equivalent to the Seiberg $E_5$ theory. Additional evidences are also provided via brane-web arguments in [100]. Nonetheless, it is unclear how to get the correct BPS states from M2 brane wrapping curves in (4.56).

2. The geometry describes a new rank-1 theory, analogous to the cases with terminal singularities or singular divisors in [7, 8].

To solve the problem, more physical quantities should be computed, such as the Higgs branch dimensions or superconformal indices. Similar subtleties appear in the cases of $\Gamma = E^{(2)}, E^{(4)}, G_3$ and $H_{168}$ as well.

## 4.3 $\Delta(54)$

In this section we discuss the case of $X = \mathbb{C}^3/\Delta(6n^2)$, $n = 3$. In this case $T_X^{6d}$ is non-very-Higgsable. To get $T_X^{5d}$, one need to decouple a single vector multiplet after the KK reduction.

### 4.3.1 6d interpretation

Starting with the singular equation

$$\begin{cases} 27u^{2k+1} + 4vwu^k - 18u^k vx - wx^2 + 4x^3 + y^2 = 0, \\ v^2 - uw = 0, \end{cases} \tag{4.58}$$

we rescale and shift the coordinates and rewrite the equation into a Weierstrass equation

$$\begin{cases} y^2 = x^3 - \frac{1}{12 \times 2^{2/3}}(216u^k v + w^2)x + 27u^{1+2k} + \frac{5}{2}u^k vw - \frac{w^3}{216}, \\ v^2 - uw = 0. \end{cases} \tag{4.59}$$

Thus the singular equation can be viewed as an elliptic fibration over a singular base with $\mathbb{Z}_2$ orbifold singularity at $u = v = w = 0$.

Another equivalent description is to replace

$$u = s^2 , \; v = st , \; w = t^2 , \tag{4.60}$$

and the Weierstrass polynomials become

$$\begin{aligned} f &= -\frac{1}{12 \times 2^{2/3}}(t^4 + 216s^{2k+1}t) \\ g &= -\frac{t^6}{216} + \frac{5}{2}s^{2k+1}t^3 + 27s^{4k+2} . \end{aligned} \tag{4.61}$$

Nonetheless, the base still has a $\mathbb{Z}_2$ singularity at $s = t = 0$. Now we discuss the tensor branch of models with small $k$.

In the case of $k = 1, n = 3$, after the base blow up $(s, t; \delta)$ (or equivalently $(u, v, w; \delta)$), the new $(f, g, \Delta)$ are

$$\begin{aligned} f &= -\frac{1}{12 \times 2^{2/3}}(t^4 + 216s^3t)\delta^2 \\ g &= \left(-\frac{t^6}{216} + \frac{5}{2}s^3t^3 + 27s^6\right)\delta^3 \\ \Delta &= s^3(3s - t)^3(3e^{\frac{2\pi i}{3}}s - t)^3(3e^{\frac{4\pi i}{3}}s - t)^3\delta^6 \end{aligned} \tag{4.62}$$

There is no codimension-two (4,6) locus any more. Over the compact $(-2)$-curve: $\delta = 0$, there is type $I_{0,ns}^*$ singular fiber and $G_2$ gauge group. There are four $I_{3,ns}$ fiber and $SU(2)$ flavor groups over four non-compact curves

$$s = 0 , \; 3s - t = 0 , \; 3e^{\frac{2\pi i}{3}}s - t = 0 , \; 3e^{\frac{4\pi i}{3}}s - t = 0 . \tag{4.63}$$

In fact, if one tunes $G_2$ gauge group over a $(-2)$-curve, the 6d gauge theory is exactly $G_2+4\boldsymbol{F}$. The four matter hypermultiplets exactly locate on the intersection points between $\delta = 0$ and (4.63). The tensor branch of the 6d SCFT is written as

$$
\begin{array}{c}
[SU(2)] \\
| \\
[SU(2)]- \quad \overset{\mathfrak{g}_2}{2} \quad -[SU(2)] \\
| \\
[SU(2)]
\end{array}
\tag{4.64}
$$

One can also collide the four intersection points into a single $(Sp(4), G_2)$ collision, and the tensor branch is

$$
[Sp(4)] - \overset{\mathfrak{g}_2}{2} \,.
\tag{4.65}
$$

The resolution geometry of this SCFT was already worked out in (4.38) of [17]. The triple intersection numbers among three compact surfaces $S_1$, $S_2$ and $S_3$ are

$$
S_1^3 = 8 \ , \ S_2^3 = -24 \ , \ S_3^3 = 8 \ , \ S_3 S_1^2 = -2 \ , \ S_2^2 S_3 = 18 \ , \ S_2 S_3^2 = -12 \,.
\tag{4.66}
$$

### 4.3.2 5d interpretation

First we consider the case of $\mathbb{C}^3/\Delta(6n^2)$ theory with $n = 3$:

$$
\begin{cases}
27u^3 + 4uvx - 18uvy - xy^2 + 4y^3 + z^2 = 0, \\
v^2 - ux = 0.
\end{cases}
\tag{4.67}
$$

We can perform the following resolution sequence

$$
(x, y, z, u, v; \delta_1) \ , \ (z, \delta_1; \delta_2) \ , \ (\delta_1, \delta_2; \delta_3) \,,
\tag{4.68}
$$

and the equations become

$$
\begin{cases}
(27u^3 + 4uvx - 18uvy - xy^2 + 4y^3)\delta_1 + z^2 \delta_2 = 0, \\
v^2 - ux = 0 \,.
\end{cases}
\tag{4.69}
$$

Note that the divisors $\delta_1 = 0$ is an empty set, $\delta_2 = 0$ and $\delta_3 = 0$ are the two compact exceptional divisors.

In the patch of $u \neq 0$, we plug $x = v^2/u$ into the first equation, and get

$$
(27u^4 + 4uv^3 - 18u^2 vy - v^2 y^2 + 4uy^3)\delta_1 + uz^2 \delta_2 = 0 \,.
\tag{4.70}
$$

The remaining codimension-two singularities are type $A_1$ at

$$
u = v = z^2 \delta_2 + 4y^3 \delta_1 = 0 \,,
\tag{4.71}
$$

(note that this curve is non-singular since $u = v = y = z = 0$ is removed in the first blow-up) and type $A_2$ at

$$
z = v - e^{2\pi im/3}y = u - e^{4\pi im/3}y = 0 \quad (m = 0, 1, 2) \,.
\tag{4.72}
$$

The resolution of each type $A_2$ singularity gives rise to a single non-compact exceptional divisor, after

$$(z, v - e^{2\pi im/3}y, u - e^{4\pi im/3}y; \delta_{4+m}) \quad (m = 0, 1, 2), \tag{4.73}$$

the non-compact exceptional divisors $\delta_{4+m}$ $(m = 0, 1, 2)$ are irreducible. The same codimension-two singular loci can be seen from the other patch $x \neq 0$ as well.

Furthermore, there is an $A_1$ singularity at $v = u = x = 0$. Hence in total there are five non-compact exceptional divisors, giving rise to $G_F \supset SU(2)^5$.

We can also compute the triple intersection number of the two compact surfaces $S_2$ : $\delta_2 = 0$, $S_3 : \delta_3 = 0$, following the algorithm in section 2.2.2:

$$
\begin{array}{c}
\boxed{S_2(-24)\;\boxed{-12}} \!\!-\!\! \boxed{18\;\;S_3(8)}
\end{array}
\tag{4.74}
$$

The topology of $S_2$ is a $\mathbb{P}^1$ fibered over the $g = 4$ curve $S_2 \cdot S_3$, and $S_3$ is a Hirzebruch surface. Nonetheless, we can blow up $S_2 \cdot S_3$ at four double points on $S_3$, and transform it to a rational configuration [16]:

$$
\begin{array}{c}
\boxed{S_2(8)\;\;\boxed{-4}} \!\!-\!\! \boxed{2\;\;S_3(4)}
\end{array}
\tag{4.75}
$$

The latter geometry gives rise to the 5d IR gauge theory $G_2 + 4\boldsymbol{F}$ (with equivalent descriptions $Sp(2) + 2\boldsymbol{AS} + 2\boldsymbol{F}$ and $SU(3)_5 + 4\boldsymbol{F}$), which indeed has $f = 5$ and $r = 2$. The flavor symmetry is $G_F = Sp(4) \times Sp(1)$.

Note that in the 6d description of $\Delta(54)$, whose resolved elliptic CY3 has three compact divisors (4.66). Now we decompactify the divisor $S_1$, following the philosophy of [13, 21], and the remaining compact divisors $S_2, S_3$ exactly correspond to the two compact divisors in this section. An additional $SU(2)$ flavor symmetry appears from 6d to 5d, because $S_2 S_1^2 = -2$ in (4.66), and $S_1 \cdot S_2$ becomes a new flavor curve that generates $SU(2)$ flavor symmetry factor.

## 4.4 $G_m$

In this section, we provide a class of examples without a straight-forward $T_X^{6d}$ interpretation, which is $\Gamma = G_m$. The $G_m$ orbifold is given by the hypersurface equation

$$wx(4x^m - y^2) + z^2 = 0. \tag{4.76}$$

The resolution of this singularity is qualitatively different for odd and even $m$. For even $m$, the codimension-two singular loci are all smooth, and the resolution procedure is the conventional hypersurface one. On the other hand, for odd $m$, the resolved equation becomes a complete intersection CY3 in the process.

### 4.4.1   $m = 2k$

For example, for $m = 2$, the equation

$$wx(4x^2 - y^2) + z^2 = 0 \tag{4.77}$$

can be resolved by

$$\left(z^{(2)}, x^{(1)}, y^{(1)}, w^{(1)}; \delta_1\right). \tag{4.78}$$

Then one can resolve the $D_4 \times A_1^3$ codimension-two singularities afterwards.

The compact exceptional divisor $S_1 : \delta_1 = 0$ is a generalized dP$_7$ of type $\mathbf{D}_4 + 3\mathbf{A}_1$. The 5d SCFT is the rank-1 $E_7$ theory.

For $m = 2k$ $(k \geq 2)$, the equation

$$wx(4x^{2k} - y^2) + z^2 = 0 \tag{4.79}$$

can be resolved by

$$
\begin{aligned}
&\left(z^{(2)}, x^{(1)}, y^{(1)}, w^{(1)}; \delta_1\right) \\
&\left(y, z, \delta_i; \delta_{i+1}\right) \quad (i = 1, \ldots, k - 1).
\end{aligned}
\tag{4.80}
$$

And then the $D_{2k+2} \times A_1^3$ codimension-two singularities can be resolved in the conventional way.

The triple intersection number among the compact exceptional divisors $S_i : \delta_i = 0$ are

$$\tag{4.81}$$

The corresponding 5d SCFT is the UV completion of IR gauge theory $Sp(k) + (2k + 4)\mathbf{F}$. The UV flavor symmetry is

$$G_F = SO(4k + 8) \times SU(2), , \tag{4.82}$$

which contains the flavor symmetry read off from codimension-two singularities as a subset.

### 4.4.2   $m = 2k + 1$

For $m = 3$, we first do the blow ups

$$
\begin{aligned}
&\left(z^{(2)}, x^{(1)}, y^{(1)}, w^{(1)}; \delta_1\right) , \ (x, z, w; \delta_2) , \ (x, y, z; \delta_3) , \ (y, z, \delta_3; \delta_4) \\
&(z, \delta_3; \delta_5) , \ (\delta_3, \delta_5; \delta_6) , \ (\delta_4, \delta_5; \delta_7)
\end{aligned}
\tag{4.83}
$$

and get

$$wx(4x^3 \delta_1 \delta_2^3 \delta_3 \delta_5 \delta_6^2 - y^2 \delta_4)\delta_3 + z^2 \delta_5 = 0 \tag{4.84}$$

Then we define

$$U = 4x^3 \delta_1 \delta_2^3 \delta_3 \delta_5 \delta_6^2 - y^2 \delta_4 \tag{4.85}$$

and blow up

$$(z, w, U; \delta_8) \,. \tag{4.86}$$

The resolved equation is

$$\begin{cases} wxU\delta_3 + z^2\delta_5 = 0 \\ U\delta_8 = 4x^3\delta_1\delta_2^3\delta_3\delta_5\delta_6^2 - y^2\delta_4 \,. \end{cases} \tag{4.87}$$

The curves on the single compact divisor $S_1 : \delta_1 = 0$ are

$$\begin{aligned} C_1 : & \quad \delta_1 = \delta_4 = 0 \\ C_2 : & \quad \delta_1 = \delta_5 = 0 \\ C_3 : & \quad \delta_1 = \delta_6 = 0 \\ C_4 : & \quad \delta_1 = \delta_7 = 0 \\ C_5 : & \quad \delta_1 = \delta_2 = 0 \\ C_6 : & \quad \delta_1 = \delta_8 = 0 \end{aligned} \tag{4.88}$$

Among these, $C_1$, $C_2$, $C_3$, $C_4$ come from the resolution of codimension-two $D_5$ singularity, and $C_1$ is the degenerate $\mathbb{P}^1$ curve with normal bundle $\mathcal{O}(-1) \oplus \mathcal{O}(-1)$. The other curves are all $(-2)$-curves, which form the Dynkin diagram of $A_3 \times A_1^2$. Hence the flavor symmetry factors from $C_i$ is $G_F = SU(4) \times SU(2)^2 \times U(1)$.

## 5 Future Directions

In this paper, we investigated various $\mathbb{C}^3/\Gamma$ orbifolds using 3d McKay correspondence and resolution techniques. Nonetheless, there are still infinitely many cases to be studied in details. We conclude the paper by listing a number of future questions.

- In the version of McKay correspondence in [96], the $\Gamma$-Hilbert scheme corresponds to a particular resolution $\widetilde{X}$ of $\mathbb{C}^3/\Gamma$. The detail of this correspondence is unknown except for the abelian $\Gamma$ [55, 96] and a few non-abelian cases [102, 103]. It would be interesting to read off more details of $\widetilde{X}$ and the 5d SCFT $T_X^{5d}$ from the group theoretic data of $\Gamma$.

- Resolving $\mathbb{C}^3/\Gamma$ for the other cases of $\Gamma$ mentioned in section 3, which are not explicitly studied in this paper. Especially, more advanced techniques are needed to compute triple intersection numbers in the cases involving polynomial blow-ups, shifting variables and singular divisors. There are also additional cases in the class (B), (C) and (D) that cannot be written as a hypersurface in $\mathbb{C}^4$ or a complete intersection in $\mathbb{C}^5$, which should be investigated in the future. In particular, they include interesting cases with a non-trivial 1-form symmetry, which is mentioned at the end of section 2.4.

- Understand the physics associated to compact divisors with log-canonical singularities, which often appears in not only the cases of $X = \mathbb{C}^3/\Gamma$, but also the isolated

hypersurface singularities [6, 7]. In the 6d descriptions of $\Delta(48)$ and $H_{168}$, these singular divisors potentially give rise to new 6d (1,0) SCFTs. The details of these (1,0) SCFTs should be elucidated in a future work.

- Study the Higgs branch of $T_X^{5d}$, using either brane web or geometric techniques. For the cases of $\Gamma = \Delta(3n^2)$, this is worked out in [100]. It would be interesting to study the other classes of $SU(3)$ subgroups as well. A more difficult question is to study the deformation of $X$, which is a non-isolated singularity.

**Acknowledgements**

We thank Bobby Acharya, Jim Halverson, Max Hubner, Horia Magureanu, David Morrison, Sakura Schafer-Nameki, Benjamin Sung and Hao Y. Zhang for helpful discussions. JT would like to thank Ying Zhang for her love and support. The work of JT is supported by a grant from the Simons Foundation (#488569, Bobby Acharya). YW is supported by National Science Foundation of China under Grant No. 12175004, by Peking University under startup Grant No. 7100603534 and by the ERC Consolidator Grant number 682608 "Higgs bundles: Supersymmetric Gauge Theories and Geometry (HIGGSBNDL)".

# A  The abelian subgroups of the exceptional finite subgroups of $SU(3)$

In this appendix we summarize the abelian subgroups $G^A$ of the exceptional finite subgroups $G \subset SU(3)$, i.e., $E^{(n)}$ and $H_{36}, H_{60}, H_{72}, H_{168}, H_{216}, H_{360}$. We will represent type of an abelian subgroup $G^A \subset G$ by the conjugacy classes of its elements. We will underline the conjugacy class if age($g$) = 2 for its representative $g \in G^A$ and overline the conjugacy class which has an age($g$) = 2 dual. For example if

$$G^A = \{1, a_1, a_2, \cdots, a_n\} \tag{A.1}$$

then we represent $G^A$ by

$$r(G^A) = \{c(1), \underline{c(a_1)}, \overline{c(a_2)}, \cdots, c_{a_n}\} \tag{A.2}$$

where $c(\cdot)$ maps a group element to the conjugacy class it belongs to and in this case age($a_1$) = 2 and $a_2$ has an age($g$) = 2 dual. Also we will adopt the notation that when $G^A \cong \mathbb{Z}_{m+1} \times \mathbb{Z}_{n+1}$ we represent it by

$$r(G^A) = \{c(a_{00}), c(a_{01}), \cdots, c(a_{0m}), \tag{A.3}$$

$$\cdots \tag{A.4}$$

$$c(a_{n0}), c(a_{n1}), \cdots, c(a_{nm})\} \tag{A.5}$$

where $a_{01}$ is the generator of $\mathbb{Z}_{m+1}$ and $a_{10}$ is the generator of $\mathbb{Z}_{n+1}$ and $a_{ij} = a_{10}^i \cdot a_{01}^j$.

Note that two different abelian subgroups $G_1^A \subset G$ and $G_2^A \subset G$ can have the same types, i.e., $r(G_1^A) = r(G_2^A)$. Physically, different abelian subgroups of $G$ that have the same $r(G^A)$ lead to identical information on the Coulomb branch. Therefore, for our purpose,

we will only list the abelian subgroups $G^A$ that have distinct $r(G^A)$. In our notation the zeroth conjugacy class is always the class of $\mathrm{Id}_{3\times 3}$.

We also write down the maximal $\mathbb{Z}_n$ subgroups whose non-identity elements all belong to $\Gamma_{1,f}$. These subgroups are further glued together into a set of Dynkin diagram of ADE type, which exactly matches the ADE type of codimension-two singularities in $\mathbb{C}^3/\Gamma$.

## A.1 Sporadic subgroups of $U(2)$

### A.1.1 $E^{(1)}$

There are in total 21 conjugacy classes and 7 abelian subgroups. There are 4 of them that are $\mathbb{Z}_6 \times \mathbb{Z}_3$ whose types are

$$
\begin{aligned}
r(G^A) = \{0, \underline{12}, 17, 2, 11, \overline{20}, \\
4, \underline{13}, \overline{16}, 5, 10, \underline{15}, \\
3, \overline{8}, 18, 6, \underline{9}, \overline{19}\}
\end{aligned}
\tag{A.6}
$$

and there are 3 of them that are $\mathbb{Z}_4 \times \mathbb{Z}_3$ whose types are

$$
\begin{aligned}
r(G^A) = \{0, 1, 2, 1, \\
\underline{9}, \underline{7}, \overline{8}, \underline{7}, \\
\overline{16}, \overline{14}, \underline{15}, \overline{14}\}.
\end{aligned}
\tag{A.7}
$$

The maximal $\mathbb{Z}_N$ subgroups

$$
\begin{aligned}
&\{0, 5, 3, 2, 4, 6\}\,, \\
&\{0, 1, 2, 1\}\,, \\
&\{0, 17, 11\}\,, \\
&\{0, 18, 10\}
\end{aligned}
\tag{A.8}
$$

are glued into the Dynkin diagram of $E_6 \times A_2 \times A_2$:

$$\tag{A.9}$$

This matches the flavor rank $f = 10$.

From the equation of $E^{(1)}$,

$$
z^3 - w(y^2 + 108x^4) = 0\,.
\tag{A.10}
$$

one can see two $A_2$ singularities along $w = y \pm 6i\sqrt{3}x^2 = z = 0$ respectively and an $E_6$ singularity along $x = y = z = 0$.

### A.1.2  $E^{(2)}$

There are in total 7 conjugacy classes and 7 abelian subgroups. There are 4 of them that are $\mathbb{Z}_3 \times \mathbb{Z}_2$ whose types are

$$r(G^A) = \{0, 3, 4, \\ 2, \overline{6}, \underline{5}\} \tag{A.11}$$

and there are 3 of them that are $\mathbb{Z}_4$ whose types are

$$r(G^A) = \{0, 1, 2, 1\}. \tag{A.12}$$

The maximal $\mathbb{Z}_N$ subgroups

$$\{0, 1, 2, 1\} \ , \\ \{0, 3, 4\} \tag{A.13}$$

are glued into the Dynkin diagram of $D_4 \times A_2$:

$$\tag{A.14}$$

This matches the flavor rank $f = 4$. Note that the $D_4$ singularity only contributes flavor rank $f = 2$, due to monodromy reduction. In fact, the binary dihedral group $\mathbb{D}_4$ is a subgroup of $E^{(2)}$, which also confirms that there is a $D_4$ factor instead of $A_3$.

From the singularity equation

$$z^3 - w(y^3 + 12\sqrt{3}ix^2) = 0 \,, \tag{A.15}$$

one can also check that there is a $D_4$ singularity along $x = y = z = 0$ and an $A_2$ singularity along $w = z = 12\sqrt{3}ix^2 + y^3 = 0$.

### A.1.3  $E^{(3)}$

There are in total 16 conjugacy classes and 13 abelian subgroups. There are 3 of them that are $\mathbb{Z}_8 \times \mathbb{Z}_2$ whose types are

$$r(G^A) = \{0, 15, 9, 14, 8, 14, 9, 15, \\ 11, \underline{6}, \underline{4}, \underline{6}, 11, \overline{7}, \overline{5}, \overline{7}\}, \tag{A.16}$$

there are 4 of them that are $\mathbb{Z}_4 \times \mathbb{Z}_3$ whose types are

$$r(G^A) = \{0, \underline{4}, 8, \overline{5}, \\ 12, \overline{3}, 13, \underline{2}, \\ 12, \overline{3}, 13, \underline{2}\} \tag{A.17}$$

and there are 6 of them that are $\mathbb{Z}_4 \times \mathbb{Z}_2$ whose types are

$$r(G^A) = \{0, \underline{4}, 8, \overline{5}, \\ 1, 10, 1, 10\}. \tag{A.18}$$

The maximal $\mathbb{Z}_N$ subgroups

$$\{0, 15, 9, 14, 8, 14, 9, 15\}\,,$$
$$\{0, 11\}\,,$$
$$\{0, 13, 12, 8, 12, 13\}\,, \tag{A.19}$$
$$\{0, 10, 8, 10\}\,,$$
$$\{0, 1\}$$

are glued into the Dynkin diagram of $E_7 \times A_1 \times A_1$:

$$\tag{A.20}$$

This matches the flavor rank $f = 9$.

From the singular equation

$$z^2 + w(108x^3 - xy^3) = 0\,, \tag{A.21}$$

there is an $E_7$ singularity along $x = y = z = 0$, an $A_1$ singularity along $w = x = z = 0$ and another $A_1$ singularity along $w = z = 108x^2 - y^3 = 0$.

### A.1.4 $E^{(4)}$

There are in total 8 conjugacy classes and 13 abelian subgroups. There are 3 of them that are $\mathbb{Z}_8$ whose types are

$$r(G^A) = \{0, \overline{7}, 1, \overline{7}, 3, \underline{6}, 1, \underline{6}\}, \tag{A.22}$$

there are 4 of them that are $\mathbb{Z}_3 \times \mathbb{Z}_2$ whose types are

$$r(G^A) = \{0, 4, 4, \\ 3, 5, 5\}. \tag{A.23}$$

and there are 6 of them that are $\mathbb{Z}_2 \times \mathbb{Z}_2$ whose types are

$$r(G^A) = \{0, 3, \\ 2, 2\}. \tag{A.24}$$

The maximal $\mathbb{Z}_N$ subgroups

$$\{0, 1, 3, 1\}\,,$$
$$\{0, 5, 4, 3, 4, 5\}\,, \tag{A.25}$$
$$\{0, 2\}$$

are glued into the Dynkin diagram of $E_6 \times A_1$:

$$\tag{A.26}$$

The number of conjugacy classes in $\Gamma_{1,f}$ matches the flavor rank $f = 5$. Note that the $E_6$ singularity only contributes $f = 4$. One can confirm that there is a binary tetrahedral subgroup $\mathbb{E}_6$ in $E^{(4)}$, instead of $\mathbb{D}_5$.

From the singular equation

$$z^2 + w(108x^4 - y^3) = 0\,, \tag{A.27}$$

there is an $E_6$ singularity along $x = y = z = 0$ and an $A_1$ singularity along $w = z = 108x^4 - y^3 = 0$.

### A.1.5 $E^{(5)}$

There are in total 16 conjugacy classes and 13 abelian subgroups. There are 3 of them that are $\mathbb{Z}_4 \times \mathbb{Z}_4$ whose types are

$$\begin{aligned}
r(G^A) = \{0, 8, 1, 9, \\
\overline{14}, 9, 4, \overline{3}, \\
7, \overline{3}, 1, \underline{2}, \\
\underline{13}, \underline{2}, 4, 8\},
\end{aligned} \tag{A.28}$$

there are 4 of them that are $\mathbb{Z}_4 \times \mathbb{Z}_3$ whose types are

$$\begin{aligned}
r(G^A) = \{0, \underline{13}, 7, \overline{14}, \\
11, \overline{12}, 15, \underline{10}, \\
11, \overline{12}, 15, \underline{10}\}
\end{aligned} \tag{A.29}$$

and there are 6 of them that are $\mathbb{Z} \times \mathbb{Z}$ whose types are

$$\begin{aligned}
r(G^A) = \{0, \overline{14}, 7, \underline{13}, \\
\overline{5}, \underline{6}, \overline{5}, \underline{6}\}.
\end{aligned} \tag{A.30}$$

The maximal $\mathbb{Z}_N$ subgroups

$$\begin{aligned}
\{0, 8, 1, 9\}\,, \\
\{0, 4, 7, 4\}\,, \\
\{0, 15, 11, 7, 11, 15\}
\end{aligned} \tag{A.31}$$

are glued into the Dynkin diagram of $E_6 \times A_3$:

$$\tag{A.32}$$

The number of conjugacy classes in $\Gamma_{1,f}$ matches the flavor rank $f = 7$. Note that the $E_6$ singularity only contributes $f = 4$, due to monodromy reduction. One can confirm that there is a binary tetrahedral subgroup $\mathbb{E}_6$ in $E^{(5)}$, instead of $\mathbb{D}_5$.

From the singular equation

$$108z^4 + w(y^2 - x^3) = 0\,, \tag{A.33}$$

there is an $E_6$ singularity along $x = y = z = 0$ and an $A_3$ singularity along $w = z = y^2 - x^3 = 0$.

### A.1.6 $E^{(6)}$

There are in total 14 conjugacy classes and 7 abelian subgroups. There are 4 of them that are $\mathbb{Z}_4 \times \mathbb{Z}_3$ whose types are

$$
\begin{aligned}
r(G^A) = \{0, \underline{10}, 3, \overline{11}, \\
5, \overline{9}, 13, \underline{4}, \\
7, \overline{8}, 12, \underline{6}\}
\end{aligned}
\tag{A.34}
$$

and there are 3 of them that are $\mathbb{Z}_4 \times \mathbb{Z}_2$ whose types are

$$
\begin{aligned}
r(G^A) = \{0, 2, 3, 2, \\
1, \overline{11}, 1, \underline{10}\}.
\end{aligned}
\tag{A.35}
$$

The maximal $\mathbb{Z}_n$ subgroups

$$
\begin{aligned}
&\{0, 12, 5, 3, 7, 13\} \\
&\{0, 2, 3, 2\} \\
&\{0, 1\}
\end{aligned}
\tag{A.36}
$$

are glued into the Dynkin diagram of $E_6 \times A_1$:

$$
\tag{A.37}
$$

This matches the flavor rank $f = 7$.

The singular equation is a complete intersection

$$
\begin{cases}
y^3 - z^2 = 108x^2 \\
u^2 - vx = 0.
\end{cases}
\tag{A.38}
$$

We impose $v \neq 0$ first and plug $x = u^2/v$ into the first equation. We see that on the $v \neq 0$ patch there is an $E_6$ singularity along $u = x = y = z = 0$. Clearly there is also an $A_1$ singularity along $u = v = x = z^2 - y^3 = 0$.

### A.1.7 $E^{(7)}$

There are in total 24 conjugacy classes and 13 abelian subgroups. There are 3 of them that are $\mathbb{Z}_8 \times \mathbb{Z}_3$ whose types are

$$
\begin{aligned}
r(G^A) = \{0, 14, 1, 15, 7, 15, 1, 14, \\
11, \underline{21}, \overline{2}, \overline{22}, \underline{8}, \overline{22}, \overline{2}, \underline{21}, \\
10, \overline{23}, \underline{3}, \underline{20}, \overline{9}, 20, \underline{3}, \overline{23}\},
\end{aligned}
\tag{A.39}
$$

there are 4 of them that are $\mathbb{Z}_6 \times \mathbb{Z}_3$ whose types are

$$
\begin{aligned}
r(G^A) = \{0, \underline{8}, \underline{10}, 7, \overline{11}, \overline{9}, \\
16, \underline{18}, 12, \overline{19}, 17, 13, \\
17, 13, 16, \underline{18}, 12, \overline{19}\}
\end{aligned}
\tag{A.40}
$$

and there are 6 of the that are $\mathbb{Z}_4 \times \mathbb{Z}_3$ whose type are

$$
\begin{aligned}
r(G^A) = \{0, 4, 7, 4, \\
\overline{11}, \overline{6}, \underline{8}, \overline{6} \\
\underline{10}, \underline{5}, \overline{9}, \underline{5}\}.
\end{aligned}
$$

(A.41)

The maximal $\mathbb{Z}_n$ subgroups

$$
\begin{aligned}
\{0, 14, 1, 15, 7, 15, 1, 14\} \\
\{0, 13, 12, 7, 12, 13\} \\
\{0, 4, 7, 4\} \\
\{0, 16, 17\}
\end{aligned}
$$

(A.42)

are glued into the Dynkin diagram of $E_7 \times A_2$:

(A.43)

This matches the flavor rank $f = 9$.

The singular equation is

$$
\begin{cases}
yv - u^3 = 0 \\
z^2 + 108x^3 = xy \, .
\end{cases}
$$

(A.44)

First we require $v \neq 0$ and plug $y = u^3/v$ into the second equation. We see that on the $v \neq 0$ patch there is an $E_7$ singularity along $u = x = y = z = 0$. Clearly there is also an $A_2$ singularity along $u = v = y = z^2 + 108x^3 = 0$.

## A.1.8 $E^{(8)}$

There are in total 32 conjugacy classes and 13 abelian subgroups. There are 3 of them that are $\mathbb{Z}_8 \times \mathbb{Z}_4$ whose types are

$$
\begin{aligned}
r(G^A) = \{0, \overline{23}, \overline{5}, \overline{31}, 8, \underline{20}, \underline{4}, \underline{30}, \\
19, \overline{7}, \overline{24}, 14, \underline{16}, 6, 27, 15, \\
11, \overline{26}, 9, \underline{18}, 11, \overline{26}, 9, \underline{18}, \\
27, 15, 19, \overline{7}, \overline{24}, 14, \underline{16}, \underline{6}\},
\end{aligned}
$$

(A.45)

there are of 4 them that are $\mathbb{Z}_8 \times \mathbb{Z}_3$ whose types are

$$
\begin{aligned}
r(G^A) = \{0, \overline{23}, \overline{5}, \overline{31}, 8, \underline{20}, \underline{4}, \underline{30}, \\
12, \underline{21}, \underline{2}, \overline{28}, 13, \underline{22}, \overline{3}, \overline{29}, \\
12, \underline{21}, \underline{2}, \overline{28}, 13, \underline{22}, \overline{3}, \overline{29}\}
\end{aligned}
$$

(A.46)

and there are 6 of them that are $\mathbb{Z}_8 \times \mathbb{Z}_2$ whose types are

$$
\begin{aligned}
r(G^A) = \{0, \overline{23}, \overline{5}, \overline{31}, 8, \underline{20}, \underline{4}, \underline{30}, \\
1, \overline{25}, 10, \underline{17}, 1, \overline{25}, 10, \underline{17}\}.
\end{aligned}
$$

(A.47)

The maximal $\mathbb{Z}_N$ subgroups

$$\begin{aligned}
&\{0, 14, 9, 15, 8, 15, 9, 14\}\ , \\
&\{0, 13, 12, 8, 12, 13\}\ , \\
&\{0, 10, 8, 10\}\ , \\
&\{19, 11, 27\}\ , \\
&\{0, 1\}
\end{aligned}$$

(A.48)

are glued into the Dynkin diagram of $E_7 \times A_3 \times A_1$:

$$\text{(A.49)}$$

This matches the flavor rank $f = 11$.

The singular equation is

$$\begin{cases}
z^2 + 108ux - uy^3 = 0 \\
u^2 - vx = 0
\end{cases}$$

(A.50)

In the patch $v \neq 0$, we plug $x = u^2/v$ into the first equation, and see an $E_7$ singularity at $z = u = y = x = 0$. In the patch $108x - y^3 \neq 0$, we plug $u = z^2/(108x - y^3)$ into the second equation, and see an $A_3$ singularity at $v = x = z = u = 0$. Finally there is also an $A_1$ singularity at $z = u = 108x - y^3 = v = 0$.

### A.1.9 $E^{(9)}$

There are in total 18 conjugacy classes and 31 abelian subgroups. There are 6 of them that are $\mathbb{Z}_5 \times \mathbb{Z}_4$ whose types are

$$\begin{aligned}
r(G^A) = \{&0, 15, 16, 16, 15, \\
&\overline{2}, \underline{6}, \overline{7}, \overline{7}, \underline{6}, \\
&9, 14, 12, 12, 14, \\
&\underline{1}, \overline{5}, \underline{3}, \underline{3}, \overline{5}\},
\end{aligned}$$

(A.51)

there are 10 of them that are $\mathbb{Z}_4 \times \mathbb{Z}_3$ whose types are

$$\begin{aligned}
r(G^A) = \{&0, \underline{1}, 9, \overline{2}, \\
&17, \overline{4}, 13, \underline{8}, \\
&17, \overline{4}, 13, \underline{8}\}
\end{aligned}$$

(A.52)

and there are 15 of them that are $\mathbb{Z}_4 \times \mathbb{Z}_2$ whose types are

$$\begin{aligned}
r(G^A) = \{&0, \underline{1}, 9, \overline{2}, \\
&11, 10, 11, 10\}.
\end{aligned}$$

(A.53)

The maximal $\mathbb{Z}_N$ subgroups

$$\{0, 14, 16, 12, 15, 9, 15, 12, 16, 14\}\,,$$
$$\{0, 13, 17, 9, 17, 13\}\,,$$
$$\{0, 10, 9, 10\}\,,$$
$$\{0, 11\}$$

(A.54)

are glued into the Dynkin diagram of $E_8 \times A_1$:

(A.55)

This matches the flavor rank $f = 9$.

The singular equation is

$$z^2 + w(-1728x^5 + y^3) = 0\,.$$

(A.56)

There is an $E_8$ singularity along $x = y = z = 0$ and an $A_1$ singularity along $w = z = y^3 - 1728x^5 = 0$.

## A.1.10    $E^{(10)}$

There are in total 27 conjugacy classes and 31 abelian subgroups. There are 6 of them that are $\mathbb{Z}_6 \times \mathbb{Z}_5$ whose types are

$$\begin{aligned} r(G^A) = \{&0, \overline{10}, \overline{20}, 2, \underline{11}, \underline{19},\\ &6, \underline{12}, \underline{24}, 5, \overline{16}, \overline{23},\\ &7, \underline{14}, \overline{25}, 3, \underline{15}, \overline{21},\\ &7, \underline{14}, \overline{25}, 3, \underline{15}, \overline{21},\\ &6, \underline{12}, \underline{24}, 5, \overline{16}, \overline{23}\},\end{aligned}$$

(A.57)

there are 10 of them that are $\mathbb{Z}_6 \times \mathbb{Z}_3$ whose types are

$$\begin{aligned} r(G^A) = \{&0, \overline{10}, \overline{20}, 2, \underline{11}, 19,\\ &26, 4, 17, \overline{22}, 8, \underline{13},\\ &17, \overline{22}, 8, \underline{13}, 26, 4\}\end{aligned}$$

(A.58)

and there are 15 of them that are $\mathbb{Z}_4 \times \mathbb{Z}_3$ whose types are

$$\begin{aligned} r(G^A) = \{&0, 1, 2, 1,\\ &\underline{11}, \underline{9}, \overline{10}, \underline{9},\\ &\overline{20}, \overline{18}, \underline{19}, \overline{18}\}.\end{aligned}$$

(A.59)

The maximal $\mathbb{Z}_N$ subgroups

$$\{0, 5, 7, 3, 6, 2, 6, 3, 7, 5\}\,,$$
$$\{0, 4, 8, 2, 8, 4\}\,,$$
$$\{0, 1, 2, 1\}\,,$$
$$\{0, 26, 17\}$$

(A.60)

are glued into the Dynkin diagram of $E_8 \times A_2$:

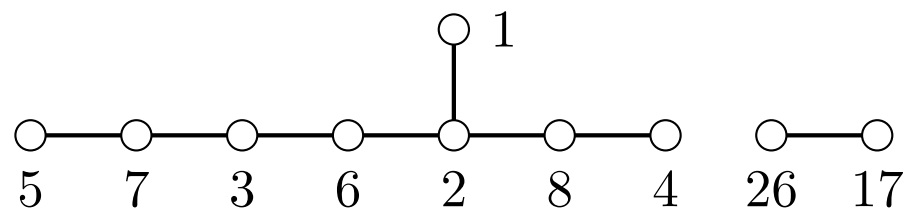

(A.61)

This matches the flavor rank $f = 10$.

The singular equation is

$$z^3 + w(-1728x^5 + y^2) = 0\,. \tag{A.62}$$

There is an $E_8$ singularity along $x = y = z = 0$ and an $A_2$ singularity along $w = z = y^2 - 1728x^5 = 0$.

### A.1.11 $E^{(11)}$

There are in total 45 conjugacy classes and 31 abelian subgroups. There are 6 of them that are $\mathbb{Z}_{10} \times \mathbb{Z}_5$ whose types are

$$\begin{aligned} r(G^A) = \{&0, \underline{29}, \underline{21}, \underline{18}, \overline{34}, 1, \underline{30}, \overline{22}, \overline{17}, \overline{33}, \\ &28, \underline{9}, 26, \underline{20}, \underline{21}, \overline{27}, \overline{8}, 25, 19, \overline{22}, \\ &19, \underline{31}, \underline{2}, \overline{33}, 15, \underline{20}, 32, \overline{3}, \overline{34}, 16, \\ &23, 16, \underline{30}, \underline{7}, 28, \overline{24}, 15, \underline{29}, \overline{6}, \overline{27}, \\ &32, \underline{18}, 23, 25, \underline{4}, \underline{31}, \overline{17}, \overline{24}, 26, \overline{5}\}, \end{aligned} \tag{A.63}$$

there are 10 of them that are $\mathbb{Z}_6 \times \mathbb{Z}_5$ whose types are

$$\begin{aligned} r(G^A) = \{&0, 35, 36, 1, 36, 35, \\ &\underline{2}, \underline{37}, \overline{38}, \overline{3}, \overline{38}, \underline{37}, \\ &\underline{4}, \underline{39}, \underline{40}, \overline{5}, \underline{40}, \underline{39}, \\ &\overline{8}, \overline{43}, \overline{44}, \underline{9}, \overline{44}, \overline{43}, \\ &\overline{6}, \overline{41}, \underline{42}, \underline{7}, \underline{42}, \overline{41}\} \end{aligned} \tag{A.64}$$

and there are 15 of them that are $\mathbb{Z}_5 \times \mathbb{Z}_4$ whose types are

$$\begin{aligned} r(G^A) = \{&0, \underline{2}, \underline{4}, \overline{8}, \overline{6}, \\ &10, \overline{11}, \underline{12}, \overline{14}, \underline{13}, \\ &1, \overline{3}, \overline{5}, \underline{9}, \underline{7}, \\ &10, \overline{11}, \underline{12}, \overline{14}, \underline{13}\}. \end{aligned} \tag{A.65}$$

The maximal $\mathbb{Z}_N$ subgroups

$$\begin{aligned} &\{0, 25, 15, 16, 26, 1, 26, 16, 15, 25\}\ , \\ &\{0, 35, 36, 1, 36, 35\}\ , \\ &\{0, 10, 1, 10\}\ , \\ &\{0, 28, 19, 23, 32\} \end{aligned} \tag{A.66}$$

are glued into the Dynkin diagram of $E_8 \times A_4$:

$$(\text{A.67})$$

This matches the flavor rank $f = 12$.

The singular equation is

$$1728 z^5 - w(x^3 + y^2) = 0 \,. \tag{A.68}$$

There is an $E_8$ singularity along $x = y = z = 0$ and an $A_4$ singularity along $w = z = y^2 + x^3 = 0$.

## A.2  $H_{36}$

There are in total 14 conjugacy classes. The representatives of each non-trivial conjugacy classes are

$$M_1 = \begin{pmatrix} 0 & -1 & 0 \\ -w & 0 & 0 \\ 0 & 0 & -w^2 \end{pmatrix}, \quad M_2 = \begin{pmatrix} 0 & -1 & 0 \\ -w^2 & 0 & 0 \\ 0 & 0 & -w \end{pmatrix}, \quad M_3 = \begin{pmatrix} 0 & 0 & 1 \\ 1 & 0 & 0 \\ 0 & 1 & 0 \end{pmatrix},$$

$$M_4 = \begin{pmatrix} 0 & 0 & 1 \\ -w^2 & 0 & 0 \\ 0 & -w & 0 \end{pmatrix}, \quad M_5 = \begin{pmatrix} -1 & 0 & 0 \\ 0 & 0 & -1 \\ 0 & -1 & 0 \end{pmatrix}, \quad M_6 = \frac{1}{3} \begin{pmatrix} -w+1 & -w+1 & -w+1 \\ -w+1 & -w-2 & 2w+1 \\ -w+1 & 2w+1 & -w-2 \end{pmatrix},$$

$$M_7 = \frac{1}{3} \begin{pmatrix} -w+1 & -w+1 & -w+1 \\ 2w+1 & -w+1 & -w-2 \\ -w-2 & -w+1 & 2w+1 \end{pmatrix}, \quad M_8 = \frac{1}{3} \begin{pmatrix} w+2 & w+2 & w+2 \\ w+2 & w-1 & -2w-1 \\ w+2 & -2w-1 & w-1 \end{pmatrix},$$

$$M_9 = \frac{1}{3} \begin{pmatrix} w+2 & w+2 & w+2 \\ -2w-1 & w+2 & w-1 \\ w-1 & w+2 & -2w-1 \end{pmatrix}, \quad M_{10} = \frac{1}{3} \begin{pmatrix} -2w-1 & w+2 & w-1 \\ w+2 & -2w-1 & w-1 \\ w-1 & w-1 & w-1 \end{pmatrix},$$

$$M_{11} = \begin{pmatrix} w^2 & 0 & 0 \\ 0 & w^2 & 0 \\ 0 & 0 & w^2 \end{pmatrix}, \quad M_{12} = \frac{1}{3} \begin{pmatrix} 2w+1 & w+2 & -w-2 \\ -w+1 & 2w+1 & -w-2 \\ -w-2 & -w-2 & -w-2 \end{pmatrix}, \quad M_{13} = \begin{pmatrix} w & 0 & 0 \\ 0 & w & 0 \\ 0 & 0 & w \end{pmatrix}$$

where $w = e^{2\pi i/3}$.

There are in total 13 abelian subgroups. There are 9 of them that are $\mathbb{Z}_4 \times \mathbb{Z}_3$ whose types are:

$$r(G^A) = \{0, 9, 5, 7,$$
$$\overline{13}, \overline{8}, \underline{2}, \overline{12}, \tag{A.69}$$
$$\underline{11}, \underline{10}, \overline{1}, \underline{6}\},$$

there are 2 of them that are $\mathbb{Z}_3 \times \mathbb{Z}_3$ whose types are

$$r(G^A) = \{0, 3, 3$$
$$\overline{13}, 3, 3, \tag{A.70}$$
$$\underline{11}, 3, 3\}$$

and there are 2 of them that are $\mathbb{Z}_3 \times \mathbb{Z}_3$ whose types are

$$r(G^A) = \{0, 4, 4$$
$$\overline{13}, 4, 4, \tag{A.71}$$
$$\underline{11}, 4, 4\}.$$

The maximal $\mathbb{Z}_N$ subgroups

$$\{0, 9, 5, 7\} \,,$$
$$\{0, 3, 3\} \,, \tag{A.72}$$
$$\{0, 4, 4\}$$

form the Dynkin diagram of $A_3 \times A_2 \times A_2$:

$$\tag{A.73}$$

The number of conjugacy classes in $\Gamma_{1,f}$ matches the flavor rank $f = 5$. Note that each of the two $A_2$ singularities only has contribution $f = 1$.

The singular equation is

$$\begin{cases} 3u^2v + 2u^3 - 36uy - 36vx - v^3 + 432z^2 = 0, \\ u^2x - u^2y - 12x^2 + 9y^2 = 0. \end{cases} \tag{A.74}$$

We can solve the first equation in Eq. A.74 for either $x$ or $y$ and plug into the second equation. Solving for $x$ then plugging into the second equation while requiring that $v \neq 0$ we will see there are a $A_3$ singularity along the locus

$$\{u = x = y = 432z^2 - v^3 = 0\} \subset \mathbb{C}^5, \tag{A.75}$$

an $A_2$ singularity along the locus

$$\{u + v = x = y = z = 0\} \subset \mathbb{C}^5 \tag{A.76}$$

and another $A_2$ singularity along the locus

$$\{u - v = x = y - \frac{u^2}{9} = z = 0\} \subset \mathbb{C}^5. \tag{A.77}$$

Solving for $y$ then plugging into the first equation while requiring that $u \neq 0$ we see an $A_2$ singularity along the locus

$$\{u + v = x = y = z = 0\} \subset \mathbb{C}^5 \tag{A.78}$$

and another $A_2$ singularity along the locus

$$\{u - v = x = y - \frac{u^2}{9} = z = 0\} \subset \mathbb{C}^5. \tag{A.79}$$

We see that the two $A_2$ singularities seen in the $u \neq 0$ patch are exactly the same $A_2$ singularities seen previously in the $v \neq 0$ patch. Therefore we conclude that the total codimension 2 ADE singularity is $A_3 \times A_2 \times A_2$. Due to monodromy reduction, the two $A_2$ factors are reduced to flavor rank 1.

## A.3  $H_{60}$

There are in total 5 conjugacy classes. The representatives of each non-trivial conjugacy classes are

$$M_1 = \frac{1}{2} \begin{pmatrix} -1 & w^3 - w^2 & -w^3 + w^2 + 1 \\ -w^3 + w^2 & -w^3 + w^2 + 1 & 1 \\ w^3 - w^2 - 1 & 1 & w^3 - v^2 \end{pmatrix},$$

$$M_2 = \frac{1}{2} \begin{pmatrix} 1 & w^3 - w^2 & -w^3 + w^2 + 1 \\ w^3 - w^2 & -w^3 + w^2 + 1 & 1 \\ w^3 - w^2 - 1 & -1 & -w^3 + w^2 \end{pmatrix},$$

$$M_3 = \frac{1}{2} \begin{pmatrix} 1 & w^3 - w^2 & -w^3 + w^2 + 1 \\ -w^3 + w^2 & w^3 - w^2 - 1 & -1 \\ -w^3 + w^2 + 1 & 1 & w^3 - w^2 \end{pmatrix},$$

$$M_4 = \begin{pmatrix} -1 & 0 & 0 \\ 0 & -1 & 0 \\ 0 & 0 & 1 \end{pmatrix}$$

where $w = e^{2\pi i/10}$.

There are in total 21 abelian subgroups. There are 6 of them that are $\mathbb{Z}_5$ whose types are

$$r(G^A) = \{0, 2, 3, 3, 2\}, \tag{A.80}$$

there are 5 of them that are $\mathbb{Z}_2 \times \mathbb{Z}_2$ whose types are

$$r(G^A) = \{0, 4, \\ 4, 4\} \tag{A.81}$$

and there are 10 of them that are $\mathbb{Z}_3$ whose types are

$$r(G^A) = \{0, 1, 1\}. \tag{A.82}$$

The maximal $\mathbb{Z}_N$ subgroups
$$\{0, 2, 3, 3, 2\} , \\ \{0, 1, 1\} , \\ \{0, 4\} \tag{A.83}$$

form the Dynkin diagram of $A_4 \times A_2 \times A_1$:

$$\begin{array}{ccccccc} \circ\!\!-\!\!\circ\!\!-\!\!\circ\!\!-\!\!\circ & & \circ\!\!-\!\!\circ & & \circ \\ 2 \quad 3 \quad 3 \quad 2 & & 1 \quad 1 & & 4 \end{array} \tag{A.84}$$

The number of conjugacy classes in $\Gamma_{1,f}$ matches the flavor rank $f = 4$. Note that the $A_4$ singularity only has contribution $f = 2$ and the $A_2$ singularity only has contribution $f = 1$.

The singular equation is

$$
\begin{aligned}
0 = {} & 10125w^2 - 13071240x^4y^2z + 446240256x^9y^2 - 408197440x^6y^3 + 174372625x^3y^4 \\
& + 25927020x^2yz^2 + 40449024x^7yz - 245088256x^{12}y - 17622576x^5z^2 - 13658112x^{10}z \\
& + 54329344x^{15} - 21994875xy^3z - 18907875y^5 - 2777895z^3.
\end{aligned}
$$

(A.85)

In this case we find an $A_4$ singularity along the locus

$$\{w = 35y - 32x^3 = 95z - 64x^5 = 0\} \subset \mathbb{C}^4,$$

(A.86)

an $A_1$ singularity along the locus

$$\{w = y - x^3 = z - x^5 = 0\} \subset \mathbb{C}^4$$

(A.87)

and an $A_2$ singularity along the locus

$$\{w = 63y - 64x^3 = 1539z - 1600x^5 = 0\} \subset \mathbb{C}^4.$$

(A.88)

Due to monodromy, the $A_4$ factor is reduced to flavor rank 2 and the $A_2$ factor is reduced to flavor rank 1.

## A.4 $H_{72}$

There are in total 16 conjugacy classes. The representatives of each conjugacy classes are

$$
M_1 = \begin{pmatrix} 0 & -1 & 0 \\ -w & 0 & 0 \\ 0 & 0 & -w^2 \end{pmatrix}, \quad
M_2 = \begin{pmatrix} 0 & -1 & 0 \\ -w^2 & 0 & 0 \\ 0 & 0 & -w \end{pmatrix}, \quad
M_3 = \begin{pmatrix} 0 & 0 & 1 \\ 1 & 0 & 0 \\ 0 & 1 & 0 \end{pmatrix},
$$

$$
M_4 = \begin{pmatrix} -1 & 0 & 0 \\ 0 & 0 & -1 \\ 0 & -1 & 0 \end{pmatrix}, \quad
M_5 = \frac{1}{3} \begin{pmatrix} -w+1 & -w+1 & -w+1 \\ -w+1 & -w-2 & 2w+1 \\ -w+1 & 2w+1 & -w-2 \end{pmatrix},
$$

$$
M_6 = \frac{1}{3} \begin{pmatrix} -w+1 & -w+1 & -w+1 \\ 2w+1 & -w+1 & -w-2 \\ -w-2 & -w+1 & 2w+1 \end{pmatrix}, \quad
M_7 = \begin{pmatrix} -w+1 & -w+1 & 2w+1 \\ -w+1 & -w-2 & -w-2 \\ -w-2 & -w+1 & -w-2 \end{pmatrix},
$$

$$
M_8 = \frac{1}{3} \begin{pmatrix} -w+1 & -w+1 & 2w+1 \\ 2w+1 & -w+1 & -w+1 \\ 2w+1 & -w-2 & 2w+1 \end{pmatrix}, \quad
M_9 = \frac{1}{3} \begin{pmatrix} -w+1 & -w+1 & -w-2 \\ -w+1 & -w-2 & -w+1 \\ 2w+1 & -w-2 & -w-2 \end{pmatrix},
$$

$$
M_{10} = \frac{1}{3} \begin{pmatrix} -w+1 & -w+1 & -w-2 \\ 2w+1 & -w+1 & 2w+1 \\ -w+1 & 2w+1 & 2w+1 \end{pmatrix}, \quad
M_{11} = \frac{1}{3} \begin{pmatrix} w+2 & w+2 & w+2 \\ w+2 & w-1 & -2w-1 \\ w+2 & -2w-1 & w-1 \end{pmatrix},
$$

$$
M_{12} = \frac{1}{3} \begin{pmatrix} w+2 & w+2 & -2w-1 \\ w+2 & w-1 & w-1 \\ w-1 & w+2 & w-1 \end{pmatrix}, \quad
M_{13} = \frac{1}{3} \begin{pmatrix} w+2 & w+2 & w-1 \\ w+2 & w-1 & w+2 \\ -2w-1 & w-1 & w-1 \end{pmatrix},
$$

$$
M_{14} = \begin{pmatrix} w^2 & 0 & 0 \\ 0 & w^2 & 0 \\ 0 & 0 & w^2 \end{pmatrix}, \quad
M_{15} = \begin{pmatrix} w & 0 & 0 \\ 0 & w & 0 \\ 0 & 0 & w \end{pmatrix}
$$

where $w = e^{2\pi i/3}$.

There are in total 31 abelian subgroups. There are 9 of them that are $\mathbb{Z}_4 \times \mathbb{Z}_3$ whose types are

$$r(G^A) = \{0, 8, 4, 8,$$
$$\overline{15}, \overline{13}, \underline{2}, \overline{13}, \tag{A.89}$$
$$\underline{14}, \underline{7}, \overline{1}, \underline{7}\},$$

there are 9 of them that are $\mathbb{Z}_4 \times \mathbb{Z}_3$ whose types are

$$r(G^A) = \{0, 6, 4, 6,$$
$$\overline{15}, \overline{11}, \underline{2}, \overline{11}, \tag{A.90}$$
$$\underline{14}, \underline{5}, \overline{1}, \underline{5}\},$$

there are 9 of them that are $\mathbb{Z}_4 \times \mathbb{Z}_3$ whose types are

$$r(G^A) = \{0, 10, 4, 10,$$
$$\overline{15}, \overline{12}, \underline{2}, \overline{12}, \tag{A.91}$$
$$\underline{14}, \underline{9}, \overline{1}, \underline{9}\},$$

and there are 4 of them that are $\mathbb{Z}_3 \times \mathbb{Z}_3$ whose types are

$$r(G^A) = \{0, \overline{15}, \underline{14},$$
$$3, 3, 3, \tag{A.92}$$
$$3, 3, 3\}.$$

The maximal $\mathbb{Z}_N$ subgroups

$$\{0, 8, 4, 8\},$$
$$\{0, 6, 4, 6\},$$
$$\{0, 10, 4, 10\}, \tag{A.93}$$
$$\{0, 3, 3\}$$

form the Dynkin diagram of $D_4 \times A_2$:

$$\tag{A.94}$$

The number of conjugacy classes in $\Gamma_{1,f}$ matches the flavor rank $f = 5$. Note that the $A_2$ singularity only has contribution $f = 1$.

The singular equation is

$$\left(-w^3 + 3wy + 432x^2\right)^2 - 4\left(3y^2 z - 3yz^2 + z^3\right) = 0. \tag{A.95}$$

There are an $A_2$ singularity along the locus

$$\{x = y - w^2 = z - w^2 = 0\} \subset \mathbb{C}^4 \tag{A.96}$$

and a $D_4$ singularity along the locus

$$\{432x^2 - w^3 = y = z = 0\} \subset \mathbb{C}^4. \tag{A.97}$$

Due to monodromy, the $A_2$ factor is reduced to flavor rank 1.

## A.5  $H_{168}$

There are in total 6 conjugacy classes. The representatives of each non-trivial conjugacy classes are

$$M_1 = \begin{pmatrix} 0 & 0 & 1 \\ 1 & 0 & 0 \\ 0 & 1 & 0 \end{pmatrix},$$

$$M_2 = \frac{1}{7} \begin{pmatrix} -w^5 - w^4 + 2w^2 - 2w + 2 & -3w^5 - 2w^4 - 4w^3 - 2w^2 - w & 3w^5 + w^4 + w^3 + 2w^2 - 1 \\ -w^5 + 2w^4 + 2w^3 - w^2 - 2 & 2w^4 - w^3 - 2w^2 - w + 2 & 4w^5 + 2w^4 + w^3 + w^2 + 2w + 4 \\ -w^5 + 3w^3 + w^2 + w + 3 & -2w^5 - 3w^4 - 3w^3 - 2w^2 - 4 & w^5 - w^4 + w^3 + 3w + 3 \end{pmatrix},$$

$$M_3 = \frac{1}{7} \begin{pmatrix} -2w^5 - 2w^4 - 3w^2 - 4w - 3 & -w^5 - w^4 + 2w^2 - 2w + 2 & 3w^5 + 3w^4 + w^2 - w + 1 \\ w^4 + w^3 - w^2 + 3w + 1 & -3w^4 - 2w^3 - 4w^2 - 2w - 3 & 2w^4 - w^3 - 2w^2 - w + 2 \\ w^5 - w^4 + w^3 + 3w + 3 & -3w^5 - 4w^4 - 3w^3 - 2w - 2 & 2w^5 - 2w^4 + 2w^3 - w - 1 \end{pmatrix},$$

$$M_4 = \begin{pmatrix} w^4 & 0 & 0 \\ 0 & w & 0 \\ 0 & 0 & w^2 \end{pmatrix},$$

$$M_5 = \frac{1}{7} \begin{pmatrix} 3w^5 + w^4 + w^3 + 3w^2 - 1 & -w^5 - w^4 + 2w^2 - 2w + 2 & -3w^5 - 2w^4 - 4w^3 - 2w^2 - 3w \\ 4w^5 + 2w^4 + 1w^3 + 1w^2 + 2w + 4 & -w^5 + 2w^4 + 2w^3 - w^2 - 2 & 2w^4 - 1w^3 - 2w^2 - w + 2 \\ w^5 - w^4 + w^3 + 3w + 3 & -w^5 + 3w^3 + w^2 + w + 3 & -2w^5 - 3w^4 - 3w^3 - 2w^2 - 4 \end{pmatrix}$$

where $w = e^{2\pi i/7}$.

There are in total 65 abelian subgroups. There are 8 of them that are $\mathbb{Z}_7$ whose types are

$$r(G^A) = \{0, \overline{4}, \overline{4}, \underline{3}, \overline{4}, \underline{3}, \underline{3}\}, \tag{A.98}$$

there are 28 of them that are $\mathbb{Z}_3$ whose types are

$$r(G^A) = \{0, 1, 1\}, \tag{A.99}$$

there are 8 of them that are $\mathbb{Z}_2 \times \mathbb{Z}_2$ whose types are

$$r(G^A) = \{0, 5, \\ 5, 5\}, \tag{A.100}$$

and there are 21 of them that are $\mathbb{Z}_4$ whose types are

$$r(G^A) = \{0, 2, 5, 2\}, \tag{A.101}$$

The maximal $\mathbb{Z}_N$ subgroups

$$\{0, 2, 5, 2\}, \\ \{0, 1, 1\} \tag{A.102}$$

form the Dynkin diagram of $A_3 \times A_2$:

$$\begin{array}{ccccc} \circ\!\!-\!\!\circ\!\!-\!\!\circ & & \circ\!\!-\!\!\circ \\ 2 \quad 5 \quad 2 & & 1 \quad 1 \end{array} \tag{A.103}$$

The number of conjugacy classes in $\Gamma_{1,f}$ matches the flavor rank $f = 3$. Note that the $A_3$ singularity only has contribution $f = 2$ and the $A_2$ singularity only has contribution $f = 1$.

The singular equation is

$$z^3 + 1728y^7 + 1008zy^4x - 88z^2yx^2 - 60032y^5x^3 + 1088zy^2x^4 + 22016y^3x^6$$
$$- 256zx^7 - 2048yx^9 - w^2 = 0 \,. \tag{A.104}$$

There are an $A_3$ singularity along the locus

$$\{w = y^2 + 4x^3 = z^2 + 20736x^7 = 0\} \subset \mathbb{C}^4 \tag{A.105}$$

and an $A_2$ singularity along the locus

$$\{w = 27y^2 + 4x^3 = 243z^2 + 256x^7 = 0\} \subset \mathbb{C}^4. \tag{A.106}$$

Due to monodromy reduction, the $A_3$ factor is reduced to flavor rank 2 and the $A_2$ factor is reduced to flavor rank 1.

## A.6 $H_{216}$

There are in total 24 conjugacy classes. The representatives of each non-trivial conjugacy classes are

$$M_1 = \begin{pmatrix} w^6 & 0 & 0 \\ 0 & w^6 & 0 \\ 0 & 0 & w^6 \end{pmatrix}, \quad M_2 = \begin{pmatrix} w^3 & 0 & 0 \\ 0 & w^3 & 0 \\ 0 & 0 & w^3 \end{pmatrix}, \quad M_3 = \begin{pmatrix} 1 & 0 & 0 \\ 0 & w^3 & 0 \\ 0 & 0 & w^6 \end{pmatrix},$$

$$M_4 = \begin{pmatrix} -1 & 0 & 0 \\ 0 & 0 & -1 \\ 0 & -1 & 0 \end{pmatrix}, \quad M_5 = \begin{pmatrix} -w^6 & 0 & 0 \\ 0 & 0 & -w^6 \\ 0 & -w^6 & 0 \end{pmatrix}, \quad M_6 = \begin{pmatrix} -w^3 & 0 & 0 \\ 0 & 0 & -w^3 \\ 0 & -w^3 & 0 \end{pmatrix},$$

$$M_7 = \frac{1}{3}\begin{pmatrix} -2w^3-1 & -2w^3-1 & -2w^3-1 \\ -2w^3-1 & w^3+2 & w^3-1 \\ -2w^3-1 & w^3-1 & w^3+2 \end{pmatrix}, \quad M_8 = \frac{1}{3}\begin{pmatrix} w^3-1 & w^3-1 & w^3-1 \\ w^3-1 & -2w^3-1 & w^3+2 \\ w^3-1 & w^3+2 & -2w^3-1 \end{pmatrix},$$

$$M_9 = \frac{1}{3}\begin{pmatrix} w^3+2 & w^3+2 & w^3+2 \\ w^3+2 & w^3-1 & -2w^3-1 \\ w^3+2 & -2w^3-1 & w^3-1 \end{pmatrix}, \quad M_{10} = \begin{pmatrix} w^2 & 0 & 0 \\ 0 & w^2 & 0 \\ 0 & 0 & w^5 \end{pmatrix}, \quad M_{11} = \begin{pmatrix} w^8 & 0 & 0 \\ 0 & w^8 & 0 \\ 0 & 0 & w^2 \end{pmatrix},$$

$$M_{12} = \begin{pmatrix} w^5 & 0 & 0 \\ 0 & w^5 & 0 \\ 0 & 0 & w^8 \end{pmatrix}, \quad M_{13} = \begin{pmatrix} 0 & 0 & w^2 \\ w^2 & 0 & 0 \\ 0 & w^5 & 0 \end{pmatrix}, \quad M_{14} = \begin{pmatrix} -w^2 & 0 & 0 \\ 0 & 0 & -w^2 \\ 0 & -w^5 & 0 \end{pmatrix},$$

$$M_{15} = \begin{pmatrix} -w^8 & 0 & 0 \\ 0 & 0 & -w^8 \\ 0 & -w^2 & 0 \end{pmatrix}, \quad M_{16} = \begin{pmatrix} -w^5 & 0 & 0 \\ 0 & 0 & -w^5 \\ 0 & -w^8 & 0 \end{pmatrix}, \quad M_{17} = \begin{pmatrix} w^4 & 0 & 0 \\ 0 & w^4 & 0 \\ 0 & 0 & w^1 \end{pmatrix},$$

$$M_{18} = \begin{pmatrix} w & 0 & 0 \\ 0 & w & 0 \\ 0 & 0 & w^7 \end{pmatrix}, \quad M_{19} = \begin{pmatrix} w^7 & 0 & 0 \\ 0 & w^7 & 0 \\ 0 & 0 & w^4 \end{pmatrix}, \quad M_{20} = \begin{pmatrix} 0 & 0 & w^4 \\ w^4 & 0 & 0 \\ 0 & w & 0 \end{pmatrix},$$

$$M_{21} = \begin{pmatrix} -w^4 & 0 & 0 \\ 0 & 0 & -w^4 \\ 0 & -w & 0 \end{pmatrix}, \quad M_{22} = \begin{pmatrix} -w & 0 & 0 \\ 0 & 0 & -w \\ 0 & -w^7 & 0 \end{pmatrix}, \quad M_{23} = \begin{pmatrix} -w^7 & 0 & 0 \\ 0 & 0 & -w^7 \\ 0 & -w^4 & 0 \end{pmatrix}.$$

where $w = e^{2\pi i/9}$.

There are in total 91 abelian subgroups. There are 4 of them that are $\mathbb{Z}_9 \times \mathbb{Z}_3$ whose types are

$$\begin{aligned}
r(G_A) = \{0, \overline{10}, \overline{17}, \underline{1}, \underline{11}, \overline{18}, \overline{2}, \underline{12}, \underline{19}, , \\
3, \underline{11}, \overline{17}, 3, \underline{12}, \overline{18}, 3, \overline{10}, \underline{19}, \\
3, \overline{10}, \underline{19}, 3, \underline{11}, \overline{18}, 3, \underline{12}, \overline{18}\},
\end{aligned} \tag{A.107}$$

there are 36 of them that are $\mathbb{Z}_9 \times \mathbb{Z}_2$ whose types are

$$\begin{aligned}
r(G^A) = \{0, \overline{17}, \underline{11}, \overline{2}, \underline{19}, \overline{10}, \underline{1}, \overline{18}, \underline{12}, \\
4, \underline{22}, \underline{14}, \underline{6}, \underline{21}, \overline{16}, \overline{5}, \overline{23}, \overline{15}\},
\end{aligned} \tag{A.108}$$

there are 27 of them that are $\mathbb{Z}_4 \times \mathbb{Z}_3$ whose types are

$$\begin{aligned}
r(G^A) = \{0, 7, 4, 7, \\
\overline{2}, \overline{9}, \underline{6}, \overline{9}, \\
\underline{1}, \underline{8}, \overline{5}, \underline{8},
\end{aligned} \tag{A.109}$$

and there are 24 of them that are $\mathbb{Z}_3 \times \mathbb{Z}_3$ whose types are

$$\begin{aligned}
r(G^A) = \{0, \underline{1}, \overline{2}, \\
13, 13, 13, \\
20, 20, 20\}.
\end{aligned} \tag{A.110}$$

The maximal $\mathbb{Z}_N$ subgroups

$$\begin{aligned}
\{0, 3, 3\}\,, \\
\{0, 7, 4, 7\}\,, \\
\{0, 13, 20\}
\end{aligned} \tag{A.111}$$

form the Dynkin diagram of $A_3 \times A_2 \times A_2$:

$$\underset{3\quad 3}{\circ\!\!-\!\!\circ} \quad \underset{7\quad 4\quad 7}{\circ\!\!-\!\!\circ\!\!-\!\!\circ} \quad \underset{13\quad 20}{\circ\!\!-\!\!\circ} \tag{A.112}$$

The number of conjugacy classes in $\Gamma_{1,f}$ matches the flavor rank $f = 5$. Note that the $A_3$ singularity only has contribution $f = 2$ and one of the $A_2$ singularity only has contribution $f = 1$.

The singular equation is

$$y^3 - \frac{1}{4}z\left(\left(432w^2 - z - 3y\right)^2 - 6912x^3\right) = 0. \tag{A.113}$$

There are an $A_2$ singularity along the locus

$$\{w = x = y - z = 0\} \subset \mathbb{C}^4 \tag{A.114}$$

and a second $A_2$ singularity along the locus

$$\{x^3 - 27w^4 = y = z = 0\} \subset \mathbb{C}^4. \tag{A.115}$$

There is another $D_4$ singularity along the locus

$$\{x = y = z - 432w^2 = 0\} \subset \mathbb{C}^4. \tag{A.116}$$

Therefore the total codimension 2 ADE singularity is $D_4 \times A_2^2$. Due to monodromy reduction, the $D_4$ is reduced to flavor rank 3 and the two $A_2$s are reduces to flavor rank 1.

## A.7 $H_{360}$

There are in total 17 conjugacy classes. The representatives of each non-trivial conjugacy classes are

$$M_1 = \begin{vmatrix} -w^5-1 & 0 & 0 \\ 0 & -w^5-1 & 0 \\ 0 & 0 & -w^5-1 \end{vmatrix},$$

$$M_2 = \begin{vmatrix} w^5 & 0 & 0 \\ 0 & w^5 & 0 \\ 0 & 0 & w^5 \end{vmatrix},$$

$$M_3 = \begin{vmatrix} -1 & 0 & 0 \\ 0 & 0 & -w^3 \\ 0 & w^7+w^2 & 0 \end{vmatrix},$$

$$M_4 = \begin{vmatrix} w^5+1 & 0 & 0 \\ 0 & 0 & w^7-w^5+w^4+w-1 \\ 0 & -w^7 & 0 \end{vmatrix},$$

$$M_5 = \begin{vmatrix} -w^5 & 0 & 0 \\ 0 & 0 & -w^7+w^5-w^4+w^3-w+1 \\ 0 & -w^2 & 0 \end{vmatrix},$$

$$M_6 = \frac{1}{5} \begin{vmatrix} -2w^7+2w^3-2w^2+1 & 2w^6+w^3+2 & w^7-2w^6-2w^3+w^2 \\ -4w^7+4w^3-4w^2+2 & -w^6-3w^3-1 & -3w^7+w^6+w^3-3w^2 \\ -4w^7+4w^3-4w^2+2 & -w^6+2w^3-1 & 2w^7+w^6+w^3+2w^2 \end{vmatrix},$$

$$M_7 = \frac{1}{5} \begin{vmatrix} -w^5+2w^4+2w-1 & \frac{3}{2}w^7-\frac{1}{2}w^5+\frac{1}{2}w^4+\frac{5}{2}w^2-\frac{1}{2} & w^7-\frac{3}{2}w^5+w^4-\frac{5}{2}w^3+\frac{3}{2}w-\frac{3}{2} \\ 6w^7-5w^6+3w^4-5w^3+5w^2+w-5 & -w^7+w^5-3w+1 & 3w^7-w^5+w^4-1 \\ -w^7+5w^6+w^5+2w+1 & -3w^7+2w^5-3w^4-2w+2 & w^7-2w^4+w \end{vmatrix},$$

$$M_8 = \frac{1}{5} \begin{vmatrix} -2w^5-w^4-w-2 & -\frac{5}{2}w^7+w^6+w^5+2w^3-\frac{3}{2}w^2-\frac{1}{2}w+\frac{3}{2} & -w^6+\frac{1}{2}w^4+\frac{1}{2}w^3-w^2+w-\frac{1}{2} \\ w^7-2w^6+2w^5+w^3-2w^2-w+1 & w^7-2w^5+2w^4-2 & -4w^7+w^6+w^5-2w^4+2w^3-4w^2-2w+4 \\ -6w^7+2w^6+5w^5-4w^4+4w^3-3w^2-3w+6 & -w^7-w^6+w^5-2w^4+3w^3-w^2-2w+3 & -w^7-w^5-w^4+w-1 \end{vmatrix},$$

$$M_9 = \frac{1}{5} \begin{vmatrix} w^7+2w^5+w^4-w^3+w^2+w-1 & -\frac{1}{2}w^7+\frac{1}{2}w^6+\frac{1}{2}w^5-2w^4-w^2-\frac{1}{2}w+\frac{3}{2} & -\frac{1}{2}w^6+w^5-w^4+\frac{1}{2}w^3+\frac{1}{2}w^2-\frac{5}{2}w+\frac{3}{2} \\ w^7+w^6-w^4+3w^2-2w+2 & -2w^7+2w^6+2w^5-2w^4+2w^3-w^2+2 & 4w^7-2w^5-2w^3+2w^2+w-3 \\ 3w^7-w^6-2w^5-4w^3+w^2+w-1 & -w^4-2w^3+2w^2-2w-1 & w^7-2w^6+w^5+w^4-w^3-w-1 \end{vmatrix},$$

$$M_{10} = \frac{1}{5} \begin{vmatrix} -w^7+w^3-w^2+3 & 3w^7-\frac{3}{2}w^6-\frac{3}{2}w^5+2w^4-2w^3+\frac{5}{2}w^2+w-3 & \frac{3}{2}w^6-w^5+\frac{1}{2}w^4-w^3+\frac{1}{2}w^2+\frac{3}{2}w-1 \\ -2w^7+w^6-2w^5+w^4-w^3-w^2+3w-3 & w^7-2w^6-2w^3+w^2 & -w^6+w^5+2w^4+2w^2+w-1 \\ 3w^7-w^6-3w^5+4w^4+2w^2+2w-5 & w^7+w^6-w^5+3w^4-w^3-w^2+4w-2 & 2w^6+w^3+2 \end{vmatrix},$$

$$M_{11} = \frac{1}{5} \begin{vmatrix} w^5-2w^4-2w+1 & -2w^7+2w^5-w+2 & 2w^7+w^4+2w \\ -2w^7-2w^5-2w^4+2w-2 & -w^7-3w^4-w & 2w^7+w^5-w^4+1 \\ 2w^7-4w^5+4w^4-4 & -2w^7+3w^5-2w^4-3w+3 & w^7-w^5-2w-1 \end{vmatrix},$$

$$M_{12} = \frac{1}{5} \begin{vmatrix} 2w^7-w^5+2w^4-2w^3+2w^2+2w-2 & w^6-2w^5-2w^2+w & -w^7-w^6-w^4+w^3+w^2-2w+1 \\ 2w^7-4w^6+2w^5+2w^4-2w^3-2w-2 & 3w^7-2w^6+3w^4-3w^3+2w^2+w-3 & w^7-w^6-w^5+w^4-w^3+3w^2-1 \\ -4w^7+4w^6+4w^5-4w^4+4w^3-2w^2+4 & 2w^7+w^6-3w^5+2w^4-2w^3+3w-2 & 2w^6+w^5+w^2+2w \end{vmatrix},$$

$$M_{13} = \frac{1}{5} \begin{vmatrix} -2w^7+2w^3-2w^2+1 & 2w^7-w^6+2w^2-2 & -w^7+w^6-w^3-w^2-1 \\ 4w^6+2w^3+4 & -2w^7+2w^6+3w^3-2w^2+3 & -3w^7+w^6+w^3-3w^2 \\ 2w^7-4w^6-4w^3+2w^2 & -w^6+2w^3-1 & -w^7-2w^6-2w^2+1 \end{vmatrix},$$

$$M_{14} = \frac{1}{5} \begin{vmatrix} -w^7+w^3-w^2-2 & -w^7+w^6+\frac{1}{2}w^5+w^4+\frac{1}{2}w^3+\frac{1}{2}w+\frac{3}{2} & \frac{3}{2}w^7-w^6-\frac{1}{2}w^5+\frac{3}{2}w^4-w^3+\frac{1}{2}w^2+2w-\frac{1}{2} \\ 2w^7+w^6+w^5+2w^4-w^3+4w^2+w & 2w^7-2w^6-3w^5+4w^4-2w^3+w^2+2w-3 & 2w^7-w^6+2w^2-2 \\ 4w^7-w^6-w^5+3w^4-5w^3+2w^2+4w-3 & -w^7+w^6-w^3-w^2-1 & -w^7+2w^6-2w^5+w^4+w^3+3w \end{vmatrix},$$

$$M_{15} = \frac{1}{5} \begin{vmatrix} 3w^5-w^4-w+3 & -\frac{3}{2}w^7+\frac{1}{2}w^6-w^5-\frac{3}{2}w^4+w^3-2w^2+\frac{1}{2} & -w^7-\frac{1}{2}w^6-\frac{1}{2}w^5-\frac{1}{2}w^4+\frac{3}{2}w^3-\frac{1}{2}w^2-2w+\frac{1}{2} \\ -5w^7+w^6-w^5-w^4+2w^3-4w^2+w+2 & -3w^7+2w^6+w^5-2w^4+4w^3-3w^2-2w+2 & 2w^7+w^4+2w \\ -w^6-2w^5+2w^4+3w^3-w^2 & 2w^7+w^4+2w & -2w^7-2w^6+w^5-2w^4+w^3-2w^2-2w \end{vmatrix},$$

$$M_{16} = \frac{1}{5} \begin{vmatrix} w^7-3w^5+w^4-w^3+w^2+w-1 & \frac{5}{2}w^7-\frac{3}{2}w^6+\frac{1}{2}w^5+\frac{1}{2}w^4-\frac{3}{2}w^3+2w^2-\frac{1}{2}w-2 & -\frac{1}{2}w^7+\frac{3}{2}w^6+w^5-w^4-\frac{1}{2}w^3 \\ 3w^7-2w^6-w^4-w^3-2w-2 & w^7+2w^5-2w^4-2w^3+2w^2+1 & w^6-2w^5-2w^2+w \\ -4w^7+2w^6+3w^5-5w^4+2w^3-w^2-4w+3 & -w^7-w^6-w^4+w^3+w^2-2w+1 & 3w^7+w^5+w^4-2w^3+2w^2-w \end{vmatrix}$$

where $w = e^{2\pi i/15}$.

There are in total 143 abelian subgroups. There are 36 of them that are $\mathbb{Z}_5 \times \mathbb{Z}_3$ whose types are

$$r(G^A) = \{0, 16, 13, 13, 16,$$
$$\underline{1}, \overline{14}, \underline{11}, \underline{11}, \overline{14}, \tag{A.117}$$
$$\overline{2}, \underline{15}, \overline{12}, \overline{12}, \underline{15}\},$$

there are of 38 them that are $\mathbb{Z}_6 \times \mathbb{Z}_2$ whose types are

$$r(G^A) = \{0, \overline{4}, \overline{2}, 3, \underline{1}, \underline{5},$$
$$3, \overline{4}, \underline{5}, 3, \overline{4}, \underline{5}\}, \tag{A.118}$$

there are 29 of them that are $\mathbb{Z}_4 \times \mathbb{Z}_3$ whose types are

$$r(G^A) = \{0, 10, 3, 10,$$
$$\overline{2}, \overline{9}, \underline{5}, \overline{9},$$
$$\underline{1}, \underline{8}, \overline{4}, \underline{8}\}, \tag{A.119}$$

there are of 20 them that are $\mathbb{Z}_3 \times \mathbb{Z}_3$ whose types are

$$r(G^A) = \{0, \underline{1}, \overline{2},$$
$$6, 6, 6,$$
$$6, 6, 6\}. \tag{A.120}$$

and there are 20 of them that are $\mathbb{Z}_3 \times \mathbb{Z}_3$ whose types are

$$r(G^A) = \{0, \underline{1}, \overline{2},$$
$$7, 7, 7,$$
$$7, 7, 7\}. \tag{A.121}$$

The maximal $\mathbb{Z}_N$ subgroups

$$\{0, 16, 13, 13, 16\}\,,$$
$$\{0, 10, 3, 10\}\,,$$
$$\{0, 6, 6\}\,,$$
$$\{0, 7, 7\} \tag{A.122}$$

form the Dynkin diagram of $A_4 \times A_3 \times A_2 \times A_2$:

$$\begin{array}{cccccccccc} 16 & 13 & 13 & 16 & 10 & 3 & 10 & 6 & 6 & 7 & 7 \end{array} \tag{A.123}$$

The number of conjugacy classes in $\Gamma_{1,f}$ matches the flavor rank $f = 6$. Note that the $A_4$ singularity and $A_3$ singularity only has contribution $f = 2$, each of the two $A_2$ singularities only has contribution $f = 1$.

## B    More examples

### B.1    $E^{(6)}$

#### B.1.1    6d interpretation

The equation is a complete intersection in $\mathbb{C}^5$:

$$\begin{cases} y^2 = x^3 - 108z^2 \\ u^2 - vz = 0\,, \end{cases} \tag{B.1}$$

which takes the form of a Weierstrass model over a base $B_2$ with $A_1$ singularity at $u = v = z = 0$. We can replace

$$u = st \, , \; z = s^2 \, , \; v = t^2 \, . \tag{B.2}$$

After the resolving the base $(u, v, z; \delta_1)$, we get the Weierstrass model on the tensor branch

$$y^2 = x^3 - 108s^4\delta_1^2 \, . \tag{B.3}$$

Since there is a $(f, g)$ vanishes to order $(4, 6)$ at $s = \delta_1 = 0$, we need to further blow up $(s, \delta_1; \delta_2)$ on the base

The final tensor branch is then

$$\overset{\mathfrak{su}(3)}{3} - 1 - [E_6] \, . \tag{B.4}$$

### B.1.2   5d interpretation

We start with the complete intersection

$$\begin{cases} y^3 - z^2 = 108x^2 \\ u^2 - vx = 0 \end{cases} \tag{B.5}$$

We first do the blow up sequence

$$(u, v, x; \delta_1) \tag{B.6}$$

and get

$$\begin{cases} y^3 - z^2 = 108x^2\delta_1^2 \\ u^2 - vx = 0 \end{cases} \tag{B.7}$$

We then work in the patch of $v = 1$, and substitute $x = u^2$ into the first equation:

$$y^3 - z^2 = 108u^4\delta_1^2 \tag{B.8}$$

We further blow up

$$\begin{aligned} & (y^{(2)}, z^{(3)}, u^{(1)}, \delta_1^{(1)}; \delta_2) \\ & (y, z, \delta_1; \delta_3) \end{aligned} \tag{B.9}$$

and the $E_6$ singularity at $u = y = z = 0$:

$$(u, y, z; u_1) \, , \; (u_1, y, z; u_2) \, , \; (z, u_1; u_3) \, , \; (u_2, u_3; u_4) \, , \; (u_3, u_4; u_5) \, , \; (u_1, u_3; u_6) \, . \tag{B.10}$$

The resolved equation is

$$y^3\delta_3 u_1 u_2^2 u_4 - z^2 u_3 = 108u^4\delta_1^2 u_1^2 u_3 u_6^2 \, . \tag{B.11}$$

The compact divisor $\delta_3 = 0$ has two components:

$$\begin{aligned} S_1 : \quad & \delta_3 = z + 6\sqrt{3}iu^2\delta_1 u_1 u_6 = 0 \\ S_3 : \quad & \delta_3 = z - 6\sqrt{3}iu^2\delta_1 u_1 u_6 = 0 \, . \end{aligned} \tag{B.12}$$

Along with $S_2 : \delta_2 = 0$, the rank of the 5d SCFT is $r = 3$. The 5d geometry can also be thought as the decoupling of vertical divisor over the $(-3)$-curve, from the 6d geometry (B.3).

We use a notation similar to [19] to show the triple intersection numbers in a picture:

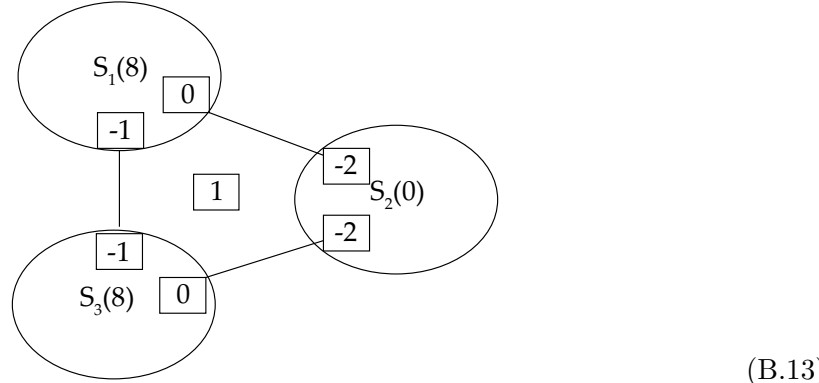

$$\text{(B.13)}$$

The number $n$ in the bracket $S_i(n)$ denotes the self-triple intersection number $S_i^3$, and the number in a square box inside $S_i$ is equal to $S_i S_j^2$ for the corresponding intersection curve $S_i \cdot S_j$. The number among three divisors $S_i$, $S_j$ and $S_j$ is the triple intersection number $S_i \cdot S_j \cdot S_k$.

The theory has an IR description of $SU(3)_0 - SU(2) - 5\boldsymbol{F}$ and UV flavor symmetry

$$G_F = E_6 \times SU(2) \times U(1). \qquad \text{(B.14)}$$

The 5d geometry can be obtained by the decoupling of the vertical divisor over the $(-3)$-curve in the 6d theory ( B.4).

## B.2  $E^{(7)}$

We present the detail of the 6d SCFT $T_X^{6d}$ associated to $\mathbb{C}^3/E^{(7)}$, which is non-very-Higgsable.

After shifting and rescaling variables, the equation is a complete intersection in $\mathbb{C}^5$:

$$\begin{cases} zv - u^3 = 0 \\ y^2 = x^3 - \frac{1}{3 \times 4^{1/3}} zx \,, \end{cases} \qquad \text{(B.15)}$$

which takes the form of a Weierstrass model over a base $B_2$ with $A_2$ singularity at $u = v = z = 0$. We can replace

$$u = st \,, \ z = s^3 \,, \ v = t^3 \,. \qquad \text{(B.16)}$$

After resolving $A_2$ base singularity, we get the Weierstrass model on the tensor branch

$$y^2 = x^3 - \frac{1}{3 \times 4^{1/3}} s^3 \delta_1^2 \delta_2 x \,. \qquad \text{(B.17)}$$

We need to do the further base blow-ups to remove cod-2 $(4, 6)$ singularities:

$$(s, \delta_1; \delta_3) \,, \ (s, \delta_3; \delta_4) \,. \qquad \text{(B.18)}$$

The final Weierstrass model is

$$y^2 = x^3 - \frac{1}{3 \times 4^{1/3}} s^3 \delta_1^2 \delta_2 \delta_3 x \,. \tag{B.19}$$

The final tensor branch is

$$\overset{\mathfrak{su}(2)}{2} - \overset{\mathfrak{so}(7)}{3} - \overset{\mathfrak{su}(2)}{2} - 1 - [E_7] \,. \tag{B.20}$$

## B.3 $\mathbb{C}^3/H_{36}$

In this case, $T_X^{6d}$ is also non-very-Higgsable, and to get $T_X^{5d}$ one needs to decouple three vector multiplets from the KK theory.

### B.3.1 6d interpretation

In this case $\mathbb{C}^3/H_{36}$ is described by the following complete intersection in $\mathbb{C}^5$

$$\begin{cases} 3u^2x + 2u^3 - 36uz - 36vx - x^3 + 432y^2 = 0, \\ u^2v - u^2z - 12v^2 + 9z^2 = 0. \end{cases} \tag{B.21}$$

We do the coordinate transformation

$$\begin{aligned} y &\to \frac{1}{12\sqrt{3}} y \\ z &= -\frac{2}{\sqrt{3}} z_1 - \frac{1}{24\sqrt{3}} v_1 + \frac{2i}{3} u_1^2 \\ v &= z_1 - \frac{1}{48} v_1 + \frac{i}{2} u_1^2 \\ u &= -2\sqrt{3} e^{\pi i/4} u_1 \,, \end{aligned} \tag{B.22}$$

and rewrite the equations as

$$\begin{cases} y^2 = x^3 + \left(36z_1 - 18iu_1^2 - \frac{3}{4} v_1\right) x + 3e^{\pi i/4} u_1(v_1 + 48z_1), \\ z_1 v_1 = u_1^4 \,. \end{cases} \tag{B.23}$$

The first equation is a Weierstrass model over a base $B_2$ with $A_3$ singularity at $z_1 = v_1 = u_1 = 0$. One can also introduce new base coordinates $(s, t)$:

$$z_1 = s^4 \,, \quad v_1 = t^4 \,, \quad u_1 = st \,. \tag{B.24}$$

The base has an $A_3$ singularity at $s = t = 0$, and it has the following toric resolution $\phi : \tilde{B}_2 \to B_2$. The resolved base $\tilde{B}_2$ is

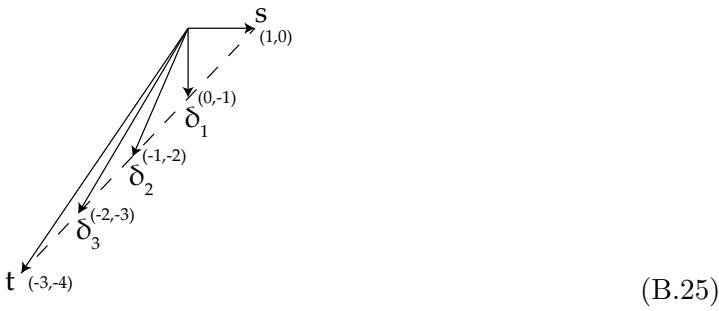

$$\tag{B.25}$$

The exceptional base divisors are $\delta_k = 0$ $(k = 1, 2, 3)$, which one-to-one correponds to lattice points $(1 - k, -k)$ in the $N$ lattice. For the line bundle $\mathcal{O}(-nK_{\tilde{B}_2})$, the monomial generator associated to the point $(p, q)$ in the dual lattice $M$ of $N$ is (for example see [104])

$$m_{(p,q)} = s^{p+n}t^{-3p-4q+n}\delta_1^{-q+n}\delta_2^{-p-2q+n}\delta_3^{-2p-3q+n}. \tag{B.26}$$

Thus the Weierstrass model after resolution is

$$y^2 = x^3 + \left(36s^4\delta_1^2 - 18is^2t^2\delta_1\delta_3 - \frac{3}{4}t^4\delta_3^2\right)\delta_1\delta_2^2\delta_3 x + 3e^{\pi i/4}st(48s^4\delta_1 + t^4\delta_3)\delta_1^2\delta_2^3\delta_3^2. \tag{B.27}$$

$$\Delta = \frac{27}{16}\delta_1^3\delta_2^6\delta_3^3((-24i\delta_1\delta_3s^2t^2 - \delta_3^2t^4 + 48\delta_1^2s^4)^3 + 144i\delta_1\delta_3s^2t^2(\delta_3^2t^4 + 48\delta_1^2s^4)^2). \tag{B.28}$$

The tensor branch is

$$
\begin{matrix}
& [2\boldsymbol{F}] & \\
& | & \\
\mathfrak{su}(2) & \mathfrak{g}_2 & \mathfrak{su}(2) \\
2 & - \quad 2 & - \quad 2 \,.
\end{matrix}
\tag{B.29}
$$

Note that the central $G_2$ gauge theory on a $(-2)$-curve should be $G_2 + 4\boldsymbol{F}$ due to anomaly cancellation, and the extra matter $2\boldsymbol{F}$ locates on the $I_1$ loci.

### B.3.2  5d interpretation

We start with the singular equation in (B.23)

$$
\begin{cases}
y^2 = x^3 + \left(36z_1 - 18iu_1^2 - \frac{3}{4}v_1\right)x + 3e^{\pi i/4}u_1(v_1 + 48z_1), \\
z_1v_1 = u_1^4 \,.
\end{cases}
\tag{B.30}
$$

We do the following resolution sequence

$$
\begin{aligned}
& (z_1, u_1, v_1; \delta_1) \,, \ (z_1, v_1, \delta_1; \delta_2) \,, \ (x, y, \delta_1; \delta_3) \,, \ (x, y, \delta_2; \delta_4) \\
& (y, \delta_4; \delta_5) \,, \ (\delta_4, \delta_5; \delta_6) \,.
\end{aligned}
\tag{B.31}
$$

The resulting equation is

$$
\begin{cases}
y^2\delta_5 = x^3\delta_3\delta_4 + \left(36z_1 - 18iu_1^2\delta_1\delta_3 - \frac{3}{4}v_1\right)\delta_1\delta_2^2\delta_4 x + 3e^{\pi i/4}u_1(v_1 + 48z_1)\delta_1^2\delta_2^3\delta_4, \\
z_1v_1 = u_1^4\delta_1^2\delta_3 \,.
\end{cases}
\tag{B.32}
$$

The compact divisors are

$$
\begin{aligned}
S_1 : \quad & \delta_3 = z_1 = 0 \\
S_2 : \quad & \delta_3 = v_1 = 0 \\
S_3 : \quad & \delta_5 = 0 \\
S_4 : \quad & \delta_6 = 0 \,.
\end{aligned}
\tag{B.33}
$$

The triple intersection numbers are

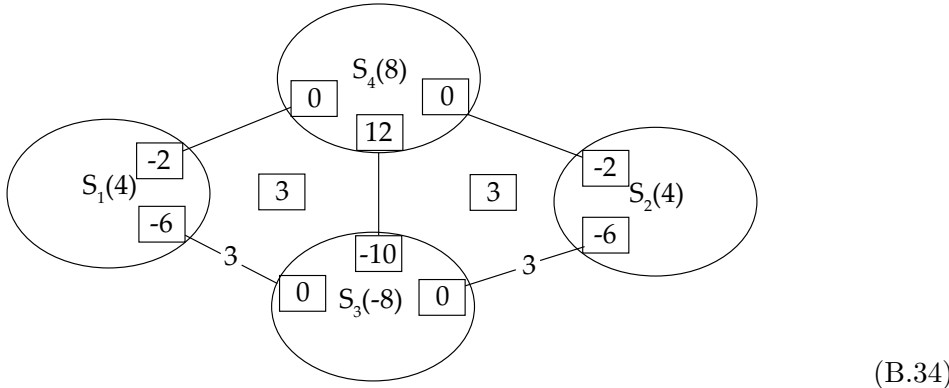

(B.34)

The rank-4 SCFT has an IR gauge theory description of

$$
\begin{array}{c}
2\boldsymbol{F} \\
| \\
SU(2)_0 - \; G_2 \; - SU(2)_0
\end{array}
$$

(B.35)

The UV flavor symmetry should be enhanced to

$$
G_F = SU(4) \times Sp(2) \,,
$$

(B.36)

where $SU(4)$ comes from the $A_3$ singularity, and $Sp(2)$ comes from the classical flavor symmetry of $G_2 + 2\boldsymbol{F}$. The 5d theory is generated by taking the KK reduction of $T_X^{6d}$ in (B.29), and decoupling the three vertical divisors over the $(-2)$-curves.

## B.4 $\;\; \mathbb{C}^3/H_{60}$

In this case after rescaling and shifting the coordinates we obtain a Weierstrass model defined by

$$
\begin{aligned}
f =& \frac{1}{27} z \left( -237160 w^2 z^3 + 15435 w^3 + 1214080 wz^6 - 2070784 z^9 \right) , \\
g =& \frac{1}{729} \big( -2327256960 w^2 z^9 + 423097360 w^3 z^6 - 38139885 w^4 z^3 \\
& + 1361367 w^5 + 6355937280 wz^{12} - 6902276096 z^{15} \big)
\end{aligned}
$$

(B.37)

whose discriminant is

$$
\Delta = \frac{49}{2187} \left( 5z^3 - w \right)^2 \left( 32z^3 - 7w \right)^5 \left( 320z^3 - 63w \right)^3 .
$$

(B.38)

The above Weierstrass model involves no codimension-two $(4,6)$ points hence there is no tensor branch. The 6D physics is given by the F-theory compactification on a local elliptic CY3 on which there is a type $I_2$ fibration along $w - 5z^3 = 0$, a type $I_{5,ns}$ fibration along $7w - 32z^3 = 0$ and a type $I_{3,ns}$ fibration along $63w - 320z^3 = 0$ all of which meet at the point $w = z = 0$ where the degree of $(f,g,\Delta)$ enhances to $(4,5,10)$.

## B.5 $\mathbb{C}^3/H_{168}$

### B.5.1 6d interpretation

In this case the Weierstrass model is given by

$$f = \frac{16}{3}z\left(-280w^2z^3 + 189w^4 - 48z^6\right),$$
$$g = \frac{64}{27}w\left(1456w^2z^6 - 12852w^4z^3 + 729w^6 - 4032z^9\right) \tag{B.39}$$

whose discriminant is

$$\Delta = -4096\left(4z^3 - 27w^2\right)^3\left(w^2 + 4z^3\right)^4. \tag{B.40}$$

At $w = z = 0$ the order of $(f, g, \Delta)$ is greater than $(4, 6, 12)$ therefore a base blow-up at $w = z = 0$ can be performed to obtain a minimal elliptic fibration given by

$$f = -\frac{16}{3}\delta z\left(280\delta w^2 z^3 - 189w^4 + 48\delta^2 z^6\right),$$
$$g = \frac{64}{27}\delta w\left(1456\delta^2 w^2 z^6 - 12852\delta w^4 z^3 + 729w^6 - 4032\delta^3 z^9\right) \tag{B.41}$$

whose discriminant is

$$\Delta = 4096\delta^2\left(27w^2 - 4\delta z^3\right)^3\left(w^2 + 4\delta z^3\right)^4 \tag{B.42}$$

where $w$ and $z$ cannot vanish simultaneously. The base blow-up is shown schematically in figure 3.

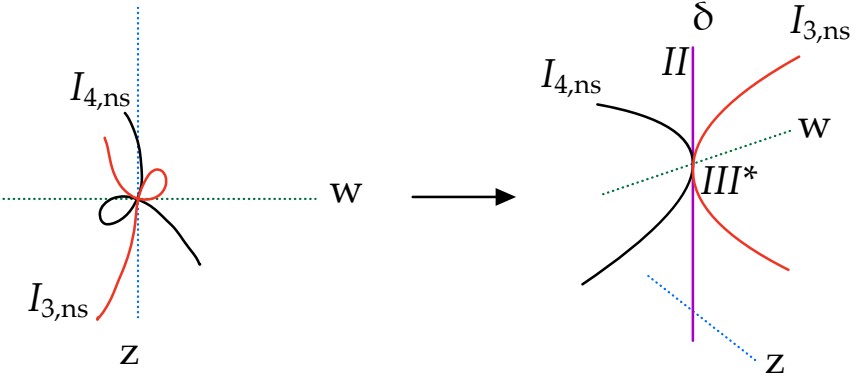

**Figure 3**: Before the base blow-up there are two self-intersecting divisors intersect at $w = z = 0$. After the base blow-up at $w = z = 0$ the $w = 0$ line and the $z = 0$ line are separated by the compact curve $\delta = 0$ on the base over which there is a type $II$ elliptic fibration except at the point $\delta = w = 0$.

Clearly at $\delta = z = 0$ nothing interesting happens but at $\delta = w = 0$ we see that all the three irreducible components of the discriminant locus given by (B.42) intersect and

the order of $(f, g, \Delta)$ on $\delta = 0$ enhances from $(1, 1, 2)$ to $(3, 5, 9)$. At this point we can let $z = 1$ hence we will study the Weierstrass model $y^2 = x^3 + fx + g$ where

$$
\begin{aligned}
f &= -\frac{16}{3}\delta \left(48\delta^2 + 280\delta w^2 - 189w^4\right), \\
g &= \frac{64}{27}\delta w \left(-4032\delta^3 + 1456\delta^2 w^2 - 12852\delta w^4 + 729w^6\right).
\end{aligned}
\tag{B.43}
$$

The discriminant of this Weierstrass model is

$$
\Delta = 4096\delta^2 \left(27w^2 - 4\delta\right)^3 \left(w^2 + 4\delta\right)^4
\tag{B.44}
$$

from which one can see that there is a type $I_{4,ns}$ fibration over $w^2 + 4\delta = 0$, a type $I_{3,ns}$ fibration over $27w^2 - 4\delta = 0$ and a type $II$ fibration over $\delta = 0$.

Now we try to resolve the singular equation $y^2 - x^3 - fx - g$ with (B.43). In this case we keep track of the new coordinates after the $i$-th blow up using subscripts $b_i$. We first perform a shift

$$
x = x_{s_1} - \frac{32}{3}w^3, \ \ \delta = \delta_{s_1} - \frac{1}{4}w^2
\tag{B.45}
$$

to put one of the codimension-two singularity along $x_{s_1} = y = \delta_{s_1} = 0$. We then apply the first blow-up

$$
(x_{s_1}, y, \delta_{s_1}; e_1)
\tag{B.46}
$$

whose proper transformation is

$$
\begin{aligned}
0 = {}& 258048\delta_{s_1 b_1}^4 e_1^2 w + 6912\delta_{s_1 b_1}^3 e_1^2 x_{s_1 b_1} - 424960\delta_{s_1 b_1}^3 e_1 w^3 + 35136\delta_{s_1 b_1}^2 e_1 w^2 x_{s_1 b_1} \\
& - 27e_1 x_{s_1 b_1}^3 + 614400\delta_{s_1 b_1}^2 w^5 + 864w^3 x_{s_1 b_1}^2 - 46080\delta_{s_1 b_1} w^4 x_{s_1 b_1} + 27y_{b_1}^2.
\end{aligned}
\tag{B.47}
$$

We then perform a second shift

$$
x_{s_1 b_1} = x_{s_2 b_1} + \frac{80}{3}w\delta_{s_1 b_1}
\tag{B.48}
$$

and a second blow-up

$$
(e_1, x_{s_2 b_1}, y_{b_1}; e_2)
\tag{B.49}
$$

whose proper transformation is

$$
\begin{aligned}
0 = {}& 16384\delta_{s_1 b_1}^4 e_{1 b_1}^2 w + 64\delta_{s_1 b_1}^2 e_{1 b_1}(4\delta_{s_1 b_1} e_{1 b_1} e_2 - 13w^2)x_{s_2 b_2} \\
& + 16w(-5\delta_{s_1 b_1} e_{1 b_1} e_2 + 2w^2)x_{s_2 b_2}^2 - e_{1 b_1} e_2^2 x_{s_2 b_2}^3 + y_{b_2}^2.
\end{aligned}
\tag{B.50}
$$

We then perform a third shift in the patch $x_{s_2 b_2} \neq 0$:

$$
w = w_{s_1} - \frac{8\delta_{s_1 b_1}^2 e_{1 b_1}}{x_{s_2 b_2}}, \ e_2 = e_{2 s_1} + \frac{448\delta_{s_1 b_1}^3 e_{1 b_1}}{x_{s_2 b_2}^2}
\tag{B.51}
$$

After the third blow-up

$$
(w_{s_1}, e_{2 s_1}, y_{b_2}; d_1)
\tag{B.52}
$$

whose proper transformation is

$$
32x_{s_2 b_2}^2 d_1 w_{s_1 b_1}^3 - x_{s_2 b_2} e_{1 b_1}(x_{s_2 b_2} e_{2 s_1 b_1} + 40\delta_{s_1 b_1} w_{s_1 b_1})^2 + y_{b_3}^2 = 0,
\tag{B.53}
$$

one can check that there are no more codimension-two singularities in the above proper transformation.

The projective relations of the blow-ups are

$$(x_{s_2 b_2} d_1 e_{2 s_1 b_1}, d_1^2 y_{b_3} e_{2 s_1 b_1}, \delta_{s_1 b_1})$$
$$(e_{1 b_1} e_{2 s_1 b_1} d_1, x_{s_2 b_2}, y_{b_3} e_{2 s_1 b_1} d_1) \tag{B.54}$$
$$(w_{s_1 b_1}, e_{2 s_1 b_1}, y_{b_3}).$$

Note that the original surface $S : \delta = 0$ becomes

$$-\frac{1}{4} d_1^2 w_{s_1 b_1}^2 x_{s_2 b_2}^2 + 432 \delta_{s_1 b_1}^4 e_{1 b_1}^2 + x_{s_2 b_2} d_1 \delta_{s_1 b_1} e_{1 b_1} (4 \delta_{s_1 b_1} w_{s_1 b_1} + e_{2 s_1 b_1} x_{s_2 b_2}) = 0. \tag{B.55}$$

To see the geometry of $S$, we define the following curves on $S$

$$C_1 : d_1 = e_{1 b_1} = y_{b_3} = 0$$
$$C_2 : e_{1 b_1} = w_{s_1 b_1} = y_{b_3} = 0$$
$$C_3 : e_{2 s_1 b_1} = 32 x_{s_2 b_2}^2 d_1 w_{s_1 b_1}^3 - 1600 x_{s_2 b_2} e_{1 b_1} \delta_{s_1 b_1}^2 w_{s_1 b_1}^2 + y_{b_3}^2$$
$$\qquad = -\frac{1}{4} d_1^2 w_{s_1 b_1}^2 x_{s_2 b_2}^2 + 432 \delta_{s_1 b_1}^4 e_{1 b_1}^2 + 4 x_{s_2 b_2} d_1 \delta_{s_1 b_1}^2 e_{1 b_1} w_{s_1 b_1} = 0 \tag{B.56}$$
$$C_4 : w_{s_1 b_1} = y_{b_3}^2 - x_{s_2 b_2}^3 e_{1 b_1} e_{2 s_1 b_1}^2 = 432 \delta_{s_1 b_1} e_{1 b_1} + x_{s_2 b_2}^2 d_1 e_{2 s_1 b_1} = 0$$
$$C_5 : \delta_{s_1 b_1} = w_{s_1 b_1} = y_{b_3}^2 - x_{s_2 b_2}^3 e_{1 b_1} e_{2 s_1 b_1}^2.$$

Note that $S$ has a curve of double point singularity along $C_1$. The intersection relation between these curves are

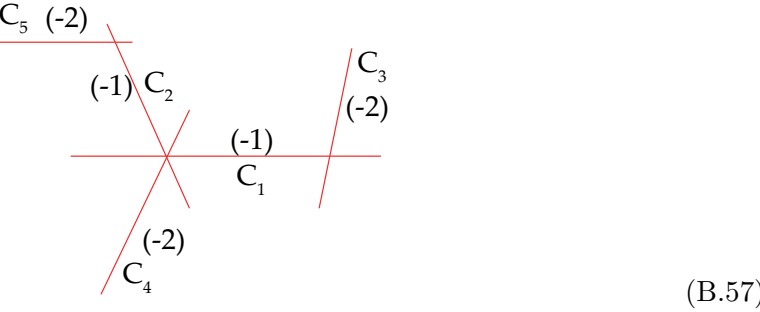

$$\tag{B.57}$$

The exceptional divisors of $Sp(2)$ are $e_{1 b_1} = 0$, $e_{2 s_2 b_1} = 0$ and the affine node $\delta_{s_1 b_1} = 0$. The exceptional divisors of $Sp(1)$ are $d_1 = 0$ and the affine node $w_{s_1 b_1} = 0$. Hence from the $(-1)$-curves on $S$, we derive the matter spectrum on the 6d tensor branch under the representations of flavor symmetry group $Sp(2) \times Sp(1)$:

$$C_1 \to (\mathbf{5}, \mathbf{2}), \ C_2 \to (\mathbf{1}, \mathbf{2}). \tag{B.58}$$

One can check that the above matter representations satisfy the 6d gauge anomaly cancellation conditions.

Similar to the discussions of $\Delta(48)$ in section 4.1.1, the 6d SCFT is different from the rank-1 E-string since the contributions to gravitational anomaly ($\Delta H_{\text{localized}} = 41$) and flavor rank (3 instead of 8) are both different from those of a rank-1 E-string.

### B.5.2  5d interpretation

In the 5d case, the first blow up is a weighted blow up

$$(x^{(2)}, y^{(3)}, z^{(1)}, w^{(1)}; \delta_1) \tag{B.59}$$

instead of blowing up the base. We will not repeat the details of the resolution. In the end, the curves on the compact divisor $S_1 : \delta_1 = 0$ are

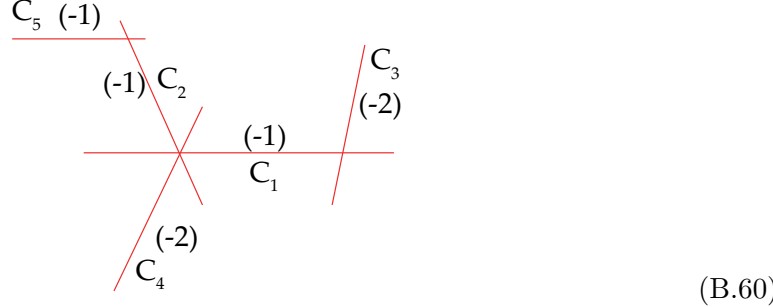

$$\tag{B.60}$$

Note that the curve $C_5$ correponds to the affine node of $Sp(2)$ and $Sp(1)$, which becomes a $(-1)$-curve after the blow down of $S$ from the 6d geometry. The flavor symmetry factors read off from the geometry are $G_F = SU(2)^2 \times U(1)$.

Note that the resolution of (A.104) is also discussed in [105].

### B.6  $E^{(2)}$

We start with the singular equation

$$z^3 - w(y^3 + 12\sqrt{3}ix^2) = 0 \,, \tag{B.61}$$

and do the blow up sequence

$$(x, y, z, w; \delta_1) \,, \; (x, y, z; \delta_2) \,, \; (x, \delta_2; \delta_3) \,, \; (\delta_2, \delta_3; \delta_4) \,. \tag{B.62}$$

We get

$$z^3\delta_2 - w(y^3\delta_1\delta_2 + 12\sqrt{3}ix^2\delta_3) = 0 \,. \tag{B.63}$$

Now we define

$$U = y^3\delta_1\delta_2 + 12\sqrt{3}ix^2\delta_3 \,. \tag{B.64}$$

After the blow up

$$(z, w, U; \delta_5) \,, \; (\delta_5, w; \delta_6) \,, \tag{B.65}$$

the resolved equation is a complete intersection

$$\begin{cases} U\delta_5\delta_6 = y^3\delta_1\delta_2 + 12\sqrt{3}ix^2\delta_3 \\ z^3\delta_5 = wU \,. \end{cases} \tag{B.66}$$

Since $\delta_2 = 0$ is an empty set, the variable $\delta_2$ can be dropped from the equations.

The compact divisor $S_1 : \delta_1 = 0$ has equation

$$\begin{cases} U\delta_5\delta_6 = 12\sqrt{3}ix^2\delta_3 \\ z^3\delta_5 = wU\,. \end{cases} \tag{B.67}$$

Now we discuss the flavor symmetry $G_F$ from non-compact divisors. We plot the configuration of curves on $\delta_1 = 0$:

$$\begin{aligned}
C_1 : &\quad \delta_1 = \delta_3 = U = z = 0 \\
C_2 : &\quad \delta_1 = \delta_4 = U\delta_5\delta_6 - 12\sqrt{3}i\delta_3 = z^3\delta_5\delta_6 - wU = 0 \\
C_3 : &\quad \delta_1 = \delta_5 = x = U = 0 \\
C_4 : &\quad \delta_1 = \delta_6 = x = z^3\delta_5 - wU = 0 \\
C_5 : &\quad \delta_1 = x = z = U = 0\,.
\end{aligned} \tag{B.68}$$

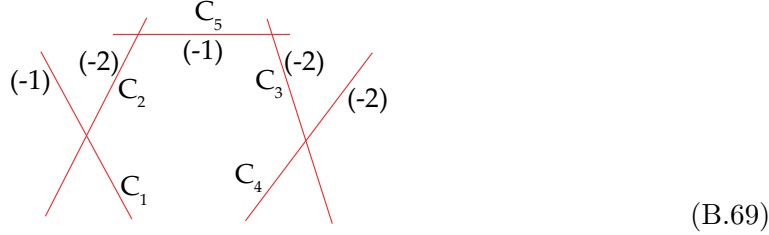

$$\tag{B.69}$$

Note that $C_1$ and $C_2$ come from the resolution of the codimension-two $D_4$ singularity, while $C_3$ and $C_4$ come from the $A_2$ singularity. $C_5$ is the curve of cusp singularity $x = z = U = 0$.

A non-trivial fact is that the curve $C_1$ has normal bundle $\mathcal{O}(-1) \oplus \mathcal{O}(-1)$. From the perspective of $D_4$ singularity over the $\delta_1$-axis, the fiber $\delta_3 = 0$ has three $\mathbb{P}^1$ components over any point $\delta_1 \neq 0$. The point $\delta_1 = 0$ is exactly the location where the fiber $\delta_3 = 0$ degenerates. Since the degenerate $\mathbb{P}^1$ fiber $\delta_1 = \delta_3 = 0$ is rigid, it should have normal bundle $\mathcal{O}(-1) \oplus \mathcal{O}(-1)$. Note that in the context of elliptic Calabi-Yau threefolds, the degenerate point is exactly the ramification point in presence of Tate monodromy [91].

Hence from the $(-2)$-curves on $S_1$, we can read off flavor symmetry factors $G_F = SU(3) \times SU(2) \times U(1)$.

## B.7 $E^{(4)}$

We start with the singular equation

$$z^2 - w(108x^4 - y^3) = 0\,, \tag{B.70}$$

and do the blow up sequence

$$\begin{aligned}
&(x^{(1)}, y^{(1)}, z^{(2)}, w^{(1)}; \delta_1)\,, \quad (x, y, z; \delta_2)\,, \quad (y, z, \delta_2; \delta_3)\,, \quad (\delta_2, z; \delta_4) \\
&(\delta_3, \delta_4; \delta_5)\,, \quad (\delta_4, \delta_5; \delta_6)\,, \quad (\delta_2, \delta_4; \delta_7)\,.
\end{aligned} \tag{B.71}$$

We get

$$z^2 - w(108x^4\delta_1\delta_2^2\delta_4\delta_7^2 - y^3\delta_2\delta_3^2\delta_5) = 0\,. \tag{B.72}$$

Now we define

$$U = 108x^4\delta_1\delta_2^2\delta_4\delta_7^2 - y^3\delta_2\delta_3^2\delta_5 \,. \tag{B.73}$$

After the blow up

$$(z, w, U; \delta_8) \tag{B.74}$$

the resolved equation is a complete intersection

$$\begin{cases} U\delta_8 = 108x^4\delta_1\delta_2^2\delta_4\delta_7^2 - y^3\delta_2\delta_3^2\delta_5 \\ z^2 = wU \,. \end{cases} \tag{B.75}$$

Since $\delta_2 = 0$ and $\delta_4 = 0$ are empty sets, the variables $\delta_2$ and $\delta_4$ can be dropped from the equations.

The compact divisor $S_1 : \delta_1 = 0$ has the equation

$$\begin{cases} U\delta_8 + y^3\delta_3^2\delta_5 = 0 \\ z^2 = wU \,. \end{cases} \tag{B.76}$$

We plot the $\mathbb{P}^1$ curves on $S_1$:

$$\begin{aligned}
C_1 : \quad & \delta_1 = \delta_3 = U = z = 0 \\
C_2 : \quad & \delta_1 = \delta_5 = U = z = 0 \\
C_3 : \quad & \delta_1 = \delta_6 = U\delta_8 + y^3\delta_3^2\delta_5 = z^2 - wU = 0 \\
C_4 : \quad & \delta_1 = \delta_7 = U\delta_8 + y^3\delta_3^2\delta_5 = z^2 - wU = 0 \\
C_5 : \quad & \delta_1 = \delta_8 = y = z^2 - wU = 0 \\
C_6 : \quad & \delta_1 = y = z = U = 0 \,.
\end{aligned} \tag{B.77}$$

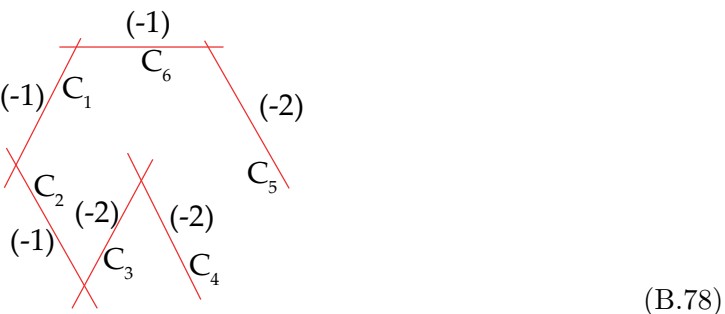

$$\tag{B.78}$$

The curves $C_1$, $C_2$, $C_3$ and $C_4$ comes from the resolution of codimension-two $E_6$ singularity. Similar to the case in the previous section, the curves $C_1$ and $C_2$ are exactly the degenerate $\mathbb{P}^1$s at the ramification point of the $E_6$ singularity. From the curves on $S_1$, one can read off the flavor symmetry factors $G_F = SU(3) \times SU(2) \times U(1)^2$.

## B.8 $G'_m$ with odd $m$

The singular equation is

$$\begin{cases} u^2 - y(x + 4y^m) = 0 \\ v^2 - xz = 0 \end{cases} \tag{B.79}$$

The codimension-two singularities are $D_{m+2} \times A_1^2$.

When $m = 1$, the singular equation is a complete intersection of two quadratics in $\mathbb{C}^5$, and after the resolution

$$(x, y, z, u, v; \delta_1) \,, \tag{B.80}$$

the compact divisor $\delta_1 = 0$ is a smooth dP$_5$. The 5d SCFT $T_X^{5d}$ is the Seiberg rank-1 $E_5$ theory. Note that the flavor symmetry factors $SU(4) \times SU(2)^2$ from the codimension-two singularities is enhanced to

$$G_F = SO(10) \,. \tag{B.81}$$

When $m = 2k + 1 > 1$, the resolution sequence that gives rise to the compact divisors is

$$\begin{aligned}
&(x, y, z, u, v; \delta_1) \\
&(x^{(2)}, u^{(1)}, v^{(1)}, \delta_j^{(1)}; \delta_{j+1}) \quad (j = 1, \ldots, k) \,.
\end{aligned} \tag{B.82}$$

The resolved equation is

$$\begin{cases} u^2 - y \left( x + 4y^{2k+1} \prod_{j=1}^k \delta_j^{2k+2-2j} \right) = 0 \\ v^2 - xz = 0 \end{cases} \tag{B.83}$$

The triple intersection numbers among $S_j : \ \delta_j = 0$ are

$$\tag{B.84}$$

After a series flops that blow down $S_{k+1}$ and blow up $S_1$, the new triple intersection numbers are

$$\tag{B.85}$$

The 5d IR gauge theory description is

$$4\boldsymbol{F} - SU(2) - SU(2)_0 - \cdots - SU(2)_0 \,. \tag{B.86}$$

The flavor symmetry factors $SO(2m + 4) \times SU(2)^2$ are generally enhanced to

$$G_F = SO(2m + 8) \,. \tag{B.87}$$

## B.9 $\mathbb{C}^3/\Delta(6n^2)$, $n = 2k$

The singularity $\mathbb{C}^3/\Delta(6n^2)$ $(n = 2k)$ has the following hypersurface equation

$$-20w^{\frac{n}{2}+1}x^3 + 36w^{\frac{n}{2}+1}xy + 108w^{n+1} + 5wx^2y^2 - 4wx^4y + wx^6 - 2wy^3 + 4z^2 = 0. \quad \text{(B.88)}$$

Apart from cod-3 singularity at $x = y = z = w = 0$, there are also cod-2 singularities at

$$
\begin{aligned}
A_1 &: \ y - \frac{x^2}{2} = z = w = 0 \\
D_{\frac{n}{2}+2} &: \ y - x^2 = z = w = 0 \\
A_1 &: \ y - \frac{x^2}{3} = z = x^3 - 27w^{\frac{n}{2}} = 0 \,.
\end{aligned}
\quad \text{(B.89)}
$$

If $3 \nmid n$, the last cod-2 loci is singular. If $3 \mid n$, the last cod-2 loci is smooth with three irreducible components.

For $n = 4$, we first do coordinate substitution

$$y = y_2 + \frac{x^2}{3}, \quad \text{(B.90)}$$

and rewrite the singular equation as

$$
\begin{cases}
0 = 4z^2 - 2wy_2^3 + 3wx^2y_2^2 - \frac{4}{3}wxy_2U + \frac{4}{27}wU^2 \\
U = x^3 - 27w^2 \,.
\end{cases}
\quad \text{(B.91)}
$$

We use the following resolution sequence

$$(y_2, z, U; \delta_1) \,, \ (x, z, w, U, \delta_1; \delta_2) \,, \ (z, w, \delta_1, \delta_2; \delta_3) \quad \text{(B.92)}$$

to get

$$
\begin{cases}
0 = 4z^2 - 2wy_2^3\delta_1 + 3wx^2y_2^2\delta_2 - \frac{4}{3}wxy_2U\delta_2 + \frac{4}{27}wU^2\delta_2 \\
0 = x^3\delta_2 - 27w^2\delta_3 - U\delta_1 \,.
\end{cases}
\quad \text{(B.93)}
$$

Then we look at the patch of $\delta_1 \neq 0$, and plug

$$U = \frac{x^3\delta_2 - 27w^2\delta_3}{\delta_1} \quad \text{(B.94)}$$

into the first equation. After the shifting

$$y_3 = y_2 - \frac{2\delta_2 x^2}{3\delta_1}, \quad \text{(B.95)}$$

the first equation becomes

$$108\delta_2\delta_3^2w^5 + 16\delta_2^2\delta_3w^3x^3 + 36\delta_1\delta_2\delta_3w^3xy_3 - \delta_1^2\delta_2wx^2y_3^2 - 2\delta_1^3wy_3^3 + 4\delta_1^2z^2 = 0. \quad \text{(B.96)}$$

We can do the further blow ups and resolve the remaining $D_4$ and $A_1$ codimension-two singularities

$$(z, w, y_3; \delta_4) \,, \ (z, \delta_4; \delta_5) \,, \ (\delta_4, \delta_5; \delta_6) \,, \ (z, w, \delta_2 x^2 + 2\delta_1\delta_4\delta_5\delta_6^2 y_3; \delta_7) \quad \text{(B.97)}$$

The resolved equation is

$$\delta_4 w (4\delta_2\delta_3\delta_7 w^2 (27\delta_3\delta_4^2\delta_5^2\delta_6^4\delta_7^2 w^2 + 4\delta_2 x^3) + 36\delta_1\delta_2\delta_3\delta_4\delta_5\delta_6^2\delta_7 w^2 xy_3$$
$$- \delta_1^2 y_3^2 (\delta_2 x^2 + 2\delta_1\delta_4\delta_5\delta_6^2 y_3)) + 4\delta_1^2\delta_5 z^2 = 0 \tag{B.98}$$

The two compact divisors are $S_2 : \delta_2 = 0$ and $S_3 : \delta_3 = 0$. The other divisors are non-compact, $S_1 : \delta_1 = 0$, $S_6 : \delta_6 = 0$ and $S_7 : \delta_7 = 0$ are irreducible while $S_5 : \delta_5 = 0$ has two components. This exactly matches $r = 2$, $f = 5$.

## B.10  $C_{3l,l}^{(1)}, 3 \mid l$

Start from the singular equation

$$w^l x^2 + 3w^{2l} x - 6w^l yz + 9w^{3l} - xyz + y^3 + z^3 = 0, \tag{B.99}$$

one can always do the following sequence of resolutions

$$(x, y, z, w; \delta_1)$$
$$(x, y, z, \delta_i; \delta_{i+1}) \quad (i = 1, \dots, l-1). \tag{B.100}$$

Nonetheless, the resulting equation is still singular and more blow ups need to be done. For $l = 3$, the resolution sequence giving rise to compact divisors are

$$(x, y, z, w; \delta_1)$$
$$(x, y, z, \delta_1; \delta_2)$$
$$(x, y, z, \delta_2; \delta_3) \tag{B.101}$$
$$(y, z, \delta_1; \delta_4).$$

The equation after this is

$$w^3 x^2 \delta_1^2 \delta_2 + 3w^6 x \delta_1^4 \delta_2^2 \delta_4^2 - 6w^3 yz \delta_1^2 \delta_2 \delta_4^2 + 9w^9 \delta_1^6 \delta_2^3 \delta_4^4 - xyz + y^3 \delta_4 + z^3 \delta_4 = 0. \tag{B.102}$$

The compact divisors are $S_i : \delta_i = 0$, $(i = 1, \dots, 4)$. Then intersection numbers among them are

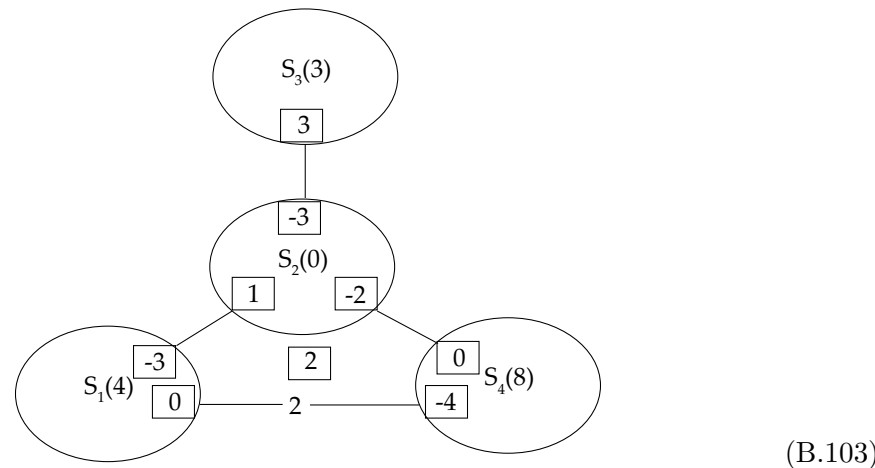

$$\tag{B.103}$$

The number "2" on a edge means that the intersection curve $S_1 \cdot S_4$ has two irreducible components, and $S_1 S_2 S_4 = 2$.

After this, there are still codimension-two singularities which can be resolved without changing the triple intersection numbers among compact divisors.

**B.11**  $D_{3l,l}^{(1)}$**,** $l = 2k$

We study the $D_{3l,l}^{(1)}$ singularity with $l = 2$. The singular equation is a complete intersection

$$\begin{cases} 4v^2u + 12vu^2 - 6x^2u + 6zu + 36u^3 - vx^2 + vz + x^3 + 3xz = 0, \\ y^2 - uz = 0. \end{cases} \tag{B.104}$$

The resolution sequence that generates compact divisors are

$$(x, y, z, u, v; \delta_1) , (\delta_1, z; \delta_2) , (\delta_2, y; \delta_3) , (\delta_2, \delta_3; \delta_4) , (\delta_3, z; \delta_5) . \tag{B.105}$$

The transformed equation is

$$\begin{cases} (4v^2u + 12vu^2 + 36u^3 - vx^2 - 6x^2u + x^3)\delta_1 + z(6u + v + 3x)\delta_5 = 0, \\ y^2\delta_3 - uz\delta_2 = 0. \end{cases} \tag{B.106}$$

The irreducible compact divisors are $S_1 : \delta_1 = 0$, $S_2 : \delta_3 = 0$, $S_3 : \delta_4 = 0$ and $S_4 : \delta_5 = 0$. The triple intersection numbers are computed with the methods in section 2.2.2:

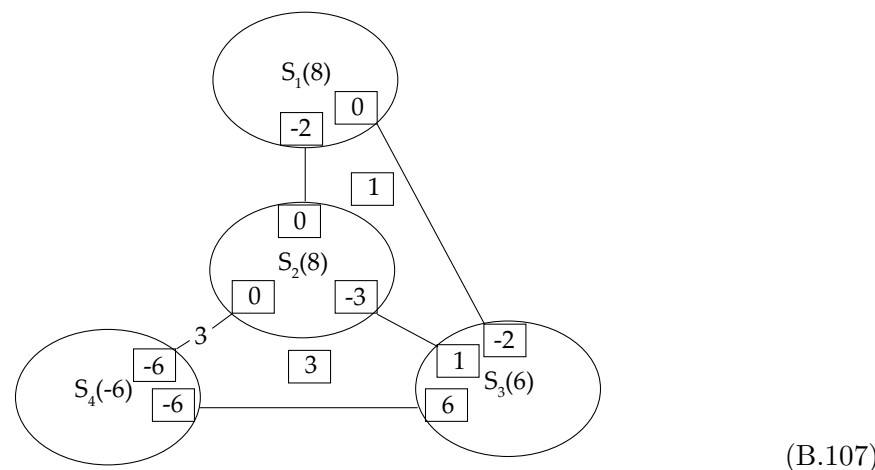

$$\tag{B.107}$$

The geometry does not have a ruling structure since the curve $S_1 \cdot S_3$ is a section curve on $S_1$ but a ruling curve on $S_3$.

**B.12**  $H_{72}$

We start with the singular equation

$$0 = -236117700w^3x^2 + 2526543w^2y^2 - 1744878w^4y + 321071w^6 \tag{B.108}$$
$$+ 638296200wx^2y + 43435278000x^4 - 1102736y^3 + 1608714000z^3 \tag{B.109}$$

We first do the resolution

$$(x, y, z, w; \delta_1) \tag{B.110}$$

and get the equation

$$0 = -236117700w^3x^2\delta_1^2 + 2526543w^2y^2\delta_1 - 1744878w^4y\delta_1^2 + 321071w^6\delta_1^3$$
$$+ 638296200wx^2y\delta_1 + 43435278000x^4\delta_1 - 1102736y^3 + 1608714000z^3 . \tag{B.111}$$

The equation is still singular at the loci

$$432x^2 - w^3\delta_1 = 164y - 9\delta_1 w^2 = z = 0$$
$$x = y - \delta_1 w^2 = z = 0\,. \tag{B.112}$$

Now we substitute

$$y = y_2 + \frac{9\delta_1 w^2}{164}\,, \tag{B.113}$$

and rewrite (B.111) as

$$(3723875\delta_1^3 w^6)/16 - 1102736y_2^3 - \frac{72075}{2}\delta_1^2 w^3(5580x^2 + 41wy_2)+ \tag{B.114}$$
$$1395\delta_1(5580x^2 + 41wy_2)^2 + 1608714000z^3 = 0$$

Now we can rewrite the equation as a complete intersection with the new variable $\Delta$

$$\begin{cases} (3723875\delta_1^3 w^6)/16 - 1102736y_2^3 - \frac{72075}{2}\delta_1^2 w^3\Delta + 1395\delta_1\Delta^2 + 1608714000z^3 = 0 \\ \Delta - (5580x^2 + 41wy_2) = 0 \end{cases}$$
$$\tag{B.115}$$

After the resolution

$$(\delta_1, y_2, \Delta, z; \delta_2), \tag{B.116}$$

the equations become

$$\begin{cases} (3723875\delta_1^3 w^6)/16 - 1102736y_2^3 - \frac{72075}{2}\delta_1^2 w^3\Delta + 1395\delta_1\Delta^2 + 1608714000z^3 = 0 \\ \Delta\delta_2 - (5580x^2 + 41wy_2\delta_2) = 0 \end{cases}$$
$$\tag{B.117}$$

The first equation is still singular at

$$12\Delta - 155\delta_1 w^3 = y_2 = z = 0\,. \tag{B.118}$$

Now we substitute

$$\Delta = \Delta_2 + \frac{155\delta_1 w^3}{12}\,, \tag{B.119}$$

and rewrite the first equation of (B.117) as

$$1395\delta_1\Delta_2^2 - 1102736y_2^3 + 1608714000z^3 = 0\,. \tag{B.120}$$

The singularity at $\Delta_2 = y_2 = z = 0$ is of du Val type $D_4$, and we can perform the following resolution sequence

$$(\Delta_2, y_2, z; \delta_3)\,,\ (\Delta_2, \delta_3; \delta_4)\,,\ (\delta_3, \delta_4; \delta_5) \tag{B.121}$$

Finally, we use the resolutions

$$(x, \delta_2; \delta_6)\,,\ (\delta_2, \delta_6; \delta_7) \tag{B.122}$$

to make the second equation in (B.117) smooth as well. The final equations are

$$\begin{cases} 1395\delta_1\Delta_2^2\delta_4 + (-1102736y_2^3 + 1608714000z^3)\delta_3 = 0 \\ (\Delta_2\delta_3\delta_4^2\delta_5^3 + \frac{155\delta_1 w^3}{12} + 41wy_2\delta_3\delta_4\delta_5^2)\delta_2 - 5580x^2\delta_6 = 0 \end{cases}$$
$$\tag{B.123}$$

The compact divisors are $\delta_1 = 0$, $\delta_6 = 0$ and $\delta_7 = 0$. The equation for $\delta_1 = 0$ is

$$\begin{cases} -1102736y_2^3 + 1608714000z^3 = 0 \\ (\Delta_2\delta_3\delta_4^2\delta_5^3 + 41wy_2\delta_3\delta_4\delta_5^2)\delta_2 - 5580x^2\delta_6 = 0 \end{cases} \tag{B.124}$$

The first equation is reducible, hence $\delta_1 = 0$ has three irreducible components. The other compact divisors are irreducible, and we get the correct 5d rank $r = 5$.

For the flavor rank, note that the sequence in (B.121) gives new non-compact divisors $\delta_4 = 0$ and $\delta_5 = 0$ as the same way as the resolution of a $D_4$ singularity. $\delta_4 = 0$ has three irreducible components and $\delta_5 = 0$ is irreducible. Along with the non-compact divisor $\Delta = 0$, they give rise to the correct flavor rank $f = 5$.

## B.13  $H_{216}$

Starting from the singular equation

$$\begin{aligned} 0 = {} & 799668536794871707782 0w^2x^3 + 5751067536004617099w^2y^2 - 52005772276829280w^2yz \\ & + 131753659993217447094w^4y + 201282072182100w^2z^2 - 1056790655463324600w^4z \\ & + 111232422480825844134 3w^6 + 21970495217646693900x^3y + 826556965553185776x^3z \\ & + 17145803149032y^2z - 8560505896492y^3 - 8610088608588yz^2 + 24791356048z^3\,, \end{aligned} \tag{B.125}$$

we first substitute variables

$$z \to z - 12420w^2 \ , \ y \to y + \frac{112887w^2}{1093}\,. \tag{B.126}$$

The equation is rewritten as

$$\begin{cases} -4778596(1791427y - 5188z)(y - z)^2 + 804043740810492U(27325y + 1028z)+ \\ 3038392539w^2(66673y - 516z)(27325y + 1028z) = 0 \\ U = x^3 + 27w^4 \end{cases} \tag{B.127}$$

We do the following resolution sequence

$$\begin{aligned} & (y, z, U; \delta_1) \ , \ (U, \delta_1, x, w; \delta_2) \ , \ (x, U, \delta_1, \delta_2; \delta_3) \ , \ (U, \delta_1, \delta_2, \delta_3; \delta_4) \\ & (\delta_1, \delta_2; \delta_5) \ , \ (\delta_1, 27325y + 1028z; \delta_6) \ , \ (\delta_2, \delta_6; \delta_7) \ , \ (\delta_5, 27325y + 1028z; \delta_8) \\ & (U, \delta_2; \delta_9) \ , \ (\delta_1, \delta_9; \delta_{10}) \ , \ (\delta_5, \delta_9; \delta_{11})\,. \end{aligned} \tag{B.128}$$

The resolved equation is

$$\begin{cases} -4778596(1791427y - 5188z)(y - z)^2\delta_1\delta_5 + 804043740810492U(27325y + 1028z)\delta_9+ \\ 3038392539w^2(66673y - 516z)(27325y + 1028z)\delta_2\delta_5\delta_7\delta_8\delta_9 = 0 \\ U\delta_1\delta_6 = x^3\delta_2\delta_3^2\delta_4 + 27w^4\delta_2^2\delta_5\delta_7\delta_8\delta_9\delta_{10}\delta_{11}\,. \end{cases} \tag{B.129}$$

The compact divisors are $S_1 : \delta_3 = 0$, $S_2 : \delta_4 = 0$, $S_3 : \delta_6 = 0$, $S_4 : \delta_7 = 0$, $S_5 : \delta_8 = 0$, $S_6 : \delta_9 = y - z = 0$, $S_7 : \delta_9 = 1791427y - 5188z = 0$, $S_8 : \delta_{10} = 0$, $S_9 : \delta_{11} = 0$. Note that the equation of $\delta_9 = 0$ has two irreducible components. The number of irreducible compact surfaces exactly matches the expected rank $r = 9$.

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
