# Peer review of "D and 6D SCFTs from $\mathbb{C}^3$ orbifolds"

_SciPost Physics_

## Round 3 · Referee Report · Lakshya Bhardwaj (Referee 1) · 2022-1-9

Strengths

  1. The paper studies 5d SCFTs by using geometric constructions in M-theory that are singular and directly construct the conformal point. A lot of the recent progress in the study of 5d SCFTs has instead been achieved by studying $smooth$ geometries which describe physics on the extended Coulomb branch of the 5d SCFTs. In such approaches, any results about the 5d SCFTs are derived by extrapolating (in a physically justified way) from the extended Coulomb branch. Thus, analysis of 5d SCFTs using $singular$ geometries is extremely important: it not only provides a way of cross-checking the results obtained using smooth geometries, but also might point to interesting effects and theories invisible in the world of smooth geometries. Systematic analyses of 5d SCFTs using singular geometries have only recently begun, and this paper is an important and timely contribution in this effort.

  2. The presentation of the paper is immaculate.

  3. The introduction contains a nice overview of all the methods used to study 5d SCFTs, which could be very helpful for other researchers in the community.

  4. The paper establishes an interesting bridge between physics and mathematics, relating 5d SCFTs to 3d McKay correspondence.

  5. There are many concrete examples illustrating the methods used in the paper.

Weaknesses

One of the questions explored in this paper is whether there is any difference between three seemingly different F-theory constructions of the 6d SCFT known as E-string theory. The three different constructions are given by using Types $I_0$, $I_1$ and $II$ fibers on a compact rational $(-1)$-curve.

The authors conjecture (e.g. in section 4.2.1) that they are different. However, their arguments for reaching this conclusion/conjecture are not clear and satisfactory. One of their arguments is based on anomalies and suggests that one of the following two possibilities hold: 1. Either the $I_1$ E-string theory is different from the standard $I_0$ E-string theory. 2. Or it is equivalent to the standard $I_0$ E-string theory along with 4 hypers. Then they say that they "conservatively" conjecture that the first possibility is the correct one. However, a conservative approach would suggest second possibility to be the correct one. In fact, they see the presence of these 4 hypers explicitly at the location of collision between $I_1$ and $I_4^s$ singularities in the Weierstrass model discussed in that section.

Another of their arguments revolves around the fact that the resolution of the geometry contains a singular $dP_6$ instead of a smooth $dP_9$. However, it is not clear from their presentation how this suggests that the $I_1$ E-string theory is different from the $I_0$ one. Let me ask a question which might be helpful in regard to sharpening (or shooting down) this argument: Can the singular $dP_6$ described in the paper be geometrically coupled to an affine $E_8$ flavor symmetry? This is natural to ask, as the smooth $dP_6$ can only be geometrically coupled to a maximum of a finite $E_6$ flavor symmetry. If singular $dP_6$ is indeed the correct description for the $I_1$ E-string theory, then it must be possible for it to admit a geometric coupling to affine $E_8$ flavor symmetry, since a model having a $(-1)$-curve carrying $I_1$ intersecting transversally a non-compact curve carrying $II^*$ (i.e. $E_8$) seems to be a consistent F-theory configuration.

Another question that might be helpful: Is it possible to deform the Weierstrass model (4.29) such that it becomes the above-mentioned model carrying $I_1$ intersecting $E_8$ flavor?

In any case, as the current draft stands, the arguments presented in prior literature (like reference [97] of the paper or the original atomic classification paper) arguing that there is a single E-string theory seem to strongly outweigh the arguments presented in the current draft arguing that there are three different E-string theories.

Notice also that similar questions can be asked for $I_0$, $I_1$ and $II$ fibers over a $(-2)$-curve, where it is widely believed that they all lead to $A_1$ $\mathcal{N}=(2,0)$ SCFT possibly with extra hypers. If authors' claim about three different E-string theories is correct, then we would also most likely obtain three different $A_1$ $\mathcal{N}=(2,0)$ SCFTs, but it is hard to imagine how to see the difference between the three different $A_1$ $\mathcal{N}=(2,0)$ SCFTs by using the various other string theory constructions of the theory.

Report

I would request the authors to revise their paper in light of the above comments.

Requested changes

If the authors have stronger arguments in favor of there being three different E-string theories, they should make a revision reflecting them.

If the authors do not have any other arguments, then I suggest them to resort to the conservative possibility that the type $I_1$ and type $II$ E-string theories are just the standard type $I_0$ E-string theory possibly with extra hypers. In this scenario, I would suggest them to modify the abstract and other places in the paper where they discuss these E-string theories.

(Actually, if the authors have strong arguments, then they should perform a systematic general analysis of the difference in the 6d theories coming from the difference in different fiber types having same associated gauge groups, and come out with another paper, as it would be a pretty exciting development deserving its own paper.)

---

## Round 3 · Referee Report · Anonymous (Referee 2) · 2022-1-31

Strengths

  1. The manuscript studies Calabi-Yau threefold singularities $C^3/\Gamma$ with a finite subgroup $\Gamma$ of SU(3) and 5d SCFTs constructed from M-theory compactified on the threefolds. This largely generalizes previous studies of 5d SCFTs using singular 3-folds and can provide a systematic approach to more directly explore the physics of 5d SCFTs at the origin of (extened-)Coulomb branches.

  2. New relationships between group theoretic data of $\Gamma$ and physical properties of 5d SCFTs such as Coulomb branch rank and flavor rank are elucidated through the 3d McKay correspondence, which can be used to correctly identify the associated 5d SCFT for a given orbifold singularity.

  3. The paper suggests a new method using the McKay quiver of $\Gamma$ to read off the 1-form symmetry of the 5d SCFT arising from $C^3/\Gamma$.

  4. A large number of concrete examples are presented as applications of the new findings in this paper.

Weaknesses

  1. In section 4.2.1, the authors conjecture that a Kodaira $I_1$ fiber over a (-1)-curve engineers a new 6d (1,0) SCFT which is different from the rank-1 E-string theory. This argument is based on the observation that the dimension of the Higgs branch of this theory is bigger than that of the rank-1 E-string theory. However, as they also mention, it seems to be more likely that this theory is equivalent to the rank-1 E-string theory together with 4 neutral hypermultiplets because they already identify the neutral hypermultiplets at the intersection point between $I_{4,s}$ and $I_1$ signular fibers and the dimension of the Higgs branch after subtracting the contribution from these neutral hypermultiplet agrees with that of the E-string theory. Thus, I think they need more evidence to support their conjecture for the new 6d SCFT.

  2. In section 4.2.2, the authors propose that the 5d SCFT from the resolution of $C^3/\Delta(48)$ is the rank-1 Seiberg $E_5$ theory. This requires an equivalence between a singular surface $S_1$ having a double point singularity and a smooth del Pezzo surface $dP_5$. This equivalence is not clearly explained.

Report

I believe that the results presented in this paper are genuinely new and will provide an important contribution to the field. I therefore recommend publishing the paper after the minor changes have been taken into account.

Requested changes

Two weaknesses pointed out above need to be clarified. Computations of topological data such as triple intersection numbers of the compact 4-cycles and the intersection of the 4-cycles with the 2nd Chern class of the 3-folds using the geometric constructions of section 4.2. may be able to provide evidences for the results and conjectures.

---

## Round 3 · Referee Report · Anonymous (Referee 3) · 2022-3-11

Strengths

  1. Comprehensive analysis of 5d/6d SCFTs on $\mathbb{C}^3/\Gamma$.
  2. Introduces new geometric methods to study generalized symmetries of such theories.
  3. Provides data base with details that are useful for future research on these topics.

Weaknesses

  1. Discrepancy between the (perceived) claim of new 6d rank 1 SCFTs in the abstract, and the evidence provided in Section 4.2.1 and 4.2.2
  2. Section 2.3.1. lacks some justification/explanation for conjecture and procedure to identify physics from geometry.

Report

The article presents important new results based on novel, generally clearly written, synergy of techniques, which is of broad interest to the community.
Up to the issues pointed out by the other referees, as well as the proposed changes below, I would recommend publication.

Requested changes

  1. This follows the requests of the previous reports. Please clarify why the results would point towards a new 6d rank-1 SCFT, or modify the abstract accordingly. Perhaps it would be helpful to comment on the fate of the 4 neutral hypermultiplets in 6d after reducing to 5d. Are there also additional hypermultiplets in the 5d interpretation of the resolution that exceeds the spectrum of an $E_5$ theory?
  2. What are the motivations/justification for Conjecture 1 on page 14? Is it just based on the patterns of the examples that have been studied? Or is there some physical motivation for it to hold?
  3. The process of "gluing" is never explained before the first paragraph on page 15. I believe it would make the article more self-contained, if more explanation for the validity and motivation of this gluing in regard to 5d theories is provided, especially since the only "explanatory" sentence here is a reference to 2d.
  4. There are a number of typos and somewhat unclear formulations that could be corrected. An incomplete list of these, which should be fixed, are 4.1: Multiple instances of "Mckay" instead of "McKay". 4.2: In the paragraph after Conjecture 1 on page 14: does the "simpler method" of writing down abelian subgroups, outlined in the second sentence, help with enumerating / constructing non-abelian subgroups, which are mentioned in the first sentence? 4.3: The $T_N$ theory is constructed with $\Gamma = \mathbb{Z}_N \times \mathbb{Z}_N$; in the last paragraph before Section 2.3.2, there is the mention of "subgroups $\mathbb{Z}^3_N \subset \Gamma$", which presumably is not meant as $\mathbb{Z}_N \times \mathbb{Z}_N \times \mathbb{Z}_N$.

---

## Editorial Decision

resubmitted